# Comparisons of bispectral and polarimetric retrievals of marine boundary layer cloud microphysics: Case studies using a LES-satellite retrieval simulator

Daniel J. Miller[1,3], Zhibo Zhang[1,2], Steven Platnick[3], Andrew S. Ackerman[4], Frank Werner[2], Celine Cornet[5], Kirk Knobelspiesse[3]

[1]Physics, UMBC, Baltimore, 21250, USA
[2]Joint Center for Earth Systems Technology, UMBC, Baltimore, MD, USA
[3]NASA Goddard Space Flight Center, Greenbelt, MD, USA
[4]NASA Goddard Institute for Space Studies, New York, NY, USA
[5]Laboratoire d'Optique Atmosphérique, Université des Sciences et Technologies de Lille, Villeneuve d'Ascq, France

*Correspondence to*: Daniel J. Miller (Daniel.J.Miller@nasa.gov)

**Abstract.** Many passive remote sensing techniques have been developed to retrieve cloud microphysical properties from satellite-based sensors, with the most common approaches being the bispectral and polarimetric techniques. These two vastly different retrieval techniques have been implemented for a variety of polar-orbiting and geostationary satellite platforms, providing global climatological datasets. Prior instrument comparison studies have shown that there are systematic differences between the droplet size retrieval products (effective radius) of bispectral (e.g. MODIS, Moderate Resolution Imaging Spectroradiometer) and polarimetric (e.g. POLDER, Polarization and Directionality of Earth's Reflectances) instruments. However, intercomparisons of airborne bispectral and polarimetric instruments have yielded results that do not appear to be systematically biased relative to one another. Diagnosing this discrepancy is complicated, because it is often difficult for instrument intercomparison studies to isolate differences between retrieval technique sensitivities and specific instrumental differences such as calibration, atmospheric correction, etc. In addition to these technical differences the polarimetric retrieval is also sensitive to the dispersion of the droplet size distribution (effective variance), which could influence the interpretation of the droplet size retrieval. To avoid these instrument-dependent complications, this study makes use of a cloud remote sensing retrieval simulator. Created by coupling a large eddy simulation (LES) cloud model with a 1-D radiative transfer model, the simulator serves as a test bed for understanding differences between bispectral and polarimetric retrievals. With the help of this simulator we can not only compare the two techniques to one another (retrieval intercomparison), but also validate retrievals directly against the LES cloud properties. Using the satellite retrieval simulator we are able to verify that at high spatial resolution (50 m) the bispectral and polarimetric retrievals are highly correlated with one another within expected observational uncertainties. The relatively small systematic biases at high spatial resolution can be attributed to different sensitivity limitations of the two retrievals. In contrast, a systematic difference between the two retrievals emerges at coarser resolution. This bias largely stems from differences related to sensitivity of the two retrievals to

unresolved inhomogeneities in effective variance and optical thickness. The influence of coarse angular resolution is found to increase uncertainty in the polarimetric retrieval, but generally maintains a constant mean value.

## 1 Introduction

The cloud droplet size distribution (DSD) is an important microphysical property of liquid-phase clouds. Given the cloud water content, it largely determines the shortwave radiative effects of clouds (Twomey, 1977). It also plays a critical role in cloud-precipitation processes (Pruppacher and Klett, 1978). As a result, anthropogenic perturbation to the DSD could lead to a variety of cloud property changes with significant climate implications (Lohmann et al., 2007).

Many satellite-based techniques have been developed to retrieve cloud DSD properties from regional to global scales. These techniques typically infer DSD properties based on an assumed size distribution shape, characterized by an effective radius ($r_e$), and an effective variance ($v_e$). One such retrieval method is called the bispectral total reflectance technique, hereafter referred to as the "bispectral technique," which simultaneously retrieves cloud optical thickness ($\tau$) and $r_e$ from a pair of cloud reflectances, typically one in the visible to near infrared (VNIR) and the other in the shortwave infrared (SWIR) or midwave infared (MWIR) spectral range (Nakajima and King, 1990b). This retrieval technique has been implemented for numerous satellite and airborne instruments, such as the Moderate Resolution Imaging Spectro-radiometer (MODIS, (King et al., 2003; Platnick et al., 2003; 2017)), the Spinning Enhanced Visible and Infrared Imager (SEVIRI, (Roebeling et al., 2006)), and the Suomi National Polar-orbiting Partnership Visible Infrared Imaging Radiometer Suite (Suomi NPP VIIRS, (Rosenfeld et al., 2014)).

A second, fundamentally different, retrieval technique is the multi-angular polarimetric reflectance technique, hereafter referred to as the "polarimetric technique". This retrieval requires multi-angular observations of the polarized reflectance in the cloudbow scattering region. In addition to $r_e$, the polarimetric technique can also retrieve $v_e$ (Bréon and Goloub, 1998). Global retrievals using the polarimetric technique were first demonstrated by the Polarization and Directionality of Earth Reflectance (POLDER, (Deschamps et al., 1994)) instruments operating from 1996 to 2013 on three different satellite platforms. The Aerosol Polarimetry Sensor (APS, (Mishchenko et al., 2007)) would have been the first space-borne multi-angular polarimeter from U.S. to provide global aerosol and cloud property retrievals. Unfortunately, it was lost as a result of the satellite launch failure in 2011, which suddenly interrupted development of polarimetric-based remote sensing in the U.S. Recognizing the great potential of polarimetric techniques for aerosol and cloud remote sensing, NASA has invested heavily in recent years on the development of airborne polarimeters, such as the Research Scanning Polarimeter (RSP, (Cairns et al., 1999)), the Airborne Multi-angle Spectro-Polarimetric Imager (AirMSPI, (Diner et al., 2013)) and the Airborne Hyper-Angular Rainbow Polarimeter (Air-HARP, (Martins et al., 2017)). Moreover, several space-borne missions are in development, such as the Multi-Angle Imager for Aerosols (MAIA, (Liu and Diner, 2017)), HARP, the Plankton, Aerosols, Cloud, ocean Ecosystem mission (PACE) and the Multi-viewing, Multi-channel, Multi-polarization imaging mission (3MI, (Marbach et al., 2013)). Each of these missions will have a multi-angular polarimeter on-board. In

the foreseeable future, we may expect to have operational global retrievals of cloud droplet size distributions from both bispectral and polarimetric methods.

The bispectral and polarimetric remote sensing techniques are the primary tools we have to obtain DSD observations on a global scale. It is therefore essential to identify and explain differences between the two techniques so we can better understand their respective advantages and limitations. A satellite retrieval intercomparison of POLDER and MODIS $r_e$ retrievals by Bréon and Doutriaux-Boucher (2005) represented one of the first attempts to identify and understand the differences between the two techniques. The main finding from this study is that the bispectral-based MODIS retrieval of $r_e$(2.13µm) (using the 2.13 µm SWIR band) is persistently 2 µm larger than the 150 km scale polarimetric-based POLDER retrieval over ocean, despite a close correlation between the two. A variety of factors, from differences in sensitivity to cloud vertical profile to influence of cloud horizontal inhomogeneity, have been suggested to explain this difference. However, as pointed out by the authors, all these factors might contribute to the difference. It is difficult, if not impossible, to untangle them in observations and determine their relative importance. In addition, POLDER observations in this study were aggregated from the nominal 6 km spatial resolution to a much coarser 150 km resolution to achieve the angular resolution needed to resolve the cloud bow. The vast difference in spatial resolution (i.e., 150 km for POLDER and 1 km for MODIS) makes the interpretation of the 2 µm $r_e$ difference between the two retrievals even more difficult.

A more recent study by Alexandrov et al. (2015) is based on observations from the recent sub-orbital Polarimeter Definition Experiment (PODEX) in 2013. In that study, the polarimetric $r_e$ retrievals for marine stratocumulus decks off the California coast from the airborne RSP instrument are compared to collocated bispectral retrievals from the Autonomous Modular Sensor (AMS). Interestingly, the two retrievals are found to be in close agreement, with a correlation of 0.928 and negligible bias of less than a micron. Beyond the clear instrument differences of the Alexandrov et al. (2015) and Bréon and Doutriaux-Boucher (2005) studies, it is still unclear how well the bispectral and polarimetric retrievals should compare to one another and what situations might cause them to differ, raising numerous questions and motivating this study.

A great challenge facing these observational studies is the intertwining of various instrument and scene dependent factors that lead to retrieval differences. For example, the polarimetric and bispectral methods have different sensitivity to the cloud vertical profile, and at the same time they are also both affected by cloud horizontal inhomogeneity (Zinner et al., 2010; Zhang et al., 2012; 2016; Miller et al., 2016). It is difficult, if not impossible, to disentangle these factors based on observations alone. This study approaches the intercomparison of bispectral and polarimetric retrievals through a different route: rather than use observational remote sensing data, synthetic retrievals are generated from large-eddy simulations (LES) of clouds. Modeling radiative transfer in an LES scene to obtain total and polarized reflectances opens up the possibility of using the LES to perform synthetic bispectral and polarimetric retrievals. This retrieval simulator framework has proven to be a useful tool in other cloud remote sensing studies (Miller et al., 2016; Zhang et al., 2012). Using this idealized simulation at high spatial resolution, we can attempt to parse the effects of unresolved sub-pixel inhomogeneity, spatial resolution, and angular resolution on the intercomparison of polarimetric and bispectral retrievals. The scale of the LES simulations (~10 km) in this study prevents us from examining resolutions as large as the standard POLDER retrieval

(~150 km), but we are able to advance understanding how spatial resolutions from 50 m to 1 km. These scales are suitable for airborne instrument comparisons, which certainly fall somewhere in this range. The use of a satellite retrieval simulator opens up two unique opportunities for developing and studying cloud microphysical retrievals: First, it provides the means to compare retrievals directly to LES cloud microphysics. Second, it allows us to perform a retrieval technique intercomparison that is independent of instrument characteristics and other differences that complicate observational studies. This study focuses on three particular questions:

- How well do the bispectral and polarimetric retrievals perform relative to the LES fields used as input to the retrievals?

- How do the bispectral and polarimetric retrieval techniques compare to one another at high spatial resolution?

- How are the bispectral and polarimetric retrieval techniques sensitive to specific observational conditions (i.e., the influence of spatial and angular resolution)?

The rest of the paper is organized as follows: Section 2 provides a brief introduction to the theoretical basis of the two retrieval techniques; Section 3 describes the LES-based satellite retrieval simulations used in this study; the comparisons between the two techniques based on the LES cases are presented in Section 4; followed by summary and discussion in Section 5.

## 2 Background

## 2. Cloud microphysical and optical properties

In satellite remote sensing DSDs are often described using theoretical distributions that fit well with in situ observations, in addition to being mathematically convenient (Deirmendjian, 1964; Tampieri and Tomasi, 1976; Martin et al., 1994; Miles et al., 2000). A popular theoretical DSD is the gamma distribution proposed by Hansen and Travis (1974):

$$N\left(r;r_e,v_e\right) \equiv N_0 C r^{(1-3v_e)/v_e} \exp\left[-r/\left(r_e v_e\right)\right],$$
(1)

where the independent variable $r$ is the cloud droplet radius, $N(r)$ is the droplet size distribution, $N_0$ is the droplet number concentration, and C is a normalization constant. The two distribution parameters are the effective radius ($r_e$) and the effective variance ($v_e$) of the DSD:

$$r_e \equiv \frac{\int_0^\infty Q_e(r) r^3 N(r) dr}{\int_0^\infty Q_e(r) r^2 N(r) dr} \approx \frac{\int_0^\infty r^3 N\left(r;r_e,v_e\right) dr}{\int_0^\infty r^2 N\left(r;r_e,v_e\right) dr} = \frac{\langle r^3 \rangle}{\langle r^2 \rangle},$$
(2)

$$v_e \equiv \frac{1}{r_e^2} \cdot \frac{\int_0^\infty Q_e(r)(r-r_e)^2 r^2 N(r) dr}{\int_0^\infty Q_e(r) r^2 N(r) dr} \approx \frac{1}{r_e^2} \cdot \frac{\int_0^\infty (r-r_e)^2 r^2 N\left(r;r_e,v_e\right) dr}{\int_0^\infty r^2 N\left(r;r_e,v_e\right) dr} = \frac{\langle r^4 \rangle \langle r^2 \rangle}{\langle r^3 \rangle^2} - 1.$$
(3)

where $\langle r^n \rangle = \int_0^\infty r^n N(r) dr$ is the $n^{th}$ moment of the DSD. For the spectral bands relevant to this study, the extinction efficiency ($Q_e$) is approximately constant (i.e., $Q_e(r) \sim const. = 2$). Thus, the relationships between $r_e$ and $v_e$ can be conveniently reduced to relations between arithmetic moments of the DSD. The DSD plays an important role in defining the bulk optical properties of a cloud. The optical property libraries used in this study are based on single-scattering Mie calculations of monodisperse droplet optical properties that are averaged with respect to size, according to the gamma DSD (Wiscombe, 1979). In addition, these single-scattering optical properties are averaged with respect to wavelength over an instrument-specific spectral response function (based on MODIS bands in this study) and solar source functions (Planck blackbody function (Planck, 1914)). The single-scattering bulk cloud optical properties are subsequently used to run radiative transfer calculations for the creation of the so-called bispectral reflectance look-up-table (LUT). This LUT is made up of pre-calculated reflectances of plane-parallel and homogeneous (PPH) clouds over a high-resolution grid of combinations of $\tau$, $r_e$, and $v_e$. Here, $\tau$ is defined in terms of the DSD:

$$\tau_{tot,\lambda} \equiv \int_{TOA}^0 \left[ \int_0^\infty Q_{e,\lambda}(r) \pi r^2 N_0 n(r) dr \right] dz$$

$$(4)$$

## 2.2 Bispectral and Polarimetric Retrieval Methods

The bispectral method retrieves $\tau$ and $r_e$ simultaneously from a pair of observed cloud reflectances, typically using a combination of VNIR and SWIR/MWIR bands. The VNIR band, with relatively negligible liquid water droplet absorption, and the SWIR/MWIR band, where droplets are moderately absorptive, can be used to remotely infer $\tau$ and $r_e$ because of this difference in sensitivity to multiple scattering (thickness) and absorption (droplet size). This method is usually implemented using a LUT, shown graphically in Figure 1(a), which has a fixed $v_e$. Cloud reflectance in the VNIR band (centered around 0.865 µm) increases with $\tau$ (gray) for a fixed $r_e$, while the reflectance in the MWIR band (centered around 3.75 µm) decreases with $r_e$ (colored) when $\tau$ is fixed. The retrieved properties are obtained by performing a two-dimensional inverse interpolation between observed reflectance and the $\tau$-$r_e$ grid. A notable characteristic of the bispectral LUT is that when the optical thickness is low ($\tau < 3$), the isolines of the LUT are more densely packed and less orthogonal, which results in reduced sensitivity and increased retrieval uncertainty (Werner et al., 2013). The bispectral technique is not particularly sensitive to $v_e$, so typically a fixed value is assumed (e.g., $v_e = 0.1$ in the operational MODIS retrieval, though it is kept as an error source in calculating pixel-level uncertainties). While different combinations of bands are used to perform the bispectral retrieval, in this study we focus on VNIR reflectances centered on 0.865 µm with the second band is selected from either a 2.13 µm centered SWIR band or a 3.75 µm centered MWIR band There are consequences for the $r_e$ retrieval depending on the particular set of bands selected. For example, a strongly absorbing SWIR/MWIR band limits penetration into the cloud and

as a result the retrieved $r_e$ is vertically weighted toward the microphysics prevalent in the uppermost part of the cloud (Platnick, 2000).

For the polarimetric retrieval, the angular pattern of the linearly polarized reflectance[1] is the source of sensitivity to cloud microphysical properties. Polarized reflectances are dominated by single-scattering because multiple scattering induces depolarization. As a result, the single-scattering polarized phase functions ($-P_{12}$) shown in Figure 1 (b) and (c) are good approximations to the observed angular pattern of polarized cloud reflectances (Bréon and Goloub, 1998). These phase functions demonstrate the sensitivity of the polarimetric retrieval to both $r_e$ and $v_e$. As $r_e$ increases in Figure 1(b) the supernumerary bow peaks (around a scattering angle of 142°) become narrower and shift toward smaller scattering angles. In contrast, as $v_e$ increases in Figure 1(c) the supernumerary bow peaks erode in magnitude, eventually smoothing out for broad DSDs ($v_e$>0.15). A consequence of this erosion of the supernumerary peaks is that the polarimetric retrieval has less sensitivity to both $r_e$ and $v_e$ for very broad DSDs. The polarimetric retrieval does not significantly rely on multispectral information, although observations in several bands may help provide stronger observational constraints due to the shift in the supernumerary bows with changing wavelength (refer to figure 3 of Bréon and Goloub (1998). The dominance of the single-scattering contributions to the polarized reflectance leads to cloud retrievals that represent microphysical properties with a mean penetration depth of $<\tau_{SS}>\leq0.5$ and sensitivity that saturates for optical depths greater than ~3 from the cloud top. The polarimetric retrieval is often based on a parametric curve fitting retrieval algorithm like the one presented in Alexandrov et al. (2012b), although there are other techniques (e.g., the Rainbow Fourier Transform technique of Alexandrov et al. (2012a), which can retrieve DSD's with arbitrary mathematical form.) The parametric technique relies on a library of $-P_{12}$ curves with varying $r_e$ and $v_e$ that are parametrically scaled and adjusted to fit the observed reflectance via a nonlinear least squares optimization procedure. This process yields the phase function that best matches the angular pattern of the observation, thus determining the $r_e$(pol) and $v_e$(pol) retrieval. The polarimetric method described above does not result in a retrieval of $\tau$; however, it can still be obtained by implementing a simplified variant of the bispectral $\tau$ retrieval. With simultaneous measurements of the total reflectance in a VNIR band and the $r_e$(pol) retrieval, a VNIR-only LUT curve can be used to perform a 1-D interpolation of the corresponding bispectral LUT curve for $R_{VNIR}(r_e(pol), \tau)$.

Both bispectral and polarimetric techniques are susceptible to a variety of retrieval uncertainties. The main objective of this study is to understand how the retrieval uncertainties influence each technique and whether they can lead to deviation between the two techniques in terms of retrieval results. In this study, we focus on five major sources of retrieval uncertainty for both techniques:

1) Cloud vertical profile: In the operational retrievals, both bispectral and polarimetric techniques assume vertically homogenous clouds. However, clouds in reality often have significant vertical variability resulting from various processes (e.g., condensational growth, coalescence, sedimentation, entrainment). Deviation from the assumed profile gives rise to many questions. For example, how do we interpret the $r_e$ and $v_e$ retrievals based on the homogenous cloud assumption? To

---

Note that throughout this paper, we will refer to "linearly polarized reflectances" simply as "polarized reflectances" in recognition of the negligible contribution of circularly polarized light in the atmosphere (Hansen, 2010).

what extent does cloud vertical profile influence the bispectral and polarimetric techniques? Note that Platnick (2000) developed a method utilizing the so-called "vertical weighting function" to interpret the $r_e$ retrieval from the bispectral method for clouds with vertically varying $r_e$ profile. Recently, Alexandrov et al., (2012b) took an approach that focused on the vertical weighting of the droplet size distribution to interpret the $r_e$ and $v_e$ retrievals from the polarimetric technique. Miller et al. (2016) demonstrated the usefulness of this vertical weighting approach for understanding both bispectral and polarimetric $r_e$ retrievals. In Section 4.1, we will apply a vertical weighting function for both techniques on the basis of the LES cloud fields, to help understand if cloud vertical structure could lead to significant differences between the two techniques.

2) Sensitivity to observational uncertainty: The uncertainties associated with observations of total and polarized reflectances can differ, indicating that uncertainty may also impact bispectral and polarimetric retrievals differently. Additionally, the two retrievals rely on different number of uncertain observations; a pair of uncertain total reflectances (bispectral) as compared to numerous uncertain polarized reflectances (polarimetric). Furthermore the different algorithmic approaches, two-dimensional interpolation vs. nonlinear optimal curve fitting introduce additional layers of complexity in terms of the impact of uncertainty. The impact of uncertainty on retrieval results for each method are highlighted and explored in section 4.2.

3) Reduced sensitivity: It can be clearly seen from Figure 1 (a) that when clouds are optically thin ($\tau$<3), the LUT for the bispectral retrieval becomes less orthogonal and the isolines of $r_e$ become more densely packed. This reduction in sensitivity can lead to significant retrieval uncertainties in bispectral techniques for optically thin clouds ($\tau$<3). Similarly, the sensitivity of the polarimetric technique to $r_e$ and $v_e$ is reduced when DSD becomes very broad (i.e., $v_e$>0.15), in which case the supernumerary bow features are barely distinguishable (Figure 1 (c)). In Section 4.3 we will investigate the impacts of the reduction of sensitivity on retrieval consistency between the two techniques.

4) Sub-pixel inhomogeneity: The impact of spatial resolution and unresolved sub-pixel cloud inhomogeneity on bispectral retrievals has been well studied (Zhang and Platnick, 2011; Zhang et al., 2012; 2016). An important conclusion from these studies is that the so-called plane-parallel homogenous bias (PPHB) can cause the bispectral technique to significantly overestimate $r_e$. In contrast, the sensitivity of the polarimetric retrieval to unresolved sub-pixel inhomogeneity and resolution has not been thoroughly studied. In Section 4.4 we will compare the impacts of sub-pixel inhomogeneity on bispectral and polarimetric techniques, and investigate whether it can cause deviation between the two techniques.

5) Angular resolution and sampling for polarimetric technique: In addition to spatial resolution, angular resolution and sampling is also important for the polarimetric technique. A coarse angular resolution may not be able to resolve the feature of the supernumerary bows. Similarly, if the scattering angles corresponding to the supernumerary bows are not or only partly sampled, then the polarimetric technique may not have enough information content for retrieval. This issue will be discussed in Section 4.5.

## 3 Model and Methodology

The satellite retrieval simulator implemented in this study is built around an LES model (DHARMA) with bin microphysics (Ackerman et al., 2004; Zhang et al., 2012; Miller et al., 2016). The LES provides freely evolving 3-D cloud microphysical properties, which are used as reference when comparing to numerically simulated retrievals. The LES in this study adopts 25 droplet size bins to represent droplet size distributions (Ackerman et al., 1995). The optical properties of each size bin are computed by bulk averaging Mie scattering properties over a highly resolved flat sub-bin droplet size distribution. The optical properties of each bin are provided as input to radiative transfer simulations based on the size distributions of the LES cloud fields. Vector radiative transfer calculations are performed using a polarized doubling-adding technique (PDA) to produce 1-D total and polarized reflectances at the horizontal resolution of the LES grid (described below) (De Haan et al., 1987). The sole consideration of 1-D retrievals avoids 3-D radiative effects and focuses this study on retrieval technique differences rather than on radiative processes. A future study will focus on the comparison of 3-D retrievals to these 1-D bispectral and polarimetric retrievals. The radiative transfer modeling in this work is performed for numerous solar zenith angles (SZA=[20, 40, 60]°), viewing zenith angles (VZA=[−70 : +70]°), and a constant relative azimuthal angle ($\Delta\Phi$ = 30°). The VZA resolution results in a scattering angle ($\Theta$) resolution on the order of 0.5°. Reflectances in spectral bands (based on MODIS spectral response functions) are centered on 0.865, 2.13, and 3.75 µm wavelengths. Total reflectances in all bands are used to produce bispectral retrievals, whereas linearly polarized reflectances in the 0.865 µm band are used to produce polarimetric retrievals. Subsequently, bispectral and polarimetric retrievals are performed on the simulated reflectances to obtain $r_e$, $v_e$, and $\tau$ retrievals. Bispectral and polarimetric retrievals are performed over a subset of observation geometries, with bispectral retrievals performed for VZA=[50, 40, 30, 20, 10, 0, −10]° and all SZA. Meanwhile, the polarimetric retrievals are performed for a SZA=20° and a range of VZA=[0:27]° that result in reflectances spanning scattering angles required to observe the primary and supernumerary bow features (i.e., $\Theta$=[135:160]°). Reflectances are also aggregated from the 50 m native LES resolution up to coarser 100, 200, 400, and 800 m horizontal resolutions to reflect the influence of different remote sensing footprints. The retrievals in this study are also performed at all of the footprint resolutions. The bispectral LUT implemented in this study spans microphysical properties $r_e$=[2:30] µm in steps of 0.5 µm and $v_e$=[0.01:0.11] in steps of 0.01. The $\tau$ retrieval in this study is anchored to the 0.865 band optical properties and spans $\tau$=[0.1:100] with 101 logarithmically spaced grid points. Including $v_e$ variability in the bispectral LUT allows for the comparison of standard MODIS-like retrievals (the $v_e$=0.1 LUT) to retrievals with other $v_e$ assumptions. The bispectral retrieval is then accomplished by performing a 2-D linear interpolation of the observed reflectances and inverting between the reflectance and retrieval space. For the polarimetric retrieval, the polarimetric phase function library spans $r_e$=[2:40] µm in steps of 0.25 µm and $v_e$=[0.01:0.3] in steps of 0.01. The polarimetric retrieval implemented in this study is based on the approach of Alexandrov et al. (2012a), fitting the polarized phase functions in their eq. (3) to the modeled polarized reflectances of the LES scene. The optimal parametric fit in the $-P_{12}$ library is determined by using a Levenberg-Marquardt nonlinear least squares algorithm. This optimal phase function is then used to identify the

corresponding $r_e$(pol) and $v_e$(pol) retrieval. As previously stated in section 2.2 the polarimetric retrieval of $\tau$ is accomplished by using a constrained 1-D version of the bispectral LUT.

The LES cloud fields are used not only to drive the radiative transfer simulations, but also to help interpret and understand the retrieval results. As mentioned in Section 2.2, it is not trivial to interpret the $r_e$ and $v_e$ retrievals based on the homogenous cloud assumption when the cloud has significant vertical structure. To address this issue, for each LES column with detailed vertical profiles of DSD, we derive two reference variables $r_e$(VW) and $v_e$(VW) from the vertical weighted (VW) integration of the DSD profile. The vertical integration is weighted by a function to account for the penetration depth and multiple scattering of radiation in the corresponding wavelength associated with each particular retrieval. Thus, $r_e$(VW) and $v_e$(VW) should be comparable to the retrieved $r_e$ and $v_e$ from the simulated reflectance (Alexandrov et al., 2012b; Miller et al., 2016; Zhang et al., 2017). The method of vertical weighting in this study is described in detail in section 2 of Miller et al. (2016), however in this study we have modified the vertical weighting function to account for multiple scattering. Motivated by the convenience and flexibility of the parametric approach proposed in eq. 4 of Zhang et al. (2017), we implement a two-variable parametric vertical weighting function:

$$W(\tau) = c\tau^b \exp\left[-a\tau\left(\frac{1}{\mu} + \frac{1}{\mu_0}\right)\right], \qquad (4)$$

where the new parameters $a$ and $b$ are introduced to account for the influence of multiple scattering effects not originally considered in the vertical weighting function discussed previously in Miller et al. (2016). The parameter $a$ scales the optical depth, modelling the enhanced transmission caused by multiple scattering whereas the parameter $b$ produces a peaked vertical weighting function associated with the expected penetration depth of the reflected light and $c$ is the normalization factor. Each of these parameters is strictly positive, and for $a$=1 and $b$=0 we obtain the original single scattering vertical weighting used in Miller et al. (2016). For smaller values of $a$ and larger values of $b$, the vertical weighting function extends deeper into the cloud, leading to droplet size distribution properties deeper in the cloud contributing more information to the vertically weighted value. For the polarimetric retrieval, $a$=1 and $b$=0 were selected due to the dominance of single scattering in polarized reflectances. In contrast, multiple scattering can significantly impact total reflectances. For total reflectances a single value of $a$ and $b$ was selected for each spectral band and observation geometry based on coefficients that fit best to numerically calculated vertical weighting functions based on the method presented in eq. 4 of Platnick (2000).[2] Generally, we found that $a$(3.75 µm) was larger than $a$(2.13 µm), as would be expected because of stronger absorption in 3.75 µm reducing transmission into the cloud. We also found that $b$ was dependent on observation geometry (scattering angle) and $b$(3.75 µm) was less than $b$(2.13 µm) because multiply scattered light in the 2.13 µm band can penetrate deeper into the cloud before scattering back out. In addition to $r_e$(VW) and $v_e$(VW), we also derive $\tau_{\mathrm{LES}}$ for each LES column simply by

---

[2] The radiatively derived vertical weighting of Platnick (2000) implicitly depends on the $r_e(z)$ profile whereas a fixed parameter vertical weighting described here does not. However, the importance of this difference should be less than the vertical variability of optical depth or extinction cross section.

integrating the extinction coefficient (for $\lambda$=0.865 μm) from cloud bottom to cloud top. The $r_e$(VW), $v_e$(VW) and $\tau_{LES}$ are used as references in the retrieval and LES property comparison in section 4.1 to understand the differences between the retrievals and the original LES fields. After obtaining the $r_e$(VW), $v_e$(VW) and $\tau_{LES}$ at the 50 m native LES resolution, they are aggregated to 100, 200, 400, and 800 m to help interpret the retrievals at these coarser resolutions. It is important to note

that there is a subtle difference between directly aggregating $r_e$(VW) or $v_e$(VW), and aggregating the DSD (i.e., $N(r)$) first and then deriving the corresponding $r_e$(VW) and $v_e$(VW). The differences between the two methods are discussed in the Appendix. The main conclusion is that, although the two aggregation methods could be different in some hypothetical cases with unrealistically large variability in the unresolved microphysics, they are essentially equivalent for practical purposes. In this study, we simply aggregate $r_e$(VW) and $v_e$(VW) from the native LES resolution of 50 m to obtain average values at

desired resolution (e.g., 800 m).

Three LES cases are the focus of this study. The first (referred to as "ATEX clean" hereafter) and second ("ATEX polluted") cases are based on an idealized case study from the Atlantic Trade Wind Experiment (ATEX), with different aerosol loadings (Stevens et al., 2001). The ATEX cases are representative of a trade wind cumulus regime in which scattered cumuli rise into a thin, broken stratocumulus layer. The third case (referred to as "DYCOMS-II" hereafter),

originally presented in Stevens et al. (2005), is an idealized setup based on clouds observed during the second research flight (RF02) of the Second Dynamics and Chemistry of Marine Stratocumulus project (DYCOMS-II) (Stevens et al., 2003). This case is representative of nocturnal marine stratocumulus under a dry inversion. The DYCOMS-II case has a domain size of 6.4×6.4×1.5 km (128x128x96 grid points), while each of the ATEX simulations has a domain size of 7.2×7.2×3 km (144×144×200 grid points). The horizontal grid spacing of these LES cases is fixed at 50 m, while the vertical grid is

stretched, with a minimum spacing of 5 m near the surface and the capping temperature inversion to better resolve small-scale turbulence there. Further details of the model setup for the DYCOMS-II case are provided in Ackerman et al. (2009). The ATEX cases are updated model runs with increased spatial resolution that are similar to the cases discussed in Fridlind and Ackerman (2011). For each LES scene a snapshot of cloud microphysical and optical properties is saved every half hour after the first hour of each simulation, resulting in numerous cloud fields. A single time step of each of the cases was selected

to be the focus of this retrieval study, each occurring ~3 hours into the simulation.

The variability of cloud optical and microphysical properties in each of the LES cases is highlighted in Figure 2 and Table 1. Spatial inhomogeneity of both optical and microphysical properties of these scenes is evident, with the ATEX polluted and DYCOMS-II cases exhibiting lower spatial inhomogeneity and the ATEX clean case being more broken and inhomogeneous. One method for quantifying the optical inhomogeneity of a cloud scene is to use the sub-pixel

inhomogeneity index,

$$H_\sigma(\text{resolution}) = \frac{\text{std}\left[R_i(0.865 \text{ μm}, 50 \text{ m})\right]}{\text{mean}\left[R_i(0.865 \text{ μm}, 50 \text{ m})\right]}, \qquad (4)$$

where the numerator and denominator are the standard deviation and mean of the native LES resolution (50 m) reflectances within a coarser resolution pixel. Thus, the value of $H_\sigma$ is computed for a coarser spatial resolution pixel (800 m in Table 1) using the highest-resolution nadir viewing reflectances (50 m). The value of $H_\sigma$ increases with increasing sub-pixel inhomogeneity, making it a useful measure for unresolved cloud variability. In addition to optical inhomogeneity, each of the
LES scenes also has characteristically different microphysical properties. The average value of $r_e$(VW) of each scene varies, in part because of the initial background CCN in each particular case but also cloud top height variability. In these LES cases $v_e$ is spatially anti-correlated with $\tau$ and organized in a cellular structure — regions with higher $\tau$ tend to have smaller $v_e$(VW) and regions with lower $\tau$ tend to have large $v_e$(VW).

## 10   4 Results and Analysis

### 4.1 Retrieval and LES Property Comparison

Before comparing the retrieval results from the two techniques to one another, we must first carry out a comparison of LES and retrieval properties to assess and understand the differences between the retrieval results and the original LES cloud fields at the native 50 m spatial resolution. This is a necessary sanity check that will help understand the accuracy and
uncertainty of our retrieval routines. More importantly, this study will help to interpret the retrievals based on homogeneous cloud assumption when the LES cloud fields have naturally inhomogeneous vertical profiles. Note that the retrievals compared throughout the following sections are compared for all combinations of viewing and solar geometries indicated in the section 3.

The bispectral retrieval comparison to LES properties in Figure 3 depicts joint histograms of $r_e$ and $\tau$ retrievals
using both the 2.13 and 3.75 µm bands against the reference values derived from the LES fields, $r_e$(VW), $v_e$(VW), and $\tau_{\text{LES}}$. It is important to note that these joint histograms are presented as the logarithmic percent of the population, to emphasize deviations from the one-to-one line. Also, the mean regression biases reported throughout this study are stated relative to the plotted axes as, $\mu_{\text{bias}}$=<y−x> and $\mu_{|\text{bias}|}$=<|y−x|> (i.e., x and y denoting x and y axes). The two bispectral $r_e$ retrievals, $r_e$(2.13 µm) and $r_e$(3.75 µm), are in agreement with the LES ground-truth (Figure 3(a) and (b)) with good correlations, both
exceeding 0.95. The biases between these two retrievals and the LES properties differ slightly. Compared to the LES, both $r_e$ retrievals have relatively small sub-micron mean biases and the mean absolute biases are also sub-micron. Additionally, it is important to note a limitation of this population: none of the LES scenes in this study have a mean cloud top $r_e$ near 10 µm. To examine the two bispectral $\tau$ retrievals, $\tau$(2.13 µm) and $\tau$(3.75 µm) in Figure 3(c) and (d), we compare them in terms of percent differences, because the regression is so highly correlated (R>0.99). A slight systematic high bias on the order of 2-
5% exists. The origin of this high bias is likely associated with deviations of the droplet size distribution from the assumed gamma distribution form. The LES size distributions sometimes exhibit longer large-droplet tails than the assumed form. As explained earlier in Section 2.2, the bispectral method suffers from a reduction of retrieval sensitivity when clouds are

optically thin. Therefore, if we sample only LES columns that are optically thick ($\tau$>3) a substantial improvement in the regression correlations of the two $r_e$ retrievals (Figure 3(e) and (f)) is achieved. While not all retrievals in the $\tau$<3 population are biased, but the majority of extreme retrieval bias outliers belong to this thin cloud population. Even after this sub-selection of the data some outliers still remain. In particular, a small population of both $r_e$(2.13 µm) and $r_e$(3.75 µm) retrievals have biases exceeding $r_e$(VW) by as much as 20 µm. The cause of these outliers and some other differences between the retrievals and LES fields will be discussed in detail in section 4.3.

The polarimetric retrieval comparison to LES properties in Figure 4 depicts comparisons of the polarimetric retrievals, $r_e$(pol) $v_e$(pol), and $\tau$(pol). The $r_e$(pol) retrieval compares very well to $r_e$(VW) (Figure 4(a)), with a regression correlation exceeding 0.98, a mean bias of less than 0.1 µm. The quality of this retrieval comparison to LES properties also further supports the single scattering definition of $r_e$(VW) for the polarimetric retrieval. In contrast, the polarimetric retrieval of $v_e$(pol) reveals a regression against $v_e$(VW) (Figure 4(c)) that does not perform quite as well. In this case the regression correlation is much weaker (R=0.71) with a mean bias of −0.011. While the mean bias is on the order of the $v_e$ LUT grid spacing, it is clear that the regression correlation is poor because of a systematic low bias for $v_e$(VW) larger than about 0.15. It should also be noted that the increased concentration of $v_e$(pol) retrievals at $v_e$=0.3 is a result of the boundaries of the retrieval space, $v_e$=[0.01, 0.3]. The upper limit of which is a consequence of the gamma distribution of Hansen and Travis (1974) becoming monotonic for $v_e$≥0.3. Comparing only the population with $v_e$(VW)≤0.15 (not shown here) results in an improved correlation of R=0.81 with negligible mean bias. The $v_e$ retrieval quality also depends on the assumption that LES droplet size distributions are accurately described using a single-mode gamma distribution. The DSD's in the LES sometimes deviate significantly from this assumption. In the context of the parametric polarimetric retrieval used in this study this is difficult to remedy or address. However, a different polarimetric retrieval, the Rainbow Fourier Transform (RFT) introduced in Alexandrov et al. (2012a) offers the possibility of retrieving an arbitrary droplet size distribution shape. The final retrieval product, $\tau$(pol) (Figure 4(e)), indicates that a more accurate a priori $r_e$ and $v_e$ estimate has little impact on the retrieval of $\tau$. As explained earlier in Section 2.2, the polarimetric method suffers a reduction of sensitivity when the DSD is broad, a finding that is consistent with previous work (Alexandrov et al., 2012b). This explains, for the $r_e$(pol) retrieval, why limiting the regression population to LES columns with $v_e$(VW)≤0.15 in Figure 4(b) increases the correlation and decreases the absolute bias. This appears to be an indication of sensitivity to degradation of the supernumerary bow features for large $v_e$, features that are necessary for reliable $r_e$(pol) and $v_e$(pol) retrievals.

For $v_e$(pol) we find that by sampling LES columns that are optically thick ($\tau$>3), there is moderate improvement in the correlation and reduced biases (Figure 4(d)). This improvement stems from the correlation between the population of optically thin clouds and high $v_e$(VW) (Figure 4(f)) that are found near cloud edges in the LES scenes. It should be noted that an increased $\tau$ does not implicitly lead to better polarimetric retrievals, but here it is observed to be a consequence of aforementioned correlated relationship between DSD and optical properties.

**4.2 Sensitivity to Measurement Uncertainty**

The measurement uncertainties of total and polarized reflectances differ, leading one to expect that bispectral and polarimetric retrievals may have different sensitivities to uncertainty. Their relationships to uncertainty are further complicated by differences in retrieval approaches; namely interpolating two independent uncertain observations in a LUT (bispectral) or curve fitting through numerous observations that are each independently uncertain (polarimetric). Targeted uncertainties for cloud and aerosol remote sensing are δDOLP=0.5% in degree of linear polarization and δI=3% in total reflectance (Knobelspiesse et al., 2012). A simple propagation of uncertainty analysis yields a polarized reflectance uncertainty of δQ=2.5% (in the principal plane). Using these uncertainties as a starting point, we can perturb the LES reflectances with uncorrelated random noise and perform retrievals that we can then compare to the original unperturbed retrievals. Note that while the focus here is on uncorrelated randomly distributed noise, other sources of observational uncertainty exist and would need to be accounted for in the context of a specific instruments' uncertainty model. As the properties of particular instruments are not the focus of this study, we will focus on this more general uncertainty analysis.

For the bispectral retrieval, a randomly distributed reflectance perturbation between +/-3% was added to each LES reflectance. A histogram of the percent bias of bispectral retrievals of $r_e$ induced by the addition of the reflectance uncertainty is shown in Figure 5(a). The mean and standard deviation of these bias distributions are stated, allowing us to interpret the results. First, the introduction of uncertainty has very little impact on the mean bias of bispectral $r_e$ retrievals (on the order of 0.1%). Second, the introduction of uncertainty results in a broad distribution of $r_e$ retrieval biases with standard deviations of 5.44% and 4.02% for the 2.13μm and 3.75μm retrievals respectively. Together these two results indicate that biases associated with measurement uncertainty will not be systematic, with absolute variability on the order of 1 μm or less for droplet sizes below 20 μm (the most prevalent population in this LES study). The impact of observational uncertainty on all of the $\tau$ retrievals is the focus of Figure 5(b). The two bispectral retrievals, $\tau(2.13\mu m)$ and $\tau(3.75)$, each have very minimal mean biases of 0.1%. However, like the biases for the effective radius retrievals, the distribution of retrieval bias is broadened to standard deviations of 8.2% and 5.6% for $\tau(2.13\mu m)$ and $\tau(3.75)$ respectively. The polarimetric $\tau(pol)$ retrieval on the other hand, being methodically quite similar to the bispectral retrievals, exhibits a small systematic low bias of about -2.43% as shown in Figure 5(b). The origin of this systematic bias is a known characteristic of single-band optical thickness retrievals, and is clearly demonstrated in figure 1 of Marshak et al. (2006). The convexity of a single-band LUT curve produces low-biased retrievals for symmetrically distributed (or averaged) reflectances. The bias distribution also has a smaller variability (3.8%) than the two bispectral retrievals, likely because the uncertainty in the SWIR/MWIR band also (weakly) influences the bispectral $\tau$ bias distributions.

The consequences of measurement uncertainty are markedly different for the polarimetric retrieval. This is a result of the polarimetric retrieval being a search for a similar curve in the phase function library, making the deviations in the magnitude of observations in any one angle less important when searching for the optimal curve – and therefore discrete $r_e$ and $v_e$ combination. The discretely binned nature of the polarimetric retrieval makes description of bias distributions like the

ones in Figure 5 problematic. One way to describe how uncertainty in polarized reflectances influences polarimetric retrievals is to describe the population of retrievals that are unchanged, and the population of retrievals that changed. After the introduction of random noise 88.1% of the polarimetric $r_e$(pol) retrievals were unbiased, with 9.1% biased high by one grid point (+0.25 µm) and 2.7% biased low by one grid point (-0.25 µm). All together, these three populations accounted for

the vast majority (99.9%) of retrieval outcomes. The percent bias of the $r_e$(pol) retrieval had a mean of 0.06% and a standard deviation of 0.78%. These results agree with previous studies, for example the finding of Shang et al. (2015) indicating that the POLDER retrieval performed well as long as reflectance uncertainty was less than 10%. It should be noted however that the sensitivity to uncertainty is also tied to the number of angular measurements available, and the properties of the droplet size distribution. The polarimetric $v_e$(pol) retrievals behaved similarly, with 85.2% of all retrievals being unaffected, 12.7%

were biased high by one grid point (+0.01) and 1.9% were biased low by one grid point (-0.01). Again, these three populations account for the vast majority of (99.9%) of retrieval outcomes. The percent bias of the ve pol retrieval had a mean of 1.14%, consistent with a 0.01 bias and a standard deviation of 22.6%. The greater tendency toward large biases for the effective variance is likely due to smoothing of polarized reflectance curves after the addition of uncertainty. The large majority of biases in the polarimetric retrieval of $v_e$ are coming from the population of $v_e$ near 0.1-0.15 where the

supernumerary bow peaks are significantly eroded and small shifts in the magnitude of reflectances at angles near these peaks could easily shift the retrieval to the next grid point.

       Overall, the lack of strong systematic biases associated with uncertainty in the case of either retrieval supports an approach of neglecting the measurement uncertainty in further analyses. Of course, this requires acknowledging that biases that are below $\delta r_e$=5%, $\delta v_e$=10%, or $\delta\tau$=7% in either retrieval are probably not as important because they likely are not

detectable due to observational uncertainty.

### 4.3 Retrieval Comparison at High Resolution

       In practice, most observational studies do not have access to the underlying cloud properties with which to compare, so instead different instruments and techniques are often compared to one another. At the native spatial resolution of the LES

(50 m) a direct intercomparison of polarimetric and bispectral retrieval techniques is possible, providing an opportunity to diagnose different sources of bias. The joint histograms of $r_e$ retrievals in Figure 6 compare the two bispectral retrievals, $r_e$(2.13 µm) and $r_e$(3.75 µm), to the polarimetric retrieval, $r_e$(pol), for all LES cases and observation geometries[4]. The regressions for the comparison of both $r_e$(2.13 µm) (Figure 6(a)) and $r_e$(3.75 µm) (panel b) indicate high correlation (R≈0.95) and have relatively small mean biases of less than a micron. A couple of notable features are evident in these regressions. **(1)**

The sign of the mean bias appears to be sensitive to the SWIR/MWIR band selection due to vertical weighting differences,

---

[4] Note that ~1% of pixels in the LES retrieval data correspond to a "failed" bispectral retrieval due to falling outside of the LUT space. These pixels are omitted from the intercomparison. Different reasons for bispectral retrieval failure are discussed in (Cho et al., 2015).

resulting in $r_e(2.13\mu m)<r_e(pol)<r_e(3.75\mu m)$. **(2)** There are numerous statistical outliers with small $r_e(pol)\sim5-9$ µm but broadly distributed $r_e(2.13$ µm) or $r_e(3.75$ µm). One way to understand these features is to constrain the data set to LES columns where both retrieval techniques yield reliable results. As discussed previously, both the bispectral and polarimetric retrievals are sensitive to biases for thin clouds ($\tau<3$) and the polarimetric retrieval is sensitive to biases for broad droplet

size distributions ($v_e>0.15$). Based on these criteria ($\tau>3$ and $v_e\leq0.15$), the constrained joint histograms (Figure 6(c) and (d)) feature much tighter regression relationships (R≈0.99) and reduced mean absolute biases are observed. These filters indicate that the poorly correlated population corresponds to situations in which both retrievals are expected to suffer from significant biases. The retrieval regression can be further improved if the bispectral retrieval is artificially provided with more complete information about the shape of the droplet size distribution. Providing the $v_e(pol)$ retrieval as an a priori assumption for the

bispectral LUT can demonstrate the sensitivity of the bispectral $r_e$ retrievals to the $v_e=0.1$ assumption. This serves as a test of how collocated bispectral and polarimetric retrievals might assist one another. To create these new retrieval results we created a bispectral retrieval LUT for different values of $v_e$ and then selected a different LUT for each pixel depending on the $v_e(pol)$ retrieval. The new coupled $r_e(2.13$ µm) retrievals (Figure 6(e)) are largely unchanged from the $v_e=0.1$ results, although a slight increase in the two biases indicates that $v_e=0.1$ was both an appropriate and sufficient assumption for the

$r_e(2.13$ µm) retrieval. In contrast, the $r_e(3.75$ µm) retrieval (Figure 6(f)) is shown to benefit from this additional a priori information. The coupled 3.75 µm result has an increased correlation, an order of magnitude smaller  bias (0.008 µm), and an absolute bias that is half as large as the original comparison (0.24 µm). The differences between the two SWIR band retrievals can be explained in two ways. Firstly, the vertically weighted DSD of the 2.13 µm SWIR band might result in a broader DSD (i.e., a larger $v_e$) compared to the 3.75 µm SWIR band, due simply to deeper penetration into cloud. This could

provide one explanation for why the $r_e(2.13$ µm) retrieval might improve with the $v_e=0.1$ assumption. Alternatively, the R(2.13 µm) reflectance might simply be less sensitive to the broader DSD shape than the R(3.75 µm) reflectance. Overall, these results demonstrate a feature well known to the cloud remote sensing community; the bispectral retrieval of $r_e$ is not particularly sensitive to $v_e$ (Nakajima and King, 1990a). Indeed, comparison of the coupled bispectral retrieval of $r_e$ to the polarimetric retrieval of $r_e$ confirms that the advantage of retrieving $v_e$ changes the bispectral retrieval of $r_e$ by less than a

micron, so it is appropriate to neglect this level of detail of the DSD for bispectral retrieval purposes. In the context of measurement uncertainty, as discussed in section 4.2, this effect would be below retrieval uncertainty. Overall, this demonstrates that when the two retrievals are compared on equal footing they are very nearly equivalent, with only slight differences leading to $r_e(2.13\mu m)<r_e(3.75\mu m)<r_e(pol)$ as would be expected based on an increasing droplet size vertical profile and vertical weighting.

30       The origin of the broadly distributed high-biased bispectral retrievals in the small droplet size regime ($r_e(pol)\sim5$ µm) stems from the ATEX polluted case. A close examination of this case reveals that there are no bispectral retrievals below 5 µm, despite approximately 3.5% of the scene being characterized by $r_e(VW)<5\mu m$.[6] This feature is a consequence of

---

[6] Additionally, ~1% of the cloudy pixels in this scene exhibit values below 4 µm.

the bispectral LUT state space[7], which spans a $r_e$ range of 5-30 µm. In contrast, the polarimetric retrieval space spans 1-30 µm. The differences between these two LUT spaces is not so much a matter of decision-making, but is more reflective of complexities of the bispectral retrieval for small $r_e$. To demonstrate this point panels (a) and (c) of Figure 7 depict the cloud reflectances from the ATEX polluted case within the respective bispectral LUT. It is obvious that the black isolines for $\tau$ and $r_e$ increasingly overlap with the standard LUT space as $\tau$ decreases. In this region of the state space, there are multiple solutions for a single reflectance pair; one solution is representative of a small $r_e$ (<5 µm, extended LUT), while the other indicates a much larger $r_e$ (≥5 µm, standard LUT). There is also a modest impact on $\tau$, but due to the curvature of the LUT this impact is less severe. The overlapping region between the standard and extended LUT is referred to as the "multiple solution space" and the amount of LUT overlap is determined by both the observation geometry and the selected spectral bands. Depending on the optical thickness, the larger $r_e$ retrieval may be significantly larger, because the extended LUT isolines cross numerous larger $r_e$ isolines in the standard LUT. The associated bispectral retrieval bias, shown in Figure 7(b) and (d), highlights the conclusion that for optically thick clouds the bispectral $r_e$ retrievals exhibits only moderate retrieval biases on the order of ±1 µm. However, for very thin clouds (near cloud edge) the retrieval bias can increase significantly. For some of these thinner clouds the retrievals also fall within the multiple solution space, so it is possible to attribute the very large biases to the presence of ambiguous retrieval results. Furthermore, the multiple solution space also provides an additional explanation for why the removal of optically thin ($\tau$<3) observations significantly improved the bispectral retrieval comparisons.

In contrast to the intercomparison of $r_e$ retrievals, the $\tau$ retrieval intercomparison in Figure 8 reveals very few differences between the bispectral and polarimetric techniques. This is not surprising, because the $\tau$(pol) retrieval is simply an implementation of the bispectral technique with additional constraints on $r_e$ and $v_e$ (as discussed in section 2.2).

### 4.4 Sensitivity to Unresolved Spatial Inhomogeneity

Unresolved spatial inhomogeneity influences the bispectral and polarimetric cloud retrievals in very different ways. Even for 100% cloudy pixels these retrievals can exhibit sensitivity to sub-pixel inhomogeneity. This section focuses on the ATEX cases because they exhibit a broader distribution of $H_\sigma$, allowing us to highlight the impact of spatial inhomogeneity on retrievals. Spatial resolution and sub-pixel inhomogeneity index ($H_\sigma$) are inherently intertwined with one another. This is demonstrated in Figure 9, where the broadening and shifting of the distribution of $H_\sigma$ for increasingly coarsened spatial resolutions is clearly demonstrated using data from both the ATEX clean and polluted cases. In light of this relationship between resolution and inhomogeneity, the inclusion of data from all spatial resolutions together broadens our sampling of different inhomogeneity regimes. To that end, Figure 10 combines all of the coarse spatial resolution data from the two ATEX cases into a single retrieval bias histogram. For the bispectral retrievals in Figure 10 (a,b) we compare to the

---

[7] Note that the MODIS LUT extends its range down to 4 µm, and in situations with multiple solutions the larger retrieval value is selected.

polarimetric retrieval, resulting in histograms that clearly show the two retrievals diverging from one another with increasing sub-pixel inhomogeneity, tends to result in larger biases. In contrast, the polarimetric $r_e$(pol) retrieval in Figure 10 (c) does not appear to have a clear systematic bias. The $v_e$(pol) retrieval in Figure 10 (d) tells a more complicated story, the median value of the bias is clearly close to zero, but there is a tendency toward low biased retrievals with increasing inhomogeneity. It should be noted that the $v_e$(VW) itself increases with increasing $H_\sigma$, which is presumably a consequence of the anticorrelation between $\tau$ and $v_e$(VW). This might explain why for large values of $H_\sigma$, where the $v_e$(VW)>0.15 population is more common, there are more negative biases.

To further emphasize how unresolved inhomogeneity can influence these two retrieval techniques, we will highlight a particularly inhomogeneous pixel from the ATEX clean case at the coarsest resolution (800m). Focusing first on the bispectral retrieval using the 2.13 µm SWIR band, the LUT scatterplot in Figure 11(a) reveals that there is significant variability in the sub-pixel (i.e., 50 m) VNIR reflectances, indicated by a large value of the sub-pixel inhomogeneity index ($H_\sigma$=0.5637). In contrast to the variability of VNIR reflectances, the microphysical properties are largely homogeneous in this 800 m pixel, indicated by the narrow distribution of sub-pixel $r_e$(VW)$_{50\,m}$ (color of the points). The sub-pixel mean of $<r_e$(VW)$>_{50\,m}$ =19.71 µm agrees well with the mean of both sub-pixel retrievals, $<r_e$(2.13 µm)$>_{50\,m}$=18.73 µm and $<r_e$(pol)$>_{50\,m}$=18.92 µm. This combination of optical inhomogeneity and microphysical homogeneity leads to an average reflectance (indicated by the black star) for the 800 m pixel that falls significantly below the $r_e$=20 µm isoline (i.e., the closest isoline to the mean sub-pixel retrievals). Thus, the coarse resolution 800 m reflectance results in an 800 m bispectral retrieval with $r_e$(2.13 µm)$_{800\,m}$=23.62 µm, which is biased high by ~4 µm. This effect is attributable to the well-documented PPH bias induced by the curvature of the bispectral LUT with respect to the optical thickness (Zhang and Platnick, 2011; Zhang et al., 2012; 2016). The PPH bias has a stronger influence on the 2.13 µm retrieval compared to the 3.75 µm retrieval (shown in Figure 11(b)) because the curvature of the LUT-space is more pronounced.

The polarimetric retrieval has a fundamentally different relationship to the unresolved sub-pixel inhomogeneity. This can be demonstrated with the sub-pixel polarized reflectance histogram in Figure 11(b). The reflectances in this figure have been binned by scattering angle to create a distribution of polarized reflectances for the 50 m sub-pixels within the selected 800 m pixel footprint. Within the plot there are also two curves, shifted in amplitude away from the histogram for clarity, that display the mean 800 m multi-angular polarized reflectance and corresponding 800 m retrieved polarized phase function (with appropriate fitting coefficients). Note that, while this histogram gives a sense of the variability of the magnitude and scale of the polarized reflectances, what ultimately matters for the coarse resolution polarimetric retrieval is the relative shape of the 800 m averaged polarized reflectance curve. It is evident from this histogram and these curves that the mean angular position of the supernumerary bow does not shift, indicating that there is no significant difference between $r_e$(pol)$_{800\,m}$, $<r_e$(pol)$_{50\,m}>$, and $<r_e$(VW)$_{50\,m}>$. This agrees with previous studies on the impact of unresolved inhomogeneity on polarimetric $r_e$ retrievals (Shang et al., 2015). In contrast, there is clear variability in the amplitude of sub-pixel polarized reflectances. This variability owes itself to both optical ($\tau$), and microphysical inhomogeneity (i.e., $v_e$(VW)>0.15) within the coarse resolution pixel. For thin clouds ($\tau$<3) the supernumerary bow amplitude is dependent on both $\tau$ and $v_e$ (Alexandrov et

al., 2012b). With $v_e$ fixed the polarized reflectance converges towards an asymptotic maximum for optically thick clouds ($\tau{\geq}3$), a consequence of increasing depolarization due to multiple scattering. Similarly, for a fixed $\tau$, reflectances corresponding to $v_e(\text{VW}){>}0.15$ also produce decreased polarization in the primary and supernumerary bow features, as discussed in section 2. Each of these effects reduces sensitivity to the cloudbow features; and thus unresolved variability in $\tau$

and $v_e$ could influence coarse resolution retrievals. For example, Shang et al. (2015) found that unresolved spatial inhomogeneity of $\tau$ and $v_e$ increased retrieval biases in $v_e(\text{pol})$, while they were not able to discern a trend in retrieval biases in their study. However, in our case study featured in Figure 11(b) we do not see a significant difference between coarse ($v_e(\text{pol})_{800\,\text{m}}$) and fine scale ($<v_e(\text{pol})_{50\,\text{m}}>$) retrievals, but both retrievals are low-biased relative to the mean LES property ($<v_e(\text{VW})_{50\,\text{m}}>$). This result was surprising, because both fine and coarse resolution retrievals were biased similarly. It

appears as though coarse resolution retrievals arrive at the same answer as the fine scale retrievals through different processes. The average of fine scale retrievals (that are systematically biased low) and the retrieval based on the average of fine scale reflectances (which are reduced for reasons discussed above) results in a similar retrieval outcome. Unlike the bispectral retrieval, where retrievals differ from one another at different resolutions, the polarimetric retrieval seems to compare well to itself at both resolutions – even when it might be biased relative to the underlying microphysics of the

physical scene. To examine this this further we performed polarimetric retrievals on subpopulations of the 50 m polarized reflectances within this 800 m pixel that omitted either the $v_e(\text{VW}){>}0.15$ or $\tau{<}3$ from the population. Removing these thin or broad droplet size distributions from the high resolution dataset had little to no impact on either the coarse resolution $r_e(\text{pol})$ or $v_e(\text{pol})$ retrieval From these results and the histogram in Figure 10 (d) it appears that the impact of spatial resolution on $v_e(\text{pol})$ retrievals is largely a consequence of an unresolved anticorrelation between $\tau$ and $v_e$ rather than a feature directly

related to spatial resolution.

### 4.4 Sensitivity to Angular Resolution and Sampling

The polarimetric retrieval requires high-resolution multi-angular data to resolve the supernumerary bow features. To test how angular resolution influences polarimetric retrievals we examined coarse spatial resolution (800 m) $r_e(\text{pol})$

retrievals at different angular resolutions. Each angular resolution (i.e., changing angular step size) was also convoluted with shifting angular sampling (i.e., changing the initial angle). This convolution is necessary in order to account for all possible sets of scattering angle observations associated with each resolution. These coarse resolution retrievals were then compared to the original high angular resolution retrieval. The results of this experiment (**Figure 12**(a)) reveal that coarsening angular resolution does not systematically bias $r_e(\text{pol})$ retrievals, although angular resolutions exceeding 3° do result in a marked

increase in retrieval variability (i.e., a constant mean bias, but increased absolute bias). In contrast, **Figure 12**(b) demonstrates that angular resolutions exceeding 3° lead to both high-biased $v_e(\text{pol})$ and increased retrieval variability. An explanation for the origin of the observed degradation in retrieval accuracy above 3° angular resolution is demonstrated in Figure 13(a). Two different polarized phase functions with $r_e{=}15$ μm and $v_e{=}[0.03, 0.2]$ (solid and dashed-dotted, respectively) are sampled at

an angular resolution of 3.5° (indicated by the gray vertical lines). This resolution is coarser than the spacing between the supernumerary bow features. As a consequence, this particular angular sampling intersects these curves at nearly the same amplitudes. This degeneracy yields a relatively low cost function during the best-fit optimization step of the polarimetric curve fitting retrieval algorithm, making it possible to obtain an inaccurate solution if this results in a cost-function

minimum. The lack of observed differences between these two curves results in a lack of $v_e$ information, which could be exacerbated by observational uncertainty. However, under different angular sampling conditions, e.g., shifting the initial angle by a few degrees to the right, the supernumerary bow peaks of the low $v_e$ curve would be sampled and the similarity between the observations of these two curves would vanish. This example highlights an important feature of multi-angular polarimetry: observations at poor angular resolutions can suffer from increased biases depending on whether or not

important angles are sampled. Generalizing this result requires determining the angular spacing of the supernumerary bow features for other $r_e$. Pursuing this, we find that decreasing cloud droplet size widens and dilates supernumerary bow features, making it easier to resolve supernumerary bow features at coarse angular resolution. The peak-to-peak distance of the supernumerary bow oscillations can be treated as the Nyquist frequency, or in this case Nyquist resolution. In signal analysis, a sampling resolution finer than the Nyquist frequency is required to appropriately resolve features of an oscillatory

signal. The Nyquist angular resolution required for resolving the supernumerary bow oscillations changes with both $r_e$ and $\lambda$ according to the behavior illustrated in Figure 13(b). This analysis indicates that multi-angular observations in a shorter wavelength spectral band would require finer angular resolutions. The Nyquist angular resolution for $\lambda$=0.865 and $r_e$=15 μm is 3°, providing an explanation for the increased variability in $r_e$(pol) and $v_e$(pol) LES retrievals at angular resolutions coarser than the Nyquist limit.

## 5 Summary and Discussion

The analysis in this study, which features comparisons of fundamentally different passive cloud property retrieval techniques, is facilitated by comparisons to LES cloud fields used as input to the retrievals. At the native LES resolution (50 m) there are promising results for both the bispectral and polarimetric retrievals (with 1-D radiative transfer assumptions).

For the bispectral retrieval, the LES comparison shows significant biases for retrievals of very thin clouds, as well as only small differences between the vertically weighted cloud properties in each of the two SWIR/MWIR bands (2.13 and 3.75 μm). Meanwhile, for the polarimetric retrieval, the comparison demonstrates that the $r_e$(pol) retrieval agrees well with the vertically weighted in situ properties of each LES scene. However, the $v_e$(pol) retrieval exhibits persistent low biases due to a lack of retrieval sensitivity to very broad droplet size distributions (i.e., $v_e$(VW)>0.15). The optical thickness retrievals from

both methods are effectively the same, with the caveat that the polarimetric technique performs the $r_e$(pol) retrieval as an a priori constraint on the $\tau$ retrieval space. Regarding $\tau$, both bispectral and polarimetric retrievals were found to have a small systematic high bias on the order of 2-5%.

The uncertainty in observed total and polarized reflectances was found to introduce only weak systematic biases in bispectral or polarimetric $r_e$ retrievals (0.1% or less). Similarly, the bispectral $\tau$ retrievals were also not systematically biased. In contrast, total reflectance uncertainty did produce a slight systematic bias of -2.43% in the polarimetric $\tau(\text{pol})$ retrieval that can be linked to the convexity of the single-band LUT used to perform the retrieval. This sort of bias, could perhaps be accounted for by introducing a Taylor expansion correction similar to the one discussed in Zhang et al. (2016) in the context of unresolved inhomogeneity. Beyond these systematic biases, we found that the induced uncertainties in the bispectral retrievals were $\delta r_e$=5% or $\delta\tau$=7%. The influence of polarimetric retrieval is likely sensitive the polarimetric LUT grid spacing, but here we found uncertainties that were less than the bispectral retrieval of $r_e$, $\delta r_e(\text{pol})$=1 to 4%, and $\delta v_e(\text{pol})$=10 to 20%. In the context of the rest of our comparison studies, the lack of systematic biases and relatively small uncertainties allowed us to discuss retrieval behavior in the absence of uncertainty.

The retrieval intercomparison of polarimetric and bispectral retrievals in this study demonstrates that both techniques yield very similar results, especially when the most reliable populations of cloud properties are selected for each method ($\tau$>3 and $v_e$ around 0.1). While the physical principles and measurement requirements are vastly different, both retrieval techniques seem to be able to capture similar information about $r_e$ and $\tau$. These results agree with high-resolution airborne observations obtained during the PODEX and ORACLES field campaigns, where RSP and AMS microphysical retrievals are compared (Alexandrov et al., 2015; Knobelspiesse et al., 2017). These high spatial resolution field campaign observations indicate that the two retrieval techniques agree well to within the tolerances also found in the present study. The bispectral $r_e$ retrievals are found to be moderately sensitive to $v_e$ in the 3.75 µm band, and less so in the less absorptive and more deeply penetrating 2.13 µm band. Coupling the retrieved $v_e(\text{pol})$ to the bispectral $r_e(3.75$ µm$)$ retrieval led to slight improvements in the $r_e(\text{pol})$ and $r_e(\text{VW})$ comparison. It should be noted that for MODIS cloud products the bias due to the $v_e$=0.1 assumption does not substantially impact the $r_e$ retrieval compared to other sources of bias (i.e., cloud inhomogeneity or 3-D radiative effects). In addition, the MODIS Collection 6 cloud product includes uncertainty estimates associated with the $v_e$ assumption. The intercomparison of the bispectral and polarimetric $\tau$ retrievals indicates that the two produce very similar results. This was to be expected, as the polarimetric technique also uses a bispectral LUT approach to derive $\tau$. When the results from the two methods diverge, the observations tend to be related to the thin cloud regimes.

The presence of a multiple solution space in the bispectral LUTs, where small droplet sizes ($r_e$<5) have the same reflectance as larger droplets, was shown to induce numerous outliers resulting in a significant high bias in the bispectral retrievals for both $r_e$ and (to a lesser extent) $\tau$. This multiple solution space likewise impacts the MODIS operational products, since the bispectral LUTs used in the MODIS collection 6 cloud products include theoretical $r_e$ solutions as low as 4 µm. However, for retrievals with multiple LUT solutions the MODIS product only reports the larger $r_e$ value, leading to a systematic bias if the observed cloud really includes a population of small droplets. As a consequence, for thin clouds with small droplet sizes one can expect the comparison of polarimetric and bispectral retrievals to disagree. This strong high-bias for small $r_e$ retrievals provides a plausible explanation for the large discrepancies observed in the small droplet size regime in the intercomparison of MODIS and POLDER retrievals (Bréon and Doutriaux-Boucher, 2005). Absent a solution to this

issue, future intercomparisons or combined climatological datasets should be limited to retrievals of $r_e$(pol) exceeding 5-7 µm (depending on the respective bispectral LUT multiple solution space properties).

At the coarse spatial resolutions of most satellite instruments, cloud inhomogeneity can significantly impact retrievals. In the context of this study we find that the influence of unresolved spatial inhomogeneity is a dominant source of bias between the polarimetric and bispectral $r_e$ retrievals. In this study we found that even for 100% cloudy pixels (at a coarse 800 m horizontal resolution) the influence of the PPH bias is significant, with the average $r_e$ bias exceeding 1 µm in the most inhomogeneous LES scene (ATEX clean). Based on these results we still expect that the overall systematic bias observed in the MODIS and POLDER intercomparison of moderate droplet size regimes is in large part attributable to the influence of this PPH bias (Bréon and Doutriaux-Boucher, 2005). Recently, great effort has been made to account for the influence of the PPH bias on bispectral MODIS retrievals. The 2-D Taylor expansion technique implemented by Zhang et al. (2016) offers the possibility of quantifying (and potentially correcting for) the impact of PPH bias on bispectral retrievals. This approach requires high spatial resolution measurements in at least one spectral band to obtain the sub-pixel reflectance variability, which is used to determine corrections for the bias of $r_e$ and $\tau$. In addition to PPH bias, 3-D radiative effects are also influenced by spatial resolution. The focus on 1-D radiative transfer in this study leaves questions for future studies regarding the influence of these 3-D radiative effects. Future work will need to identify the relative differences between 3-D radiative effects on total and polarized reflectances and retrievals.

Sufficient angular resolution is one of the more important requirements of the polarimetric retrieval technique. We find that resolving the multi-angular polarized reflectance at a resolution coarser than the Nyquist angular resolution of the supernumerary bow results in greater uncertainty ($r_e$(pol) and $v_e$(pol)) and biased ($v_e$(pol)) polarimetric retrievals. The required angular resolution is dependent both on droplet size and wavelength. Future cloud polarimetric instrumentation should consider these angular resolution requirements. While we have not explicitly tested the so-called "super-pixel" approach implemented for POLDER retrievals, these coarse spatial and angular resolution studies lead to some anticipated biases induced by this technique. We would expect such an approach to further bias $v_e$(pol) retrievals low, due to the lack of sensitivity to unresolved high-$v_e$ populations. In addition, this current study indicates that $r_e$(pol) retrieval variance might increase, but the mean bias might not increase significantly. However, if there is significant correlation between the unresolved $r_e$ and $v_e$ populations within an observation footprint, the mean $r_e$ bias would be expected to suffer.

Ultimately, the utitlity of any optical property dataset depends on the science questions for which the dataset will be used. These questions may focus on the determination of domain-averaged water mass, radiative flux calculations, or microphysical process studies on a range of scales. The appropriate retrieval may differ for each of these science questions and as a consequence the comparison of the bispectral and polarimetric retrievals discussed here ought to be viewed through the lens of a particular application.

**Acknowledgements**

The hardware used in the computational studies is part of the UMBC High Performance Computing Facility (HPCF). The facility is supported by the U.S. National Science Foundation through the MRI program (grant nos. CNS-0821258 and CNS-1228778) and the SCREMS program (grant no. DMS-0821311), with additional substantial support from the University of Maryland, Baltimore County (UMBC). See www.umbc.edu/hpcf for more information on HPCF and the projects using its resources.

**Appendix**

We often treat the droplet size distribution observed by in-situ instruments (on the order of meters) as relatable to the inferred size distribution properties obtained by remote sensing retrievals (on the order of kilometers). This mathematical analysis addresses how resolution and scale influence the inferred cloud microphysical distribution. The modified gamma-distribution not only suits observations of in-situ cloud droplet size distributions, but it also exhibits several useful mathematical relationships:

$$
\begin{aligned}
\langle r^2 \rangle &= r_e^2 (v_e - 1)(2v_e - 1) \\
\langle r^3 \rangle &= r_e^3 (v_e - 1)(2v_e - 1) \\
\langle r^4 \rangle &= r_e^4 (v_e - 1)(2v_e - 1)(v_e + 1)
\end{aligned}
\tag{5}
$$

From a retrieval perspective all droplet size distributions are treated as gamma-distributed. There is a potential disconnect here, from the perspective of scale analysis, when retrievals at a 50 m spatial resolution (our LES resolution) and retrievals at 1 km (MODIS retrieval resolution), or even 150 km (POLDER retrieval resolution) each are being treated as gamma-distributed. However, not all droplet microphysics information is created equal; the droplet size distributions at higher resolution (subscript, $i$) influence the low-resolution (subscript, $lr$) droplet size distributions. With high-resolution information the different moments of the coarser resolution droplet size distribution should be able to be constructed from the high-resolution microphysics. For a distribution made up of the summation of gamma size distributions the moments of the low-resolution distribution can be expressed by the following relationship, because summation and integration are each linear operators:

$$
\begin{aligned}
\langle r^n \rangle_{lr} &= \int_r r^n \left[ \sum_i^k N_i \left( r, r_{e,i}, v_{e,i} \right) \right] dr \\
\langle r^n \rangle_{lr} &= \sum_i^k \left[ \int_r r^n N_i \left( r, r_{e,i}, v_{e,i} \right) dr \right] = \sum_i^k \left[ \langle r^n \rangle_i \right]
\end{aligned}
\tag{6}
$$

With this mathematical rule in mind, the values of $r_e$ and $v_e$ for the low-resolution droplet size distribution can be obtained by substitution into eq. (2) and eq. (3):

$$
r_e' \equiv \frac{\langle r^3 \rangle_{lr}}{\langle r^2 \rangle_{lr}} = \frac{\sum_i^k \langle r^3 \rangle_i}{\sum_i^k \langle r^2 \rangle_i} ,
\tag{7}
$$

$$v'_e \equiv \frac{\langle r^4 \rangle_{lr} \langle r^2 \rangle_{lr}}{\left(\langle r^3 \rangle_{lr}\right)^2} - 1 = \frac{\sum_i^k \langle r^4 \rangle_i \langle r^2 \rangle_i}{\left(\sum_i^k \langle r^3 \rangle_i\right)^{2x}} - 1 \ . \tag{8}$$

Henceforth, we will refer to the $r'_e$ and $v'_e$ relationships in eq. (7) and eq. (8) as microphysical "aggregation rules." It should be noted that these rules fundamentally treat the DSD as gamma-distributed at all scales.

The microphysical aggregation rules allow for the explanation of some features of the coarse polarimetric retrieval experiments displayed in Shang et al., (2015). Referring to the inhomogeneous polarimetric retrieval experiments in table 2 and figure 4 of their paper, we reproduced their results and calculated the corresponding $r_e$' and $v_e$' in our **Table 2**, which contains the same retrieval examples and corresponding $r_e$' and $v_e$' results for the cases examined in their study. There is a clear difference between the mean $r_e$ or $v_e$ and the polarimetric retrieval results. Using the microphysical aggregation rules defined above, we derived that the appropriate distribution properties, $r_e'$ and $v_e'$, are generally in closer agreement with the polarimetric retrievals of $r_e$(pol). These results offer a possible explanation as to why the polarimetric retrieval does not agree with the average of the sub-scale microphysics in Shang et al.'s study. A couple of things should be noted here: **1)** When there is little variability in the unresolved $r_e$ (e.g., $r_e$=[15,20] $\mu$m) the mean, retrieval, and the estimated mixture are generally all in agreement (e.g., $<r_e>$=17.5, $r_e$(pol)=18, and $r_e$'=18.2 $\mu$m). **2)** When large variability in the unresolved $r_e$ (e.g., $r_e$=[5,20]) is present, both the retrieved and estimated mixture strongly favor the larger droplet effective radius (e.g., $r_e$(pol)=19 and $r_e$'=19.12 $\mu$m). **3)** Large variability in unresolved $r_e$ sometimes results in large differences between $v_e$(pol) and $v_e$'. The last two points are likely a consequence of the resulting coarse resolution (multi-modal) distribution differing significantly from the gamma-distribution assumption stated previously.

Applying this analysis to the aggregation of LES scene microphysics will allow for the determination of how accurate a spatial mean aggregation reflects the true coarse resolution microphysical parameters. We first assumed that all of the highest resolution vertically weighted size-distributions can be assumed to be appropriately characterized by a gamma distribution with $r_e$=$r_e$(VW) and $v_e$=$v_e$(VW). We then aggregated these LES microphysical properties at the 50 m native resolution to increasingly coarser resolutions (100, 200, 400, and 800 m), using both the mean and the aggregation rules. We found that the differences between the two techniques are negligible ($\Delta r_e$~0.01 $\mu$m and $\Delta v_e$~0.001) and do not significantly vary with final resolution. Apparently, the importance of the aggregation rules in the LES are far less important than what we had found in the multiple-moment cases tested in Shang, et al. (2015). One clear difference between the these multiple moment cases and the LES was that the toy models are reductive bimodal distributions, exhibiting very large sub-scale microphysical inhomogeneity in $r_e$. This non-physical variability is something that is not commonly observed in the LES or in observational studies. To address this, we performed a theoretical examination of how important the aggregation rules are for calculating the bias between simple average aggregation and mathematical rule aggregation. In this experiment we

established various distributions of unresolved DSD's with varying $r_e$ and $v_e$ populations. These joint distributions of $r_e$ and $v_e$ were used to test how the variance (i.e., the unresolved variability) would influence the average and mathematical rule aggregated results. This test confirmed, that large differences between the simple average and mathematical aggregation rules requires spatial inhomogeneity of microphysics that much larger than those observed in the LES or typical observational studies. Based on these results we recommend that future studies focusing on the effect of unresolved microphysical inhomogeneity on polarized retrievals should consider more realistic inhomogeneity conditions on both $r_e$ and $v_e$.

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

**Table 1: Mean values ($\mu$) and standard deviations ($\sigma$, in parenthesis) of various optical ($\tau$ and $H_\sigma$) and microphysical properties ($r_e$(VW) and $v_e$(VW)) of the LES scenes examined in this study. Note that vertically weighted properties are listed for the polarimetric vertical weighting function and $H_\sigma$ for nadir viewing 0.865 μm reflectance. Cloudy pixels is defined using a threshold of $\tau_{\mathrm{LES}}$>0.1.**

| LES Case | CCN Concentration [#/cm$^3$] | Scene Cloud Fraction | $\tau$ [unitless] | $r_e$(VW) [$\mu$m] | $v_e$(VW) [unitless] | $H_\sigma$(800 m) [unitless] |
|---|---|---|---|---|---|---|
| DYCOMS-II | 60 | 0.998 | 17.95 (6.22) | 15.52 (1.00) | 0.071 (0.11) | 0.13 (0.10) |
| ATEX Clean | 40 | 0.941 | 7.90 (8.02) | 16.93 (2.62) | 0.16 (0.12) | 0.42 (0.17) |
| ATEX Poll. | 600 | 0.985 | 17.48 (14.71) | 7.29 (0.91) | 0.13 (0.068) | 0.24 (0.13) |

Table 2: The influence of unresolved microphysical inhomogeneity on polarimetric retrievals is explored in Shang et al. (2015). There results are replicated here and compared to the arithmetic mean $r_e$ ($<r_e>$), and the mathematical aggregation results ($r_e$' and $v_e$') defined in eq. (7) and eq. (8).

| Sub-scale Size Distribution Mixture | | Arithmetic Mean | Polarimetric Retrieval | | Aggregation Rules | |
|---|---|---|---|---|---|---|
| $r_e$ | $v_e$ | $<r_e>$ | $r_e$(pol) | $v_e$(pol) | $r_e$' | $v_e$' |
| [5, 10] | [0.01, 0.01] | 7.5 | 8.0 | 0.10 | 9.00 | 0.060 |
| [5, 15] | [0.01, 0.01] | 10.0 | 14.5 | 0.01 | 14.00 | 0.056 |
| [5, 20] | [0.01, 0.01] | 12.5 | 19.0 | 0.01 | 19.12 | 0.044 |
| [10, 15] | [0.01, 0.01] | 12.5 | 13.0 | 0.05 | 13.46 | 0.040 |
| [10, 20] | [0.01, 0.01] | 15.0 | 16.5 | 0.10 | 18.00 | 0.060 |
| [15, 20] | [0.01, 0.01] | 17.5 | 18.0 | 0.01 | 18.20 | 0.028 |
| [5, 10, 15] | [0.01, 0.01, 0.01] | 10.0 | 12.0 | 0.10 | 12.85 | 0.069 |
| [5, 10, 20] | [0.01, 0.01, 0.01] | 11.7 | 14.0 | 0.10 | 17.38 | 0.087 |
| [5, 15, 20] | [0.01, 0.01, 0.01] | 13.3 | 17.5 | 0.02 | 17.69 | 0.049 |
| [10, 15, 20] | [0.01, 0.01, 0.01] | 15.0 | 16.0 | 0.10 | 17.07 | 0.055 |

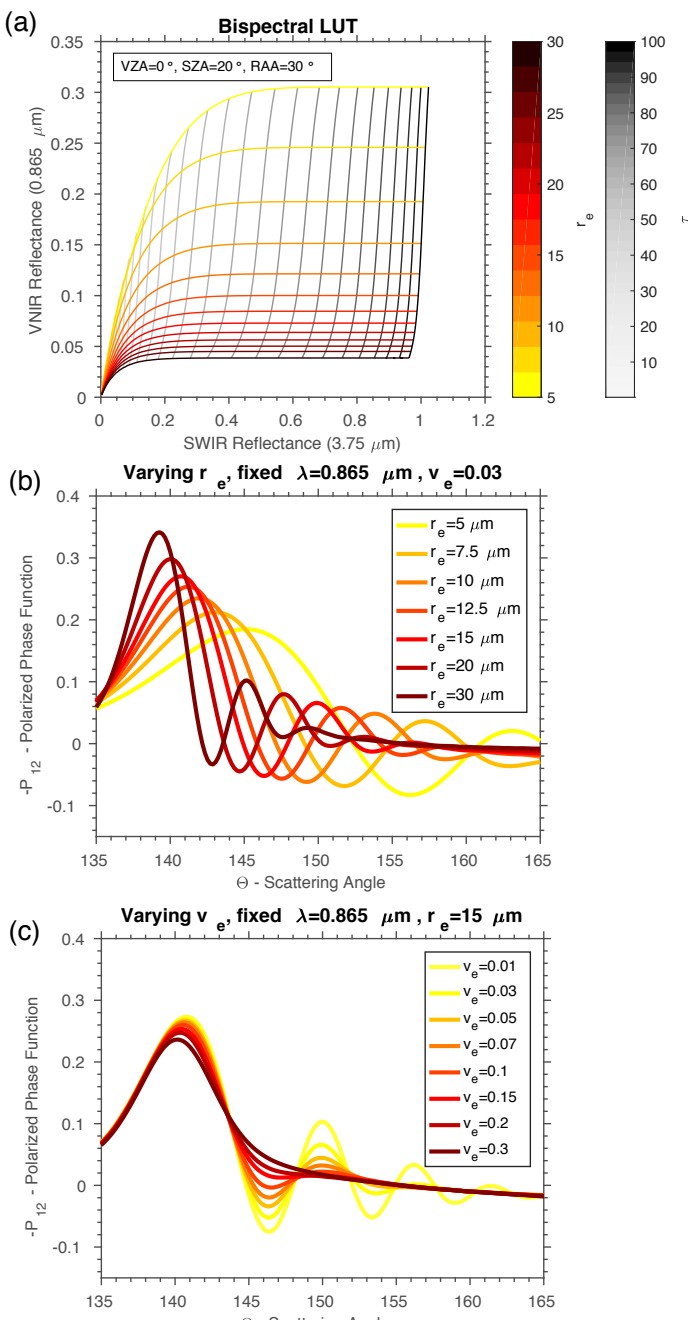

**Figure 1: Demonstrations of the microphysical sensitivity of the bispectral and the polarimetric techniques. Panel (a) features the bispectral LUT exhibiting sensitivity to $r_e$ (colored iso-lines), due to absorption in the SWIR/ reflectance. The VNIR reflectances provide sensitivity to optical thickness (gray iso-lines). Panels (b) and (c) demonstrate the sensitivity of polarimetric technique to $r_e$ and $v_e$ respectively. The supernumerary bow peaks of the polarized phase function ($-P_{12}$) shift and become narrower with increasing droplet size ($r_e$), whereas the peaks erode in magnitude for broadened droplet size distributions ($v_e$).**

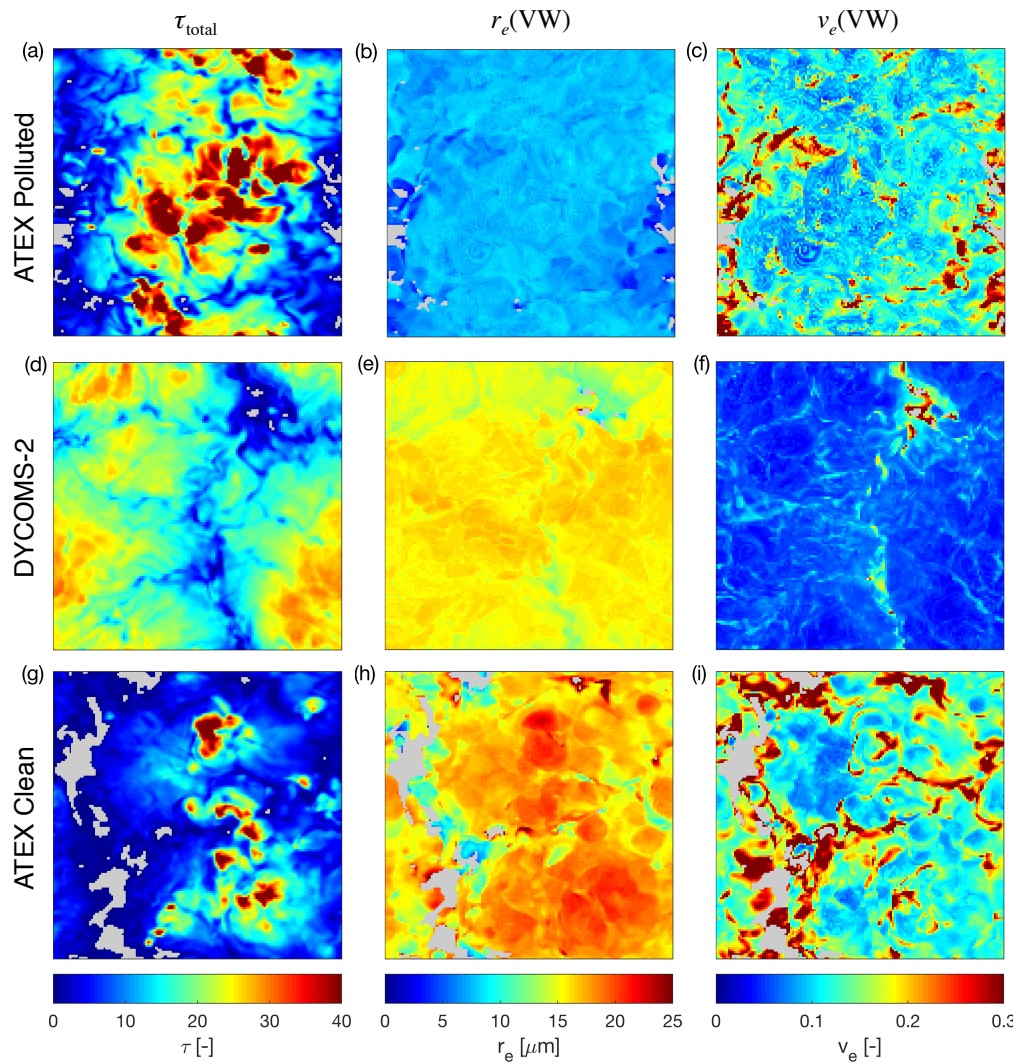

**Figure 2:** The optical and microphysical properties ($\tau$, $r_e$, and $v_e$) of the LES cases examined in this study. The panels are arranged such that each LES case appears row-wise and the different properties are appear column-wise. Note that the vertically weighting functions used for the displayed $r_e$(VW) and $v_e$(VW) correspond to single-scattering assumptions. Cloud-free masking in each of the images appears in gray. Refer to sections 2 and 3 for discussion and definition of each of these properties. Axes labels have been removed to enlarge each map, but the spatial dimensions of each scene are roughly 7 x 7 km (refer to section 3 for the specific resolutions of each LES case.)

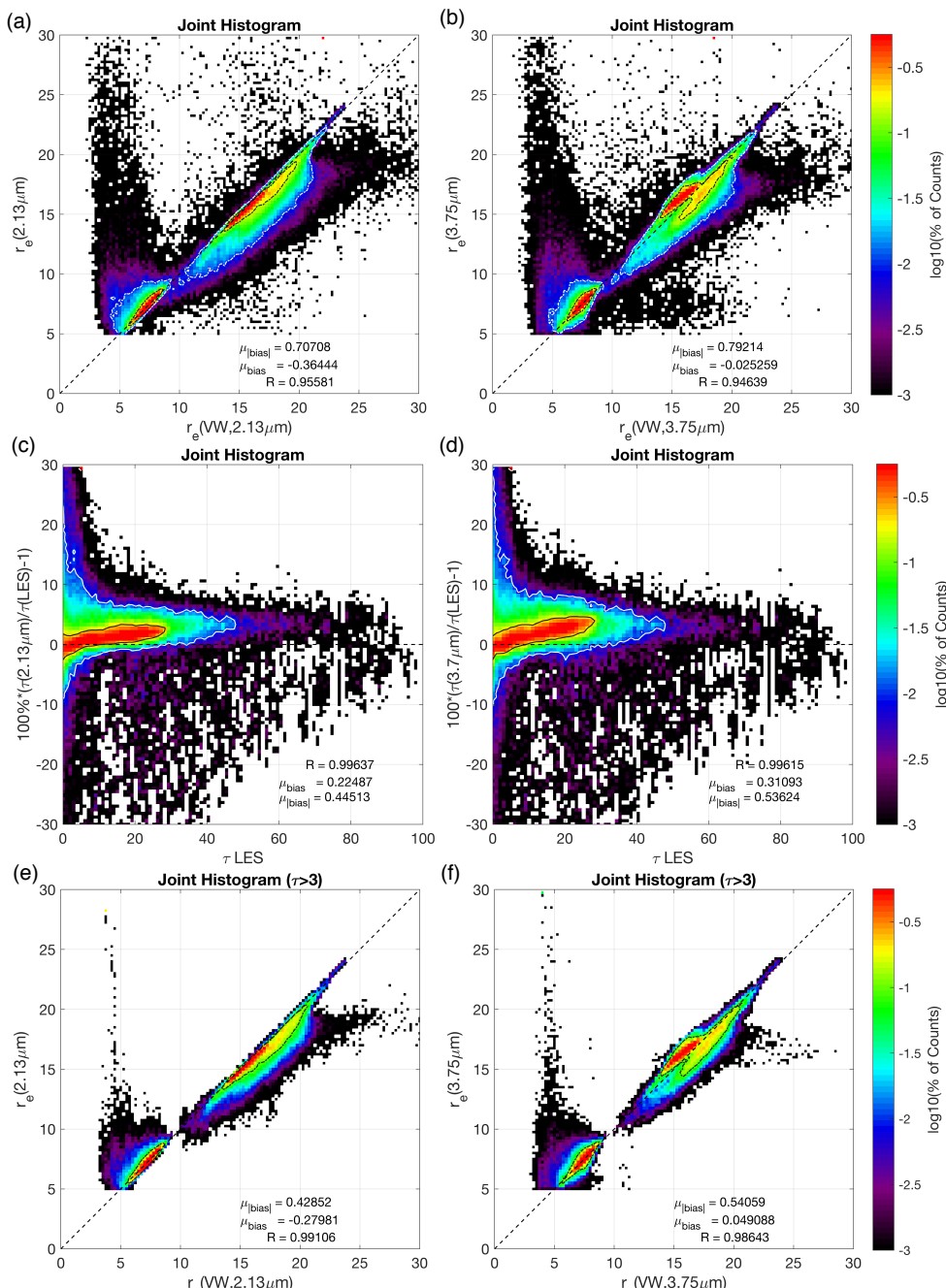

**Figure 3: Joint histogram regressions of $r_e$ and $\tau$ in all LES cases comparing the bispectral retrievals to the LES cloud microphysical properties. Panels (a) and (b) are regressions of the bispectral $r_e(2.13\ \mu m)$ and $r_e(3.75\ \mu m)$ retrievals against the physical analogue $r_e$(VW). Panels (c) and (d) are regressions of the bispectral $\tau(2.13\ \mu m)$ and $\tau(3.75\ \mu m)$ retrievals against the physical $\tau$(LES). Panels (e) and (f) display the regression of the bispectral $r_e(2.13\ \mu m)$ and $r_e(3.75\ \mu m)$ retrievals for only optically thick pixels ($\tau > 3$). Note that in each panel the correlation is quantified with a linear correlation coefficient (R) and the black and white contours encompass 66% and 95% of the population, respectively.**

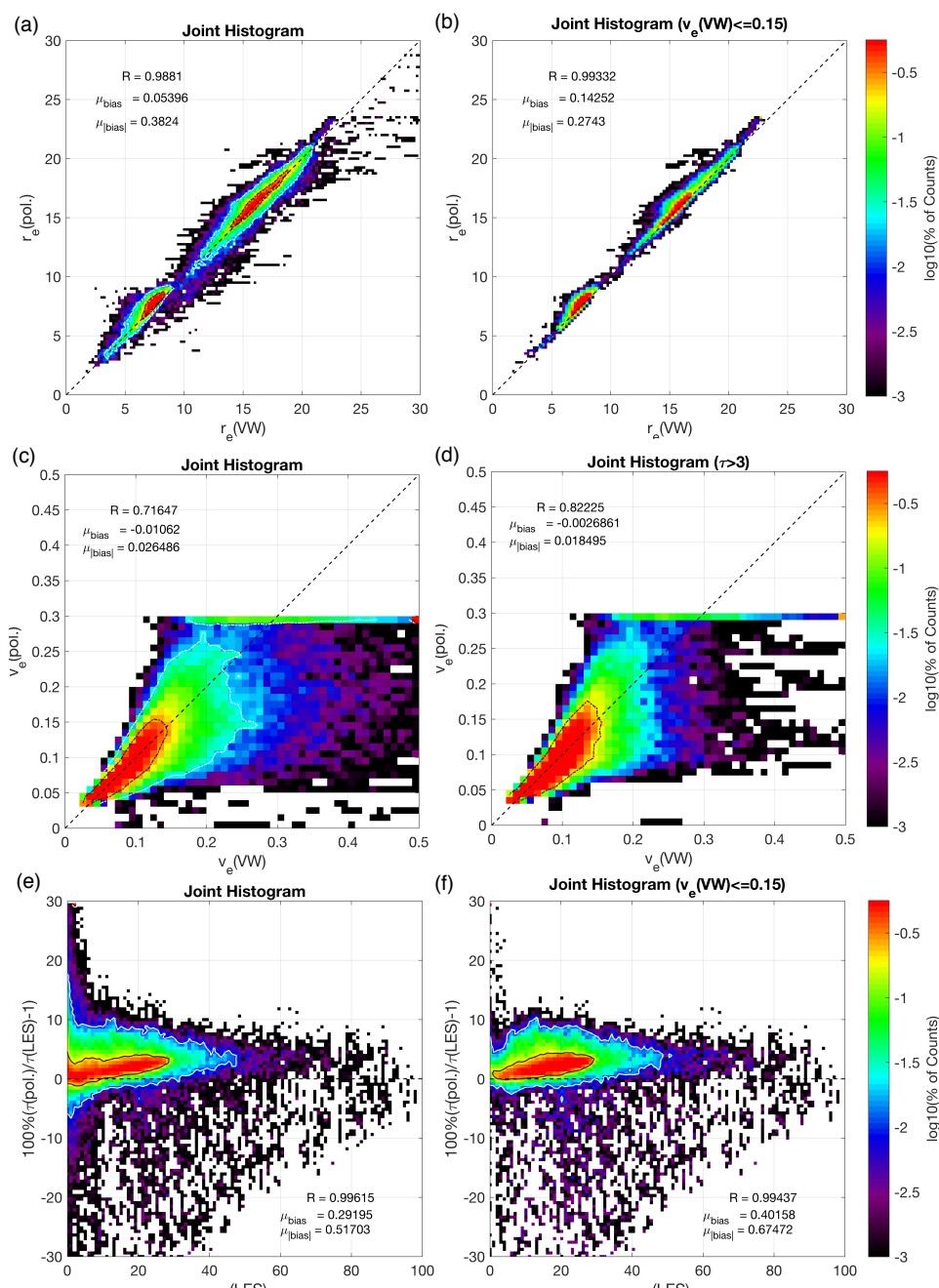

**Figure 4: Joint histogram regressions of $r_e$, $v_e$, and $\tau$ in all LES cases comparing the polarimetric retrievals to the LES cloud microphysical properties. Panel (a) depicts the regression of the polarimetric $r_e$(pol) retrieval against the physical analogue $r_e$(VW), while panel (b) is sub-selection of the same regression for low $v_e$. Panel (c) depicts the regression of the polarimetric $v_e$(pol) retrieval against the physical analogue $v_e$(VW), while panel (d) is a sub-selection of the same regression for thick clouds ($\tau$>3). Panel (e) depicts the regression of the polarimetric $\tau$(pol) retrieval against the physical analogue $\tau$(LES), while panel (f) is sub-selection of the same regression for low $v_e$. Note that in each panel the correlation is quantified with a linear correlation coefficient (R) and the black and white contours encompass 66% and 95% of the population, respectively.**

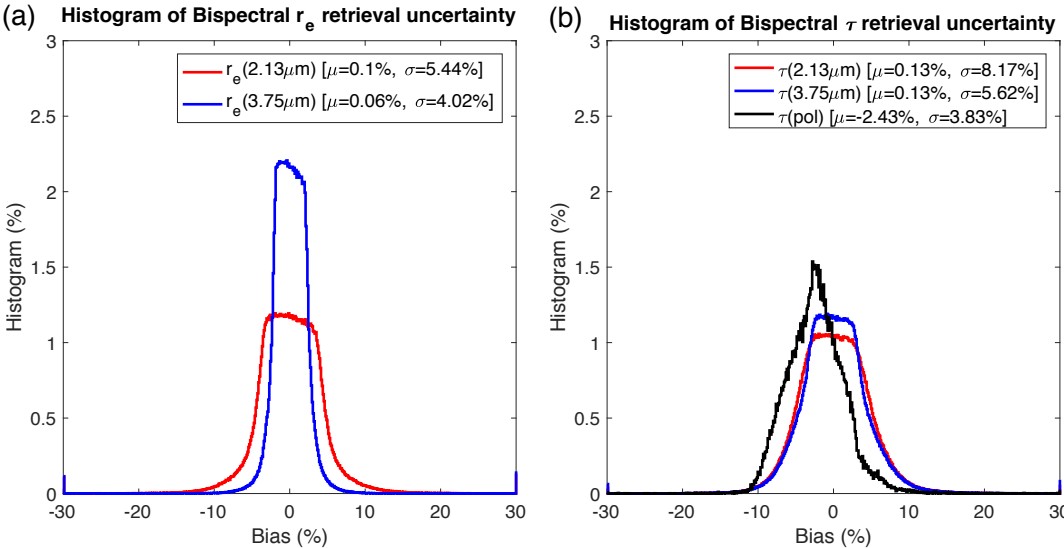

**Figure 5:** Histograms of the percent retrieval bias of retrievals based on perturbed reflectances stated relative to unperturbed retrievals. Panel (a) displays retrieval biases for the bispectral $r_e$ retrieval. Panel (b) displays retrieval biases for the bispectral and polarimetric $\tau$ retrievals. Refer to the text for more information about the polarimetric $r_e$ and $v_e$ retrieval biases.

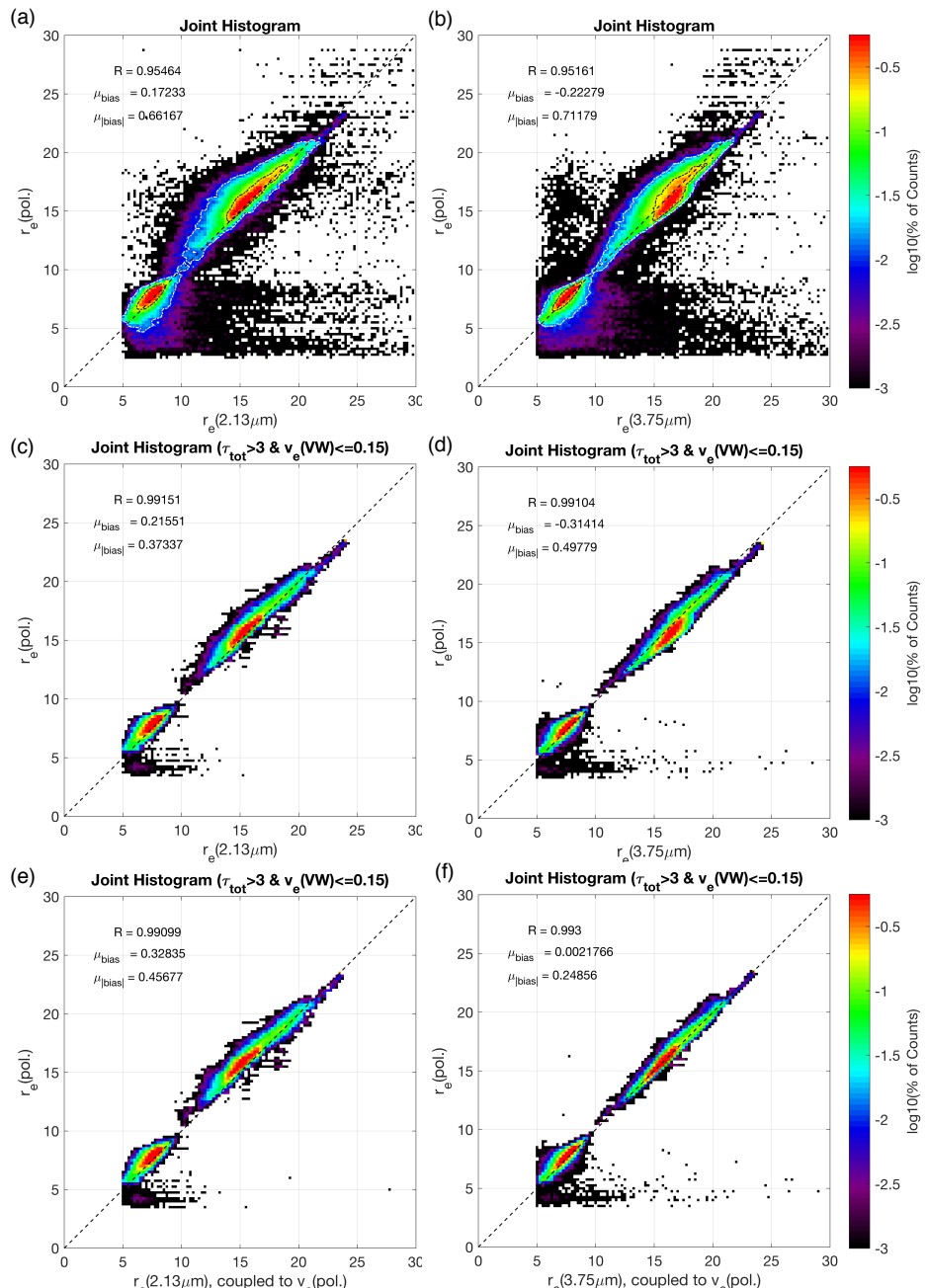

**Figure 6: Joint histogram regressions of $r_e$ retrievals for all LES cases comparing the bispectral and polarimetric techniques.** Panels (a) and (b) display the unfiltered regressions of $r_e$(pol) at 0.865 $\mu$m wavelength against the $r_e$(2.13 $\mu$m) and $r_e$(3.75 $\mu$m) bispectral retrievals. After introducing filters to these regressions to remove thin clouds ($\tau$<3) and broad droplet size distributions ($v_e$>0.15) panels (c) and (d) the retrieval intercomparison improves. Panels (e) and (f) each replicate the results from the previous selection criteria but additionally provide bispectral retrieval in this regression with $v_e$(pol) as an a priori for each retrieval. In each panel the quality of the correlation is quantified and a black iso-contour is drawn surrounding 75% data in the histogram. Note that in each panel the correlation is quantified with a linear correlation coefficient (R) and the black and white contours encompass 66% and 95% of the population, respectively.

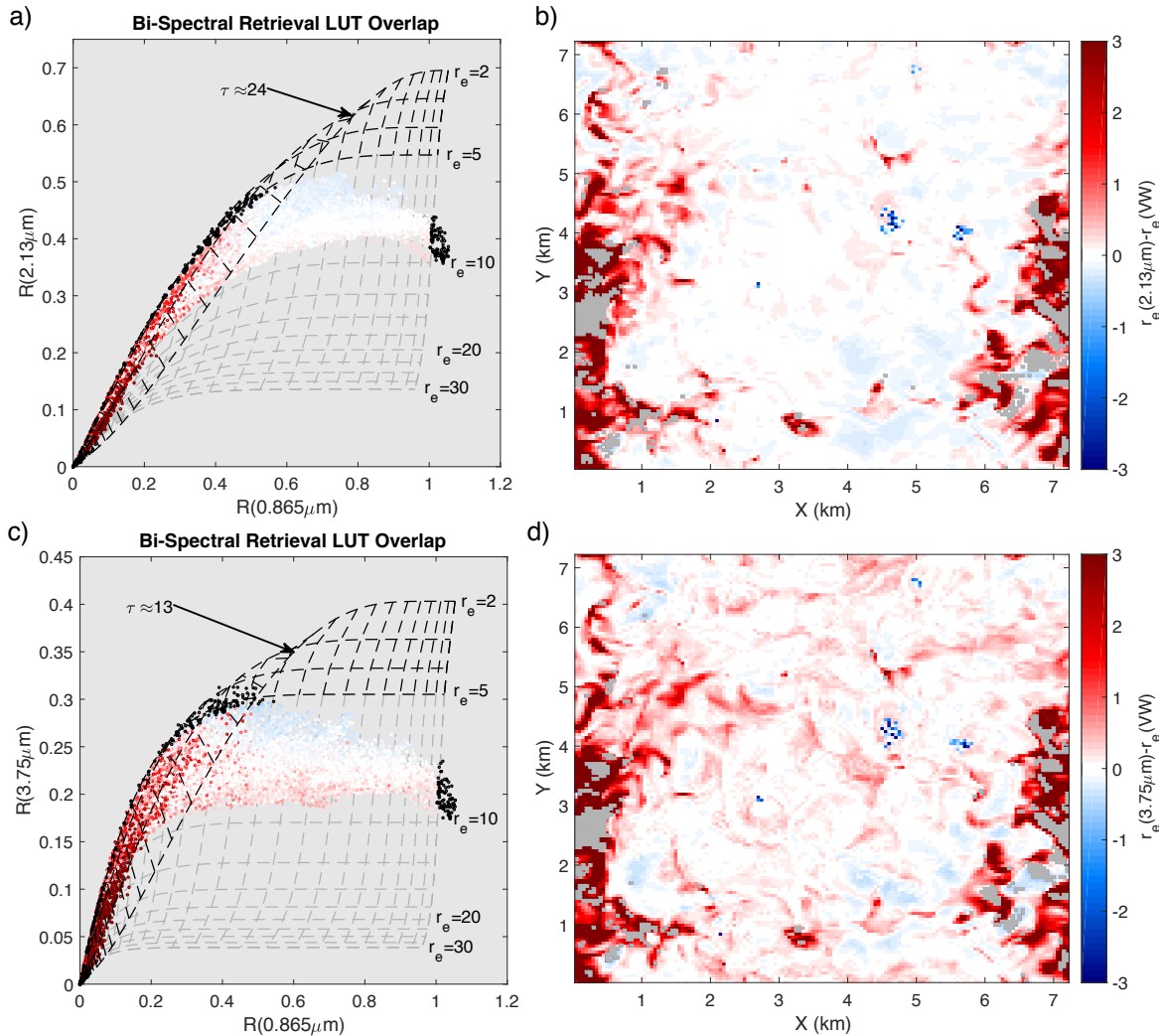

**Figure 7:** Panel (a) and (c) depict the standard bispectral LUT (light gray dashed lines) for both SWIR bands with the scattered reflectance points for the ATEX polluted LES case plotted overtop. The scatterplot is colored by the bias between the bispectral retrieval and the physical reference ($r_e$(bispectral)−$r_e$(VW)). Spatial maps of this bias are shown for context in panels (b) and (d). Note that some points are colored in black to indicate retrieval failure due to falling outside the LUT space. In addition to the standard LUT, an extended LUT including droplet sizes from 2-4$\mu$m is included (black dashed lines), revealing an overlapping region of the two LUT for smaller $\tau$ referred to as the "multiple solution space".

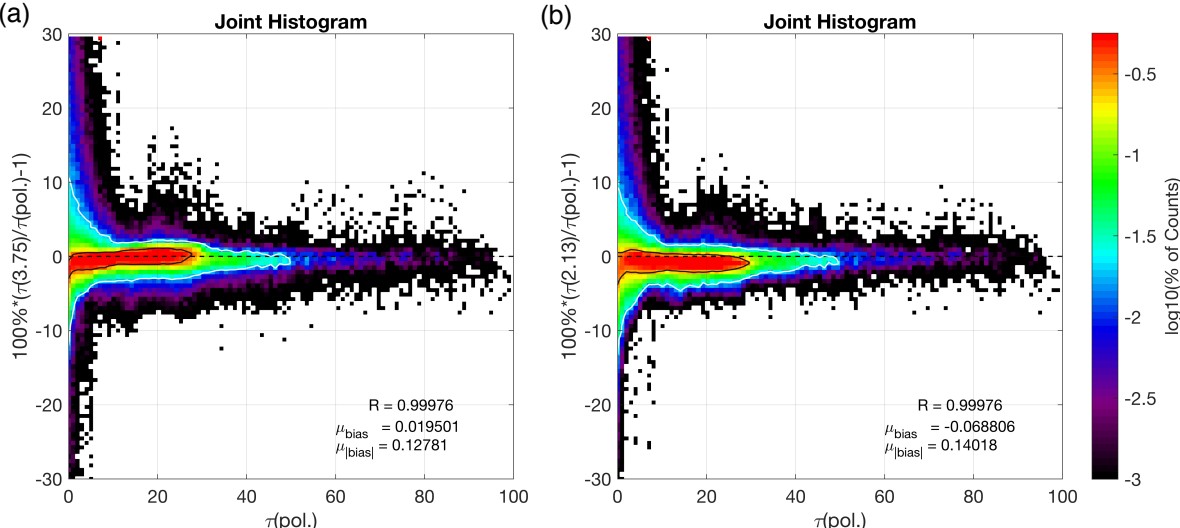

**Figure 8: Joint histogram regressions of $\tau$ retrievals for all LES cases comparing the bispectral and polarimetric techniques. Panel (a) and (b) display the $\tau$(2.13 μm) and $\tau$(3.75 μm) retrievals respectively. In each panel the quality of the correlation is quantified and the black and white population density iso-contours are drawn surrounding 66% and 95% of the data respectively.**

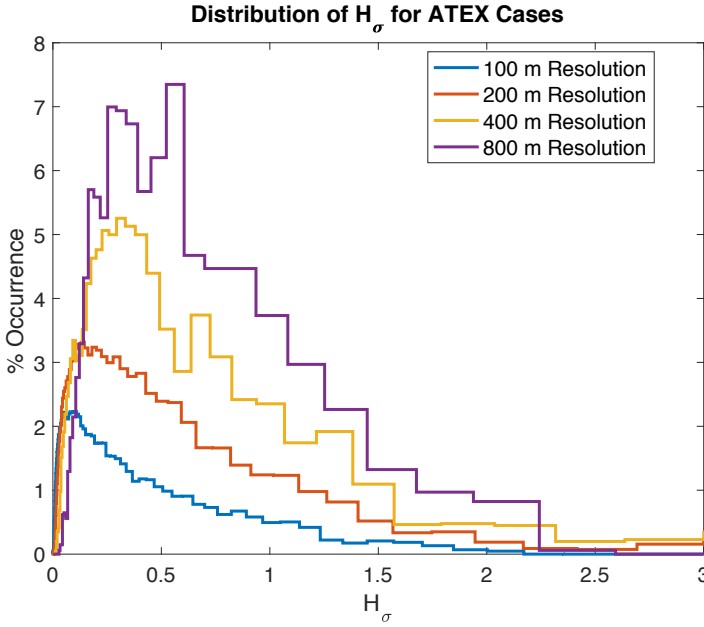

**Figure 9:** Probability distributions of $H_\sigma$ for the combined ATEX polluted and clean datasets at all coarsened spatial resolution (100, 200, 300, 400, 800 m).

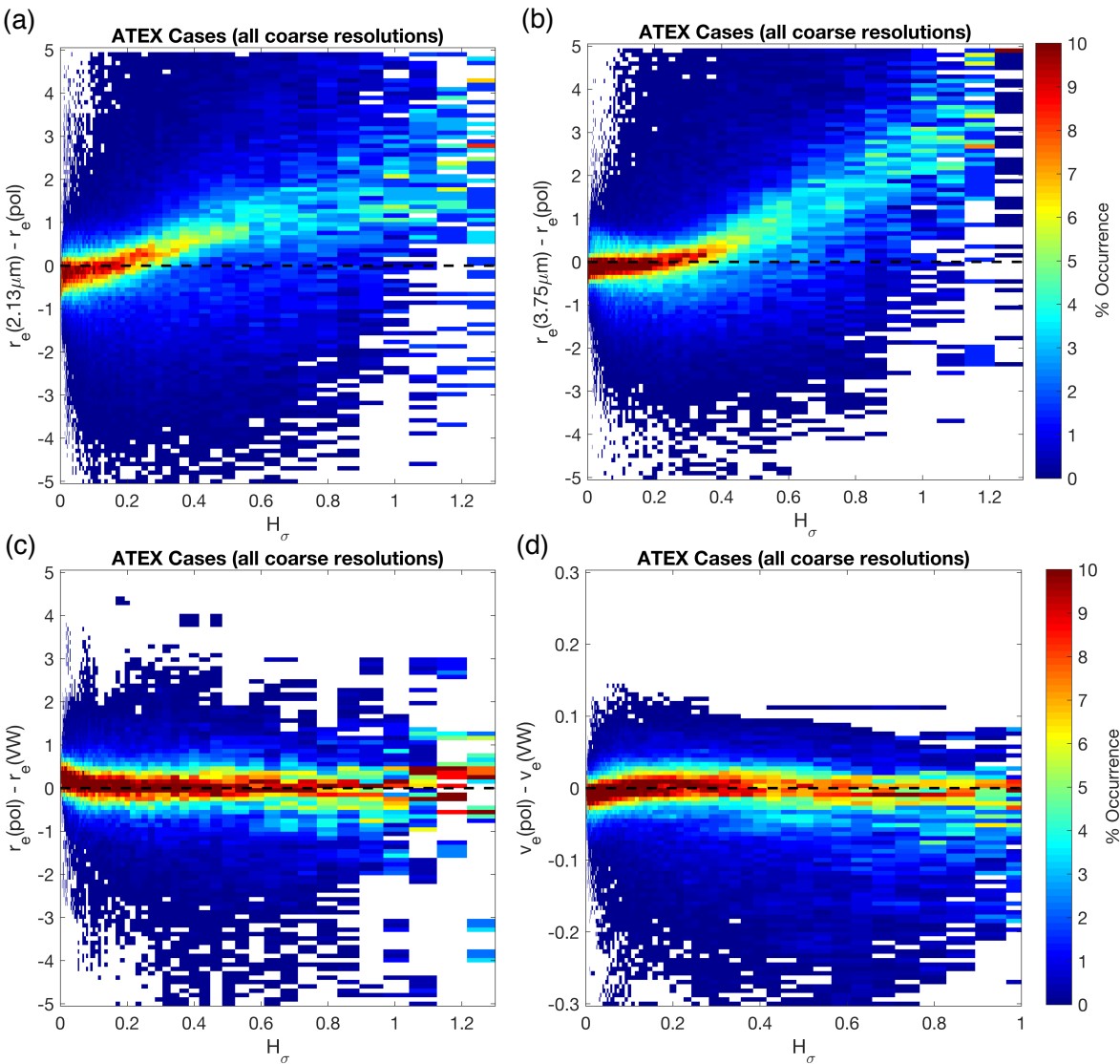

**Figure 10: Joint histograms of retrieval biases (relative to each relevant vertically weighted LES property) with respect to $H_\sigma$ for the combined ATEX clean and polluted datasets at all observation geometries and including all coarsened spatial resolutions (100, 200, 300, 400, 800 m). The color bar indicates percent occurrence. Panels (a) and (b) depict the difference between the two bispectral $r_e$ retrievals and the polarimetric retrieval, while panels (c) and (d) depict biases for the polarimetric $r_e$ and $v_e$ retrieval against $r_e$(VW).**

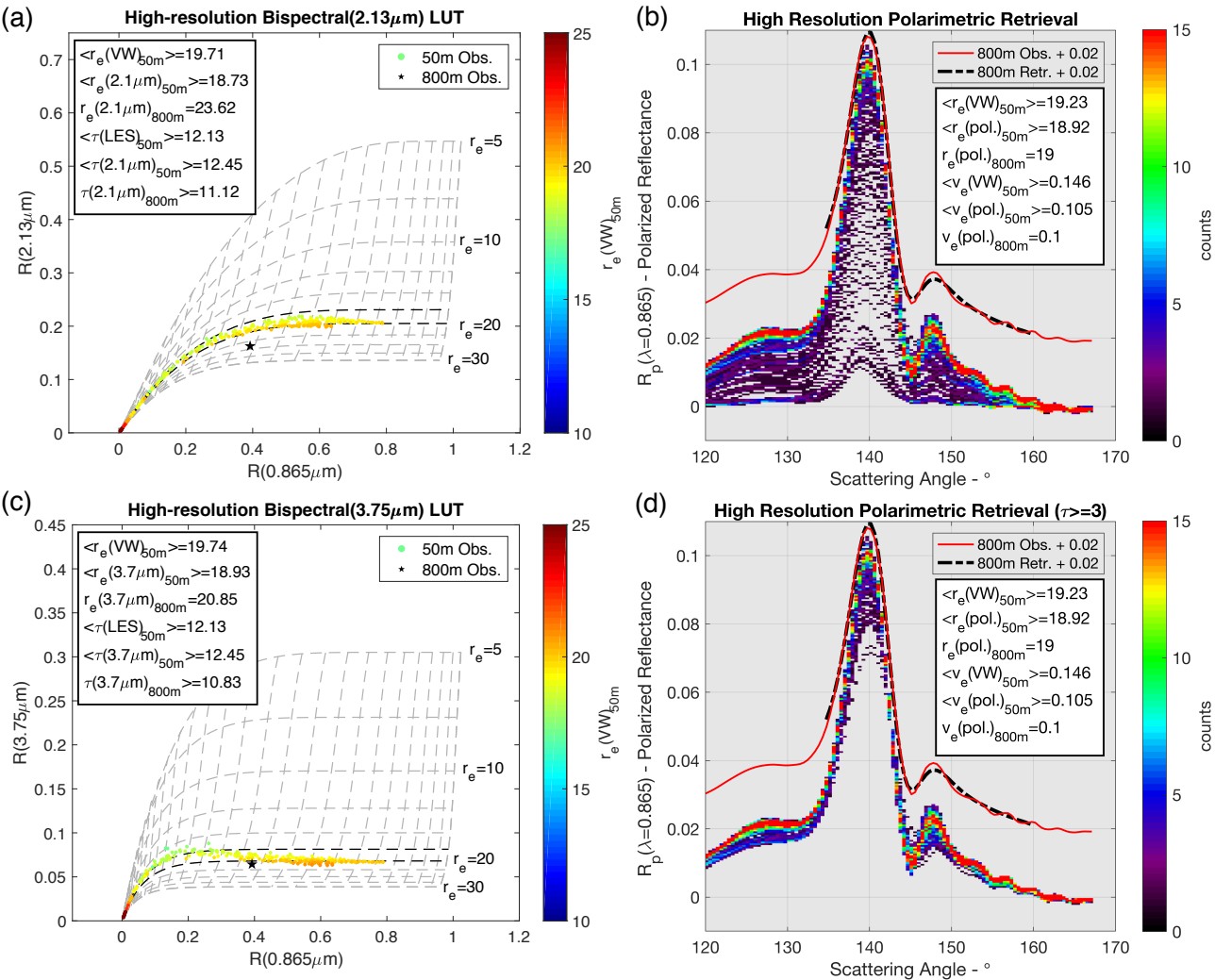

**Figure 11: Panels (a) and (c) depict the bispectral LUT's and 50 m reflectances for the 2.13 and 3.75 $\mu$m bispectral retrievals respectively for a particularly inhomogeneous 800 m pixel. The scattered points correspond to 50 m reflectances with color corresponding to $r_e$(VW), while the black star corresponds to the 800 m reflectance pair (the average of the 50 m data). The polarimetric reflectance distribution histograms in panels (b) and (d) address how the high-resolution (50 m) reflectance distribution influences the polarimetric retrieval at coarse resolution (800m). The two curves (plotted with a 0.02 reflectance shift for clarity) are the 800 m observed reflectance (black dashed curve) and the 800 m retrieval (red solid curve). All of these figures include statistics on the high-resolution averages of physical properties and retrievals along with their coarse resolution counterparts for comparison.**

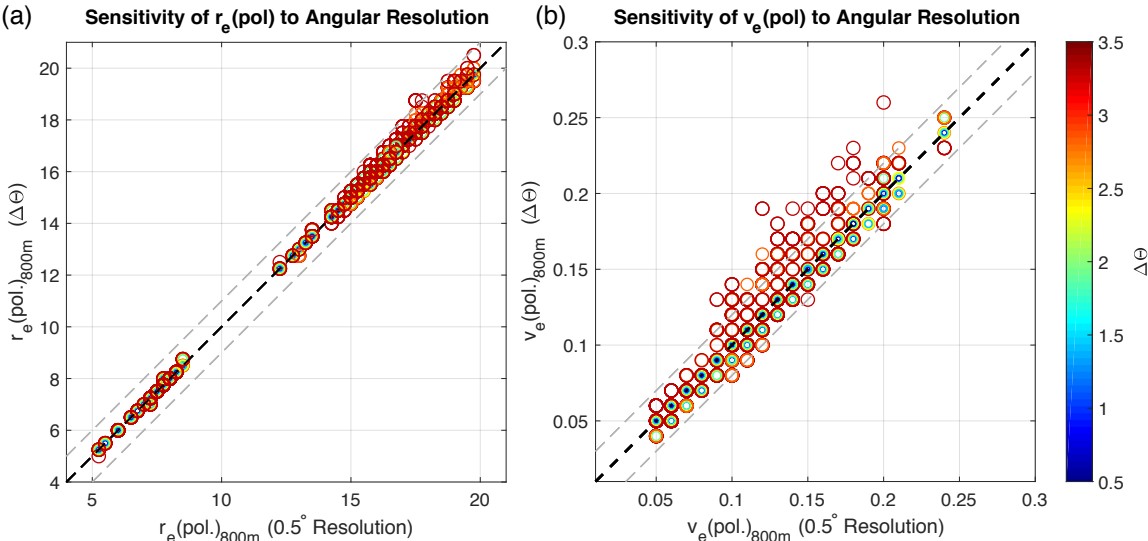

**Figure 12: Angular resolution sensitivity experiments examining polarimetric retrievals of $r_e$ (panel a) and $v_e$ (panel b) for all LES scenes at the 800 m spatial resolution. The color and size of scattered points denote the angular resolution of each retrieval. The gray dashed lines denote the ±1 step in the LUT space of the polarimetric retrieval.**

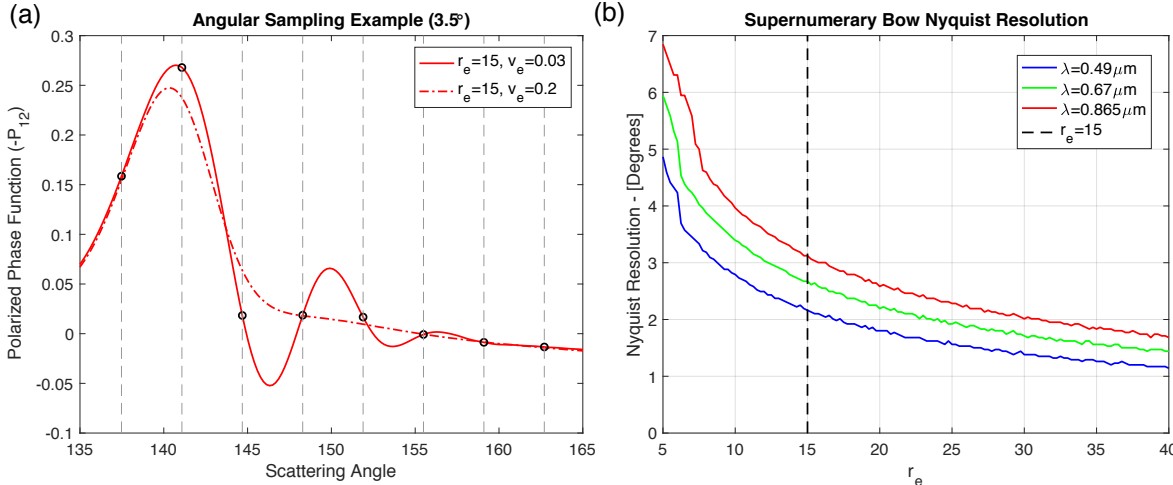

**Figure 13: Panel (a) features the polarized phase functions $r_e$=15 (red) at $v_e$=0.03 (solid) and $v_e$=0.2 (dashed). Grey dashed lines and circles indicate a 3.4° observation sampling of the phase functions. The Nyquist resolution is obtained by measuring the peak-to-peak distance of the supernumerary bow oscillations and dividing that distance in half. The Nyquist resolution changes as a function of $r_e$ and $\lambda$ as shown in panel b, where the gray vertical line highlights the Nyquist resolutions required for the $r_e$=15 case.**

