# Peer review of "Comparisons of bispectral and polarimetric retrievals of marine boundary layer cloud microphysics: Case studies using a LESsatellite retrieval simulator"

_Atmospheric Measurement Techniques, 2017_

## Referee Comment (RC1) · Anonymous Referee #1 · 4 Oct 2017

**1   General remarks**

The manuscripts provides a detailed analyses of two well established cloud retrieval methods using passive satellite measurements of solar radiation. Bispectral and polarimetric retrieval of cloud optical thickness, droplet effective radius, and effective variance are compared for cases of liquid maritime clouds. The study is not based on measurements of real clouds. To analyze the limits of the physics behind the retrieval approaches, cloud fields provided by LES model runs are used to generate synthetic radiation measurements by radiative transfer simulation. This approach has the advantage of being independent on different uncertainties introduced by real observations

and that the retrieved quantities can be compared to the truth given by the LES model. It is concluded that the bispectral retrieval shows a higher uncertainty for the retrieval of cloud droplet size compared to polarimetric retrieval while cloud optical thickness agrees between both approaches.

The results presented by this study are of high value for current and future satellite remote sensing. Retrieval uncertainties which originate from the general limitations of the retrieval algorithms are clearly quantified and may help to improve the interpretation of satellite cloud products. In this regard, the manuscript provides an important contribution to current and future research and is worth to be published.

However, in my opinion the manuscript lacks of some major issues which have to be reassessed in detail before publishing the manuscript. By neglecting measurement uncertainties, the retrieval comparison might be only of academic value because it does not reflect the real uncertainties of both retrieval approaches when real satellite observations are considered. Furthermore, new developments of the bispectral retrieval are not considered in the study and limit the conclusions for future satellite employments. The bias of the bispectral retrieval is surprisingly high considering the ideal setup of the study. A more accurate treatment of the vertical weighting function of the bispectral retrievals needs to be applied in order to guaranty the comparability with the LES and the polarimetric retrieval.

Below, I compiled a list of comments which have to be considered in a revised version of the paper. There might be some contradictory statements resulting from my misinterpretation of the text when first reading. I am sure the authors will know how to weight in such cases and how to improve the text to avoid misinterpretations by other readers.

**2 Major comments**

**Neglecting measurement uncertainties**

I understand the approach of the authors to use synthetic measurements generated from LES cloud fields and radiative transfer simulations. This approach leaves only a limited number of causes which can explain the difference between both retrieval approaches, such as the complexity of the cloud representation in the radiative transfer model (vertical profile). In this regard, the study provides good insight into the physics of the retrieval approaches. This is worth to be published but might be only of academic value. However, the conclusions on the performance of the retrieval approaches might change when measurement errors are considered. Uncertainties of the spectral radiance measured by the satellite sensors, e.g. radiometric calibration, might propagate differently in both retrieval approaches. An uncertainty of spectral radiance might have larger consequences on retrieved cloud properties compared to uncertainties in the polarimetric measurements. To judge, which retrieval algorithm provides the more accurate cloud properties when applied to real satellite measurements such a propagation of the measurement uncertainty has to be considered and analyzed. This should not replace the current results of the study. Please keep these results. I rather suggest to add an additional exercise with focus on the propagation of measurement uncertainties. On basis of the available data set, this should be easy to realize. The simulated radiances which are the exact synthetic measurement are available. By generating synthetic measurements including a measurement uncertainty and propagating through the retrieval algorithm should already give an estimate of these retrieval uncertainties. Your motivation to use a LES cloud field and IPA simulations would still hold for such a study, as 3D-radiative effect, etc. still can be ruled out. Only the propagation of pure sensor uncertainties will be analyzed.

[Figure]

**Vertical weighting functions**

As discussed by the authors, the vertical weighting function is essential to compare retrieved cloud properties with the LES model clouds. Therefore, I am wondering why a relative crude assumption for the weighting function of the bispectral retrieval is assumed. The two-way transmittance function is valid for single-scattering only which holds for the polarimetric retrieval where single scattering features are extracted from the measurements. But the bispectral retrieval certainly are effected by multiple scattering. Platnick (2000) clearly shows that the vertical weighting functions significantly extend into the lower cloud layers. Even for $3.7\,\mu$m cloud layers at optical thickness larger than $\tau > 2$ contribute to the weighting function while the 2WT weighting already becomes zero for $\tau > 2$.

First, I was surprised by the relative large differences between bispectral retrieval and LES-truth because the setup of the study was chosen well and should not allow large differences. But the treatment of the vertical weighting functions may explain these differences. Considering the idealized setup using the LES clouds and the independent pixel approximation to generate the synthetic measurements, I do not see many sources of error than the vertical distribution of cloud particles and how these are represented in the radiative transfer model. I assume, that the calculation of the synthetic measurements and the calculation of the LUTs use the same radiative transfer code. For the synthetic measurements, the vertical cloud profile is considered, but not for the LUTs of the retrieval. So the radiative transfer code itself is no issue.

The inaccurate treatment of the vertical weighting fits also to the results shown in Figure 3a. The slight shift of the bispectral retrieval to smaller particle sizes compared to the 2WT weighing might result from different vertical weighting function. While the 2WT weighting only considered the larger particles at cloud top, the bi-spectral retrieval is also influenced by smaller particles at lower cloud levels. This could already lead to

the observed differences. Therefore, I suggest to use a more realistic vertical weighting function for comparing the bispectral retrieval with the LES model. The weighting function considering multiple scattering can be easily calculated by the method presented by Platnick (2000). As an approximation the weighting functions can be calculated assuming vertically homogeneous clouds as for the retrieval LUTs. With this assumption they can be easily extracted from the LUTs as the slope of reflectance with increasing optical thickness. So all required simulations should be available.

**Radiance Ratio Retrieval**

The manuscript motivates the study by future satellite missions providing multi-angular polarimetric observations. However, also the classic bispectral observation will profit from continuous improvement of the retrieval algorithms. In recent studies, the radiance ratio retrieval approach has been proposed (actually by one of the co-authors) to reduce some limitations of the bispectral retrieval (Werner el al. 2013, Ehrlich et al. 2017, LeBlanc et al. 2015, Brückner et al. 2014). Using ratios of spectral radiance instead of absolute radiance improves the orthogonality of the LUTs and the impact of measurement uncertainties. Therefore, some limitations discussed for the bispectral retrieval might be improved. E.g. LUTs of radiance ratios are more spread for small cloud optical thickness, the PPH-bias is likely reduced having more orthogonal LUTs (similar to the differences between $2.1\,\mu$m and $3.7\,\mu$m). This improved retrieval approach should be considered in the manuscript as a third retrieval approach if it aims to be relevant for future satellite observations.

*Werner, F., Siebert, H., Pilewskie, P., Schmeissner, T., Shaw, R. A., and Wendisch, M.: New airborne retrieval approach for trade wind cumulus properties under overlying cirrus, J. Geophys. Res.-Atmos., 118, 3634–3649, https://doi.org/10.1002/jgrd.50334, 2013.*

*Ehrlich, A., Bierwirth, E., Istomina, L., and Wendisch, M.: Combined retrieval of Arctic*

[Figure]

*liquid water cloud and surface snow properties using airborne spectral solar remote sensing, Atmos. Meas. Tech., 10, 3215-3230, https://doi.org/10.5194/amt-10-3215-2017, 2017.*

*LeBlanc, S. E., Pilewskie, P., Schmidt, K. S., and Coddington, O.: A spectral method for discriminating thermodynamic phase and retrieving cloud optical thickness and effective radius using transmitted solar radiance spectra, Atmos. Meas. Tech., 8, 1361–1383, https://doi.org/10.5194/amt-8-1361-2015, 2015.*

*Brückner, M., Pospichal, B., Macke, A., and Wendisch, M.: A new multispectral cloud retrieval method for ship-based solar transmissivity measurements, J. Geophys. Res., 119, 11338–11354, https://doi.org/10.1002/2014JD021775, 2014.*

**Polarimetric Retrieval**

How meaningful are the results of the study on effects of the horizontal resolution for the polarimetric retrieval? In the motivation it was mentioned, that for POLDER a footprint of 150 km has to be used to obtain measurements of the cloudbow? This is far from the scales analysed here with the LES clouds. Is the spatial resolution of future spaceborne polarization sensors comparable to the scales analyzed in this study? The results presented in the manuscript suggest, that in the scales analyzed here, polarimetric measurements are not strongly effected by cloud inhomogeneities. Can this conclusion also be transferred to larger spatial scales? These issues should be discussed somehow in the manuscript.

**3  Minor comments**

**P1 L1:** Title: The study is limited to three very specific cases of liquid low level cloud over the ocean (trade wind cumulus, stratocumulus). At least "liquid clouds" has to be added in the title. "marine" or similar indicating, that only clouds over water have been

analyzed should also be considered. Retrieval of ice clouds will certainly differ from the study presented here. Also results for clouds over land can differ due to surface albedo and the different cloud dynamics over land (vertical profile).

**P5 L10:** The reflectance at SWIR and VNIR bands both depend on optical thickness and effective radius. It is simply wrong to indicate that the sensitivities are decoupled. The lookup table shown in the manuscript clearly reveal the non-orthogonality especially for small optical thickness. This coupling has different implication on the bispectral retrieval (PPH-bias) which partly are already used to discus the retrieval biases.

**P8 L12:** The polarized phase function and the modeled polarized reflectance are two different quantities as far as I understood. How these can be fitted to each other? The degree of linear polarization calculated from polarized reflectance would be comparable to P12.

**P9 L27:** Eq. 4: Wouldn't it be better to use/write the size of the coarser resolution pixel into brackets of the mean value. Instead $R(0.865\,\mu\mathrm{m},\ 50\,\mathrm{m})$ better $R(0.865\,\mu\mathrm{m},\ 800\,\mathrm{m})$? The mean value is calculated for the coarse resolution pixel and independent on the fine resolution of 50 m.

**P10 L11:** Is the comparison only done for a specific solar zenith angle or are all simulations mixed? In Sect. 4.2, Fig. 5 it was explicitly mentioned that all cases and geometries are included. Should be done here as well.

**P11 L3:** Footnote: Why this was written as footnote? The explanation given in the footnote should be presented directly in the main text because it is needed to understand the systematic bias. Putting such parts into a footnote only disturbs the flow of reading.

**P14 L5:** Figure 8: This comparison has to be done with respect to the LES-truth (see also comment to Figure 5). Only then you can judge which retrieval has a bias and which not. Comparing both retrieval to each other merges effects and does not tell which retrieval is closer to reality. In P 14 L 9 the differences between bispectral

and polarimetric retrieval are rated by assuming the polarimetric retrieval to be the truth. This should be avoided as also the polarimetric retrieval may have caused these differences. You should always refer to the truth solution which is given by the LES cloud fields.

**P14 L10:** typo: "less" and "lower"

**Figure 1:** Panel a): Something is wrong because the color codes do not fit to spectral bands! Likely the labeling of x-y axis is switched.

**Figure 2:** Indicate horizontal scale!

**Figure 3:** Typo in caption: "or" should be "of"

**Figure 5:** I do not see a need for these plots. Comparing both approaches separately to the LES-truth already tells where the uncertainties of the individual approaches are. Comparing both to each other makes interpretation only very difficult but does not give any new conclusions. Both retrieval have to be compared to the LES-truth. The comparison in figure 5 also results in some incorrect conclusions (at least when these are only followed from Fig. 5 alone). The polarimetric retrieval has been found to be better compared to the bispectral retrieval. But this conclusion can not come from Fig. 5 because Fig 5 does not compare to the truth values. Therefore, I suggest to remove Fig. 5 or exchange by similar comparisons with the LES-truth. Also the corresponding discussion (P 12 L 20-30) should use the LES-truths as the reference.

**Figure 6:** Some data does not fit into the LUT. Is this necessary? The range of optical thickness can be extended in your simulations? You should be able to calculate the maximum optical thickess from the LES field in advance. Or is there any other reason why these data does not fit?

**Figure 6:** What is the range of optical thickness? Can be labeled similar to the particle size.

**Figure 7:** This figure is also not needed. Both results have already been compared to

[Figure]

the LES-truth.

**Figure 8:** Very hard do distinguish the color code and circle size. Especially the size of the circles is not visible in the center of the data cloud. Only outliers are visible.

[Figure]

---

## Referee Comment (RC2) · Anonymous Referee #2 · 27 Nov 2017

This is a review of the manuscript titled "Comparisons of bispectral and polarimetric cloud microphysical retrievals using LES-Satellite retrieval simulator" submitted to AMTD by Miller et al.

The paper discusses the biases in retrieved drop effective radius and cloud optical thickness using two independent approaches, namely the bi-spectral approach and the polarimetric approach. Both methods are evaluated using simulated measurements based on large eddy simulations and 2D radiative transfer. Biases in retrieval products caused by vertical and horizontal inhomogeneity are evaluated.

The work follows previous work by the same authors, especially that published in Miller

et al. (2016) and Zhang et al. (2012). Those previous papers lacked the focus on polarimetric retrievals, so this paper is a useful addition to those studies. The polarimetric method is the more robust method, as also shown here, but requires multi-view polarization measurements at specific viewing geometries, which makes it not applicable everywhere. The polarimetric method is often seen as a means to validate the bi-spectral retrievals. Therefore, this comparison of the two methods is useful to better understand such comparisons.

However, in my opinion, some of the means of presentation are difficult to interpret and not very effective. Also, while focus is on the effective radius retrievals, the optical thickness retrievals are also evaluated, but some rather surprising outliers in the optical thickness retrievals are not explained. Finally, the authors should provide more references in the results section to put their results in perspective to other relevant papers. Below I will provide more details to these major comments and will follow with some minor comments

Major comments:

1) The conclusions about polarimetric and bi-spectral retrievals are not new. As mentioned in the paper, the bispectral method is already well studied by several papers, including Miller et al. (2016) and Zhang et al. (2012). The polarimetric results are consistent with those by, for example, Alexandrov et al. (2012) and Shang et al. (2015). All of the papers mentioned above are referenced in the manuscript, but mostly in the introduction or conclusions. The authors should provide more references in the results section and put their results in perspective to the 4 papers mentioned above, and other relevant papers. For example, the reduced sensitivity of effective variance at high values, the effects of vertical variation and insensitivity of the approach to optical thickness are all discussed by Alexandrov et al. (2012). The sensitivity to sub-pixel inhomogeneity in the polarimetric approach is discussed by Shang et al. (2015). Retrieval biases in the bi-spectral approach is discussed by Miller et al. Specifically, Miller et al. conclude that biases are especially large in "transition zone" at the cloud edges. I

find it surprising that this is not mentioned here at al. In summary, when discussing the results, please also discuss the appropriate references to set these results in context with these previous studies.

2) The optical thickness retrievals are also evaluated in this study, but hardly discussed. I find it rather surprising that even at large optical depths, large biases occasionally occur. As mentioned, the optical depth retrievals are similarly biased using the polarimetry size retrievals. Looking at equation 4, I am curious what could cause these optical depth biases if not errors in assumed size distribution? It would be informative to further explore this bias in the paper.

3) In figure 5, the effect of the fixed variance of 0.1 on the bispectral retrievals are evaluated. This is done by coupling the bispectral retrievals to the variance retrieved with the polarimetric approach. However, you showed that this retrieved variance often significantly differs from the true value. It performs best in the case around a value of 0.1, which is the value assumed for the bispectral method. So, in this test, often still a 'wrong' variance is used. It is difficult, if not impossible, to deduce solid conclusions from the present results. The authors should couple the bispectral retrievals to the true (2WT) value of variance for each gridbox for this evaluation.

4) Figure 8 convolves a lot of information and is therefore hard to interpret. The aim is to show the effect of pixel size and inhomogeneity on the bi-spectral and polarimetric technique. Here the two methods are compared, making it impossible to tell which technique is biased where and how. The authors concluded that "it is evident that as the spatial retrieval footprint reaches 800 m the sub-pixel inhomogeneity tends to increase and the re(2.13 $\mu$m) retrieval suffers from an increasingly high bias relative to the polarimetric retrieval." I do not get this from looking at this figure. Using the colors and open circles of different sizes produces a pretty colorful blob, but individual circles are impossible to spot mostly. I see some large circles with high biases, but have the impression that most are hiding in the colorful blob. The conclusions and abstract state that the methods compare well for high resolutions, but biases appear

at coarser resolution. This conclusion appears to be based on this figure. Looking at figure 5, I see some very large differences between the methods for some cases, so the bispectral method also has issues at these fine resolutions, if not more. Also, the abstract and conclusion states "This bias largely stems from differences related to sensitivity of the two retrievals to unresolved inhomogeneities in effective variance and optical thickness." This suggests that the polarimetric retrievals also have sensitivity to the spatial resolution. They probably have somewhat (as concluded by Shang et al.), but this is not at all evident from this figure and analysis. Please produce figures that more clearly and systematically support these conclusions (or other conclusions). I suggest producing separate plots showing biases from true effective radius values in bi-spectral and polarimetric techniques as a function of H. Possibly the resolution can be on the y-axis and biases can be color coded?

Minor comments:

In figure 3, the cases with tau<3 are removed, revealing a better result. However, I am wondering how the 2D histogram for tau<3 looks like. Do the bulk of these retrievals still perform well, or are they all biased? That is not clear from these plots.

Figure 4 shows that the uncertainty in the retrievals of effective variance is rather high. Firstly, please relate your findings with those found by Alexandrov et al. (2012). Secondly, please discuss the appropriateness of the assumption of a gamma distribution for these LES fields. Is the model producing size distributions that can be well described by a gamma distribution? If not, this could explain part of the spread found in your results. Also, please discuss the possibility of non-parametric size distributions from polarimetry, as presented by Alexandrov et al. (2012; J. Quant. Spectrosc. Radiat. Transfer, 113, 2521-2535, doi:10.1016/j.jqsrt.2012.03.025.)

Figure 9 shows the variation in polarization measurements for sub-pixels in a 800m pixel. For the case including the thin cloud parts, there appears to be a substantial spread, but this is mostly in absolute magnitude. Please note in the text that the polarimetry technique is not sensitive to the absolute magnitude of the measurements, and these variations are therefore not an issue for this technique.

Text edits:

Page 2, line 9: add "which" before "simultaneously" Page 2, line 13: Add "Suomi" in front of "National" (and NPP)

---

## Author Response (AR1)

**Author Responses:**

**Anonymous Referee #1**

**General remarks:**
*The manuscripts provides a detailed analyses of two well established cloud retrieval methods using passive satellite measurements of solar radiation. Bispectral and polarimetric retrieval of cloud optical thickness, droplet effective radius, and effective variance are compared for cases of liquid maritime clouds. The study is not based on measurements of real clouds. To analyze the limits of the physics behind the retrieval approaches, cloud fields provided by LES model runs are used to generate synthetic radiation measurements by radiative transfer simulation. This approach has the advantage of being independent on different uncertainties introduced by real observations and that the retrieved quantities can be compared to the truth given by the LES model. It is concluded that the bispectral retrieval shows a higher uncertainty for the retrieval of cloud droplet size compared to polarimetric retrieval while cloud optical thickness agrees between both approaches.*

*The results presented by this study are of high value for current and future satellite remote sensing. Retrieval uncertainties, which originate from the general limitations of the retrieval algorithms are clearly quantified and may help to improve the interpretation of satellite cloud products. In this regard, the manuscript provides an important contribution to current and future research and is worth to be published.*

*However, in my opinion the manuscript lacks of some major issues which have to be reassessed in detail before publishing the manuscript. By neglecting measurement uncertainties, the retrieval comparison might be only of academic value because it does not reflect the real uncertainties of both retrieval approaches when real satellite observations are considered. Furthermore, new developments of the bispectral retrieval are not considered in the study and limit the conclusions for future satellite employments. The bias of the bispectral retrieval is surprisingly high considering the ideal setup of the study. A more accurate treatment of the vertical weighting function of the bispectral retrievals needs to be applied in order to guaranty the comparability with the LES and the polarimetric retrieval. Below, I compiled a list of comments which have to be considered in a revised version of the paper. There might be some contradictory statements resulting from my misinterpretation of the text when first reading. I am sure the authors will know how to weight in such cases and how to improve the text to avoid misinterpretations by other readers.*

**Major comments:**

1. **Neglecting measurement uncertainties**

   *I understand the approach of the authors to use synthetic measurements generated from LES cloud fields and radiative transfer simulations. This approach leaves only a limited number of causes which can explain the difference between both retrieval approaches, such as the complexity of the cloud representation in the radiative transfer model (vertical profile). In this regard, the study provides good insight into the physics of the retrieval approaches. This is worth to be published but might be only of academic value. However, the conclusions on the performance of the retrieval approaches might change when measurement errors are considered. Uncertainties of the spectral radiance measured by the satellite sensors, e.g. radiometric calibration, might propagate differently in both retrieval approaches. An uncertainty of spectral radiance might have larger consequences on retrieved cloud properties compared to uncertainties in the polarimetric measurements. To judge, which retrieval algorithm provides the more accurate cloud properties when applied to real satellite measurements such a propagation of the measurement uncertainty has to be considered and analyzed. This should not replace the current results of the study. Please keep these results. I rather suggest to add an additional exercise with focus on the propagation of measurement uncertainties. On basis of the available data set, this should be easy to realize. The simulated radiances which are the exact synthetic measurement are available. By generating synthetic measurements including a measurement uncertainty and propagating through the retrieval algorithm should already give an estimate of these retrieval uncertainties. Your motivation to use a LES cloud field and IPA simulations would still hold for such a study, as 3D-radiative effect, etc. still can be ruled out. Only the propagation of pure sensor uncertainties will be analyzed.*

   a. I have added a section discussing the impact of uncorrelated uncertainty on bispectral and polarimetric retrievals. While the discussion is admittedly limited because of the desire of not choosing a specific instrument uncertainty model, I feel like this current approach puts the rest of the biases discussed into context. It should be noted however that the behavior of uncertainty for these two retrievals is highly algorithm and instrument dependent.

2. **Vertical weighting functions**

   *As discussed by the authors, the vertical weighting function is essential to compare retrieved cloud properties with the LES model clouds. Therefore, I am wondering why a relative crude assumption for the weighting function of the bispectral retrieval is assumed. The two-way transmittance function is valid for single-scattering only which holds for the polarimetric retrieval where single scattering features are extracted from the measurements. But the bispectral retrieval certainly are effected by multiple scattering. Platnick (2000) clearly*

*shows that the vertical weighting functions significantly extend into the lower cloud layers. Even for 3.7 μm cloud layers at optical thickness larger than τ > 2 contribute to the weighting function while the 2WT weighting already becomes zero for τ > 2.*

*First, I was surprised by the relative large differences between bispectral retrieval and LES-truth because the setup of the study was chosen well and should not allow large differences. But the treatment of the vertical weighting functions may explain these differences. Considering the idealized setup using the LES clouds and the independent pixel approximation to generate the synthetic measurements, I do not see many sources of error than the vertical distribution of cloud particles and how these are represented in the radiative transfer model. I assume, that the calculation of the synthetic measurements and the calculation of the LUTs use the same radiative transfer code. For the synthetic measurements, the vertical cloud profile is considered, but not for the LUTs of the retrieval. So the radiative transfer code itself is no issue.*

*The inaccurate treatment of the vertical weighting fits also to the results shown in Figure 3a. The slight shift of the bispectral retrieval to smaller particle sizes compared to the 2WT weighing might result from different vertical weighting function. While the 2WT weighting only considered the larger particles at cloud top, the bi-spectral retrieval is also influenced by smaller particles at lower cloud levels. This could already lead to the observed differences. Therefore, I suggest to use a more realistic vertical weighting function for comparing the bispectral retrieval with the LES model. The weighting function considering multiple scattering can be easily calculated by the method presented by Platnick (2000). As an approximation the weighting functions can be calculated assuming vertically homogeneous clouds as for the retrieval LUTs. With this assumption they can be easily extracted from the LUTs as the slope of reflectance with increasing optical thickness. So all required simulations should be available.*

a. I am not sure that a vertical weighting based on homogeneous vertical profile assumptions would be appropriate for this analysis. The LES clouds are not vertically homogeneous and there can be significant extinction cross-section variability within cloud vertical profiles. The intent of the applied vertical weighting techniques is to account for that vertical inhomogeneity directly.

b. The authors agree that the single scattering (2WT) vertical weighting may not sufficiently describe the behavior of scattering in the 2.13 μm spectral band. One of the reasons for implementing a single vertical weighting definition throughout this paper was to ensure that we were not comparing retrievals to a "moving target." Additionally, from our previous work in Miller et al. (2016) we knew that $r_e$(2WT) matched the $r_e$(3.75 μm) spectral retrieval reasonably well.

c. Despite the large variability in the comparison of vertical weighting and bispectral retrievals it is important to note that the mean biases are quite small. The logarithmic histogram in the joint PDF can possibly be emphasizing features that are occurring for clouds that have significant vertically inhomogeneous cloud tops.

d. To test the impact of using a more accurate vertical weighting function we implemented the approach described in equation 4 of Zhang et al. 2017, which includes an additional factor that accounts for multiple scattering contributions:

$$W(\tau) = c\tau^b \exp\left[-\tau\left(\frac{1}{\mu} + \frac{1}{\mu_0}\right)\right].$$

Where the $\tau^b$ factor is introduced to account for multiple scattering, and the rest is the same as previously defined in the paper. For b=0 we get back the original 2WT vertical weighting used previously. Below is a figure displaying the scene-averaged vertical weighting functions from the DYCOMS-II case, note that similar results (throughout this discussion) were found in other LES cases. Each of the curves corresponds to a different value of b and each is also color coded to indicate the resulting average vertically weighted cloud microphysical property $r_e$(VW) and $v_e$(VW). Increasing the value of b causes the weighting function to extend deeper into the cloud, changing the mean value of $r_e$(VW) and $v_e$(VW). The change in the mean value of $r_e$(vw) for these different weightings does not exceed 0.5 µm and the mean value of $v_e$(VW) changes less than 0.05. The small variability in $r_e$ has a lot to do with the actual profile of $r_e(\tau)$, which has little variability near cloud top. In contrast the large variability in $v_e$ is related to the rapid change in $v_e(\tau)$ just below cloud top. It should be noted that the b=0 vertical weighting has a similar $r_e$(VW) result as the b=9 case, which offers an explanation as to why $r_e$(2WT) just happened to be a decent point of comparison.

[Figure]

**Figure 1: Scene average vertical weighting functions for the DYCOMS-II case with varying values of the parameter $b$. Lines are colored by the corresponding vertically weighted properties ($r_e$(VW,2.13) or $v_e$(VW,2.13))**

Using this parameterization, we can tune the "quality of vertical weighting" by comparing the shape of the distribution of the bias between $r_e(2.13)$ or $r_e$ (3.75) with respect to their respective vertical weightings $r_e$(VW,2.13) and $r_e$(VW,3.75). By searching this histogram with different values of $b$ we can find a parameterization that minimizes the mean bias, as well as the variability of the distribution – indicating a more appropriate vertical weighting definition for the bispectral retrieval in question. Several animated gif's of the behavior of this histogram as a function of b can be found here:

$r_e$(2.13)-$r_e$(VW,2.13) Histogram varying $b$
$r_e$(2.13)-$r_e$(VW,2.13) Histogram varying $b$ (FOR $\tau$>5 ONLY!)
$r_e$(3.75)-$r_e$(VW,3.75) histogram varying $b$

$r_e$(3.75,coupled to $v_e$(pol))-$r_e$(VW,3.75) Histogram varying $b$

Where for the 3.75μm spectral band we found that a $b$ coefficient of 0.3 minimized mean bias and variability (when coupled with $v_e$ information), whereas for 2.13μm a value of $b$=10 was found to be optimal. It is also important to note that the appropriate vertical weighting also depends on total optical thickness. As a consequence of the deep penetration of the 2.13 vertical weighting function some extremely high biased values of $r_e$(VW) exist for $\tau_{tot}$<5. This threshold was selected because $\tau$=5 is roughly the location of the peak of this vertical weighting function. For the ~5% of profiles with $\tau_{tot}$<5 we simply define $b$=0 in order to avoid applying an inaccurate multiple scattering vertical weighting.

The regression histograms of these new results are shown in Figure 2. The primary finding of this comparison is that the new flexible vertical weighting function produces results for the 2.13 μm band that have a far tighter regression. There is also a new bump in the comparison for 2.13 with $\tau_{tot}$>3 only, but this is associated with the population of high-biased $r_e$(VW) that remain below $\tau_{tot}$<5.

I also found a possible mistake in my plotting routines, I was only plotting nadir viewing $r_e$(2WT) and $v_e$(2WT) values for comparisons to the polarimetric retrieval. I have modified these results to include the other viewing geometries and those are shown in Figure 3.

[Figure]

**Figure 2:** Refer to Figure 3 of the original manuscript for more figure information. This new version compares to a more flexible vertical weighting definition (denoted VW) than the single-scattering 2WT scheme.

[Figure]

**Figure 3: Refer to Figure 4 of the original manuscript for more figure information. These polarimetric retrieval comparison plots have been updated to include data from all the same viewing geometries used in the bispectral retrieval.**

Zhang, Z., X. Dong, B. Xi, H. Song, P. L. Ma, S. J. Ghan, S. Platnick, and P. Minnis (2017), Intercomparisons of marine boundary layer cloud properties from the ARM CAP-MBL campaign and two MODIS cloud products, *J. Geophys. Res.*, *122*(4), 2351–2365, doi:10.1002/2016JD025763.

**3. Radiance Ratio Retrieval**

*The manuscript motivates the study by future satellite missions providing multi-angular polarimetric observations. However, also the classic bispectral observation will profit from continuous improvement of the retrieval algorithms. In recent studies, the radiance ratio retrieval approach has been proposed (actually by one of the co-authors) to reduce some limitations of the bispectral retrieval (Werner el al. 2013, Ehrlich et al. 2017, LeBlanc et al. 2015,*

*Brückner et al. 2014). Using ratios of spectral radiance instead of absolute radiance improves the orthogonality of the LUTs and the impact of measurement uncertainties. Therefore, some limitations discussed for the bispectral retrieval might be improved. E.g. LUTs of radiance ratios are more spread for small cloud optical thickness, the PPH-bias is likely reduced having more orthogonal LUTs (similar to the differences between 2.1 μm and 3.7 μm). This improved retrieval approach should be considered in the manuscript as a third retrieval approach if it aims to be relevant for future satellite observations.*

Werner, F., Siebert, H., Pilewskie, P., Schmeissner, T., Shaw, R. A., and Wendisch, M.: New airborne retrieval approach for trade wind cumulus properties under overlying cirrus, J. Geophys. Res.-Atmos., 118, 3634–3649, https://doi.org/10.1002/jgrd.50334, 2013.

Ehrlich, A., Bierwirth, E., Istomina, L., and Wendisch, M.: Combined retrieval of Arctic C5 liquid water cloud and surface snow properties using airborne spectral solar remote sensing, Atmos. Meas. Tech., 10, 3215-3230, https://doi.org/10.5194/amt-10-3215- 2017, 2017.

LeBlanc, S. E., Pilewskie, P., Schmidt, K. S., and Coddington, O.: A spectral method for discriminating thermodynamic phase and retrieving cloud optical thickness and effective radius using transmitted solar radiance spectra, Atmos. Meas. Tech., 8, 1361– 1383, https://doi.org/10.5194/amt-8-1361-2015, 2015.

Brückner, M., Pospichal, B., Macke, A., and Wendisch, M.: A new multispectral cloud retrieval method for ship-based solar transmissivity measurements, J. Geophys. Res., 119, 11338–11354, https://doi.org/10.1002/2014JD021775, 2014.

a. We did a sensitivity study comparing the standard MODIS approach to a ratio retrieval built using the available bands in the MODIS LUT reflectance library. We found that for plane-parallel homogeneous clouds the ratio retrieval resulted in a reduction of uncertainty << 1%. Both retrieval approaches have significant issues for low optical thicknesses. The reflectance ratio method discussed in Werner et al. 2013 makes use of hyperspectral observations with 2 nm resolution. Our current thinking is that this approach becomes less fruitful with wider spectral bands. The MODIS spectral bands are significantly wider than this (on the order of 100nm), as are many imaging radiometers as a result of instrument design trade offs between spectral resolution and spatial coverage/resolution. Because the LUT used in this study is based on MODIS it is unfortunately not straightforward to reimplement a ratio retrieval for a hypothetical hyperspectral instrument using the same datasets.

b. One of the primary reasons for applying the ratio retrieval is to minimize the PPH bias associated with the impact of the curvature of the NJK LUT for broken/inhomogeneous clouds (at coarse resolutions). For the standard Nakajima-King approach, our group has been working on a technique that can provide a simple correction or estimate of this bias discussed in Zhang et al. (2016) and others. As a consequence, we believe that the PPH bias will be less severe with the development of this technique -- removing one of the motives for implementing a ratio retrieval.

Zhang, Z., F. Werner, H. M. Cho, and G. Wind (2016), A framework based on 2-D Taylor expansion for quantifying the impacts of subpixel reflectance variance and covariance on cloud optical thickness and effective radius retrievals based on the bispectral method, *Journal of Geophysical Research: Atmospheres*, doi:10.1063/1.4975502.

Werner, F., Siebert, H., Pilewskie, P., Schmeissner, T., Shaw, R. A., and Wendisch, M.: New airborne retrieval approach for trade wind cumulus properties under overlying cirrus, J. Geophys. Res.-Atmos., 118, 3634–3649, https://doi.org/10.1002/jgrd.50334, 2013.

4. **Polarimetric Retrieval**
*How meaningful are the results of the study on effects of the horizontal resolution for the polarimetric retrieval? In the motivation it was mentioned, that for POLDER a footprint of 150 km has to be used to obtain measurements of the cloudbow? This is far from the scales analyzed here with the LES clouds. Is the spatial resolution of future spaceborne polarization sensors comparable to the scales analyzed in this study? The results presented in the manuscript suggest, that in the scales analyzed here, polarimetric measurements are not strongly effected by cloud inhomogeneities. Can this conclusion also be transferred to larger spatial scales? These issues should be discussed somehow in the manuscript.*

a. You are correct to point out a "vastness of scale" problem between LES (50 m) and the POLDER satellite retrievals (~50-150 km depending on what paper you look at). The intent of this work is not to compare to the POLDER instrument, but to the newer actively developing spaceborne polarimeters that are expected to launch in the next decade. Perhaps we can make that argument more clearly in the introduction. The POLDER retrieval and its resolution are highlighted for historical context on spaceborne polarimetric retrievals because it's still the only spaceborne polarimeter that has looked at cloud microphysics.

b. The latest polarimetric instruments intended for spaceborne applications such as HARP (see link below) are currently aiming for a retrieval spatial resolution of ~2.5 km for cloud retrievals. That means that the LES retrievals shown here at up to 800 m resolutions are within an order of magnitude of the expected results. At 800 m we have 64 pixels of observations, and to avoid the poor statistics for coarse resolutions we stopped there. It should also be noted that the airborne versions of the polarimeters currently being developed often make observations at nadir resolutions within the ranges discussed here.
https://userpages.umbc.edu/~martins/laco/harp.htm

**Minor comments**

1. *P1 L1: Title: The study is limited to three very specific cases of liquid low level cloud over the ocean (trade wind cumulus, stratocumulus). At least "liquid clouds" has to be added in the title. "marine" or similar indicating, that only clouds over water have been analyzed should also be considered. Retrieval of ice clouds will certainly differ from the study presented here. Also results for clouds over land can differ due to surface albedo and the different cloud dynamics over land (vertical profile).*

   a. Response: This is a good point. The title will be amended as follows to indicate the focus on marine boundary layer clouds:
   "Comparisons of bispectral and polarimetric retrievals of marine boundary layer cloud microphysics: Case studies using a LES-satellite retrieval simulator"

2. *P5 L10: The reflectance at SWIR and VNIR bands both depend on optical thickness and effective radius. It is simply wrong to indicate that the sensitivities are decoupled. The lookup table shown in the manuscript clearly reveal the non-orthogonality especially for small optical thickness. This coupling has different implication on the bispectral retrieval (PPH-bias) which partly are already used to discus the retrieval biases.*

   a. This was not the intention, I will parse that sentence more carefully. Refer to amended statement on line 16 of page 5.

3. *P8 L12: The polarized phase function and the modeled polarized reflectance are two different quantities as far as I understood. How these can be fitted to each other? The degree of linear polarization calculated from polarized reflectance would be comparable to P12.*

a. All retrievals in this study are performed on polarized reflectance (Q) after transformation of the polarization state to the principal plane (so that the magnitude of U is approximately 0). The polarized phase function is equal to the polarized reflectance Q in the principal plane for single scattering. The degree of linear polarization is not used in this retrieval.

4. *P9 L27: Eq. 4: Wouldn't it be better to use/write the size of the coarser resolution pixel into brackets of the mean value. Instead R(0.865 μm, 50 m) better R(0.865 μm, 800 m)? The mean value is calculated for the coarse resolution pixel and independent on the fine resolution of 50 m.*
   a. No, the mean value at 800m is calculated using the 50m data. The values inside the brackets are simply the highest resolution data I have. For example, an 800m pixel has 16x16 pixels at 50m within it that are used in this calculation to get the mean and standard deviation. Hopefully this has been further clarified in the text.

5. *P10 L11: Is the comparison only done for a specific solar zenith angle or are all simulations mixed? In Sect. 4.2, Fig. 5 it was explicitly mentioned that all cases and geometries are included. Should be done here as well. P11 L3: Footnote: Why this was written as footnote? The explanation given in the footnote should be presented directly in the main text because it is needed to understand the systematic bias. Putting such parts into a footnote only disturbs the flow of reading.*
   a. All geometries and simulations are mixed; I will make an effort to highlight that in the text. And the footnote will be worked in to flow with the test.

6. *P14 L5: Figure 8: This comparison has to be done with respect to the LES-truth (see also comment to Figure 5). Only then you can judge which retrieval has a bias and which not. Comparing both retrieval to each other merges effects and does not tell which retrieval is closer to reality. In P14 L9 the differences between bispectral and polarimetric retrieval are rated by assuming the polarimetric retrieval to be the truth. This should be avoided as also the polarimetric retrieval may have caused these differences. You should always refer to the truth solution which is given by the LES cloud fields.*
   a. Refer to the response to the second reviewer regarding figure 8.

7. *P14 L10: typo: "less" and "lower"*

8. *Figure 1: Panel a): Something is wrong because the color codes do not fit to spectral bands! Likely the labeling of x-y axis is switched.*
   a. Indeed, this was an accidental figure editing error that got fixed already but didn't make it into the submitted copy. It is now updated correctly.

9. *Figure 2: Indicate horizontal scale!*
   a. I can include axes labels for these, but they would get even smaller... I have indicated resolution and scale in the figure as an alternative.

10. *Figure 3: Typo in caption: "or" should be "of"*

11. *Figure 5: I do not see a need for these plots. Comparing both approaches separately to the LES-truth already tells where the uncertainties of the individual approaches are. Comparing both to each other makes interpretation only very difficult but does not give any new conclusions. Both retrieval have to be compared to the LES-truth. The comparison in figure 5 also results in some incorrect conclusions (at least when these are only followed from Fig. 5 alone). The polarimetric retrieval has been found to be better compared to the bispectral retrieval. But this conclusion can not come from Fig. 5 because Fig 5 does not compare to the truth values. Therefore, I suggest to remove Fig. 5 or exchange by similar comparisons with the LES-truth. Also the corresponding discussion (P 12 L 20-30) should use the LES-truths as the reference.*
   a. These plots are necessary because, as indicated in the introduction, this is the sort of plot used in any instrument intercomparison. Put another way, for observational data sets, there is no such thing as an LES truth. Both retrieval approaches have already been compared to the LES truth in Figure 3 and 4. I added additional discussion at the opening of the intercomparison section to highlight the purpose of this approach.

12. *Figure 6: Some data does not fit into the LUT. Is this necessary? The range of optical thickness can be extended in your simulations? You should be able to calculate the maximum optical thickness from the LES field in advance. Or is there any other reason why these data does not fit?*
   a. The LUT has a maximum optical thickness below the maximum optical thickness of the LES. Unfortunately, I did not have this version of the LES when we first made the LUT several years ago and so here we are today. At this point it would be computationally expensive to extend the LUT for all geometries and combinations of $r_e$, $v_e$, $\tau$ – and it wouldn't alter any conclusions.

13. *Figure 6: What is the range of optical thickness? Can be labeled similar to the particle size.*
   a. The range of the optical thickness was stated in the background section ($\tau$=[0.1:100] with 101 logarithmically spaced grid points). I would prefer not to further clutter the figure as optical thickness isn't the primary focus on the paper.

14. *Figure 7: This figure is also not needed. Both results have already been compared to the LES-truth. Figure 8: Very hard do distinguish the color code*

*and circle size. Especially the size of the circles is not visible in the center of the data cloud. Only outliers are visible*

    a. As stated previously, the comparison figures are part of the primary objective of the paper. Folks comparing these two techniques to one another once they are in orbit don't have the advantage of comparison to LES, so this type of plot is useful in their context. This figure is intended to emphasize that the two retrievals of $\tau$ perform approximately the same when they are compared to one another.

    b. Regarding figure 8 we have made substantial changes to the way we are attempting to visually display this conclusion. Refer to our response to the second reviewer for more information.

**Anonymous Referee #2**

*This is a review of the manuscript titled "Comparisons of bispectral and polarimetric cloud microphysical retrievals using LES-Satellite retrieval simulator" submitted to AMTD by Miller et al. The paper discusses the biases in retrieved drop effective radius and cloud optical thickness using two independent approaches, namely the bi-spectral approach and the polarimetric approach. Both methods are evaluated using simulated measurements based on large eddy simulations and 2D radiative transfer. Biases in retrieval products caused by vertical and horizontal inhomogeneity are evaluated. The work follows previous work by the same authors, especially that published in Miller et al. (2016) and Zhang et al. (2012). Those previous papers lacked the focus on polarimetric retrievals, so this paper is a useful addition to those studies. The polarimetric method is the more robust method, as also shown here, but requires multi-view polarization measurements at specific viewing geometries, which makes it not applicable everywhere. The polarimetric method is often seen as a means to validate the bi-spectral retrievals. Therefore, this comparison of the two methods is useful to better understand such comparisons. However, in my opinion, some of the means of presentation are difficult to interpret and not very effective. Also, while focus is on the effective radius retrievals, the optical thickness retrievals are also evaluated, but some rather surprising outliers in the optical thickness retrievals are not explained. Finally, the authors should provide more references in the results section to put their results in perspective to other relevant papers. Below I will provide more details to these major comments and will follow with some minor comments*

**Major comments:**

1. The conclusions about polarimetric and bi-spectral retrievals are not new. As mentioned in the paper, the bispectral method is already well studied by several papers, including Miller et al. (2016) and Zhang et al. (2012). The polarimetric results are consistent with those by, for example, Alexandrov et al. (2012) and Shang et al. (2015). All of the papers mentioned above are referenced in the manuscript, but mostly in the introduction or conclusions. The authors should provide more references in the results section and put their results in perspective to the 4 papers mentioned above, and other relevant papers. For example, the reduced sensitivity of effective variance at high values, the effects of vertical variation and insensitivity of the approach to optical thickness are all discussed by Alexandrov et al. (2012). The sensitivity to sub-pixel inhomogeneity in the polarimetric approach is discussed by Shang et al. (2015). Retrieval biases in the bi-spectral approach is discussed by Miller et al. Specifically, Miller et al. conclude that biases are especially large in "transition zone" at the cloud edges. I find it surprising that this is not mentioned here at all. In summary, when discussing the results, please also discuss the appropriate references to set these results in context with these previous studies.

a. We have worked to integrate more citations and commentary on previous results into the data and analysis sections. Hopefully this more fully acknowledges that these uncited observations are not new, that was not our intention.

b. The importance of the cloud top transition zone discussed in Miller et al. 2016 is tied more directly to LWP retrievals than it is to the re retrieval directly. Getting a droplet size that is reduced from its maximum by entrainment processes can bias the LWP because of assumptions made about cloud vertical profile. However, in the context of retrieving the cloud top droplet size, the transition zone is a valid modification of the cloud top microphysics. It should be noted, that the transition zone would also modify the vertical weighting, so the both retrievals and the LES proxy should agree with one another in the presence of the transition zone.

c. In light of the previous point, some further commentary about the applications of cloud droplet size retrievals to different science questions was included at the end of the summary and discussion.

2. *The optical thickness retrievals are also evaluated in this study, but hardly discussed. I find it rather surprising that even at large optical depths, large biases occasionally occur. As mentioned, the optical depth retrievals are similarly biased using the polarimetry size retrievals. Looking at equation 4, I am curious what could cause these optical depth biases if not errors in assumed size distribution? It would be informative to further explore this bias in the paper.*

a. The accuracy of all of these optical thickness retrievals depends on whether the VNIR band asymmetry parameters (phase functions) and extinction efficiencies associated with those distributions and their effective radii deviate significantly from values associated with the same effective radius in the LUTs (gamma dist., assumed variance). In this sense, the droplet size distributions (DSD) of the LES are not exactly gamma-distributed, and thus the combination of phase functions, effective radii, and variances, are not exactly connected directly with the assumption of the form assumed by the bispectral and polarimetric retrievals. For example, the LES distributions sometimes have multiple distribution modes. As a consequence, the size distribution differences reduce the cross section distribution (second moment of the DSD) that is used to define $\tau_{tot}$. This difference in size distribution becomes even more severe for the distributions that are clearly non-gamma with ve>0.3. This is demonstrated by the figure below depicting the median DSD's and cross section distributions for each LES case. The results highlight that the cross section distribution has a reduced peak relative to the expected gamma-distribution. This largely stems from the increased number of droplets in small bins (relative to gamma) as well as an increased number of droplets in the larger bins. Surprisingly, the area and

volume distributions of these populations are similar in a relative sense, resulting in values of $r_e$ that are consistent, but low biasing $\tau_{\text{tot}}$.

[Figure]

Figure 4: The panels on the left depict droplet size distributions for different populations of each LES case. The red curves denote distributions with non-gamma $v_e > 0.3$, and the blue curves indicate the population with $v_e < 0.3$. The black curves indicate the gamma distribution corresponding to the scene average $r_e$ and $v_e$ combination. The panels on the right are area distribution populations derived from the DSD's on the left.

b. Biases can also be associated with the vertical profile assumption in the LUT. The vertical distribution of the extinction coefficient is not homogeneous as conventionally assumed in the formulation of the bispectral LUT. In the adiabatic model the cloud top $r_e$ is functionally defined with respect to $\tau_{tot}$, indicating that it might be possible that biases in retrieval of one would impact the other.

c. I don't know if there is a lot that can be done regarding these limitations in the context of the two retrieval approaches discussed in this study. Creating retrieval LUT that assumes all clouds are vertically adiabatic is problematic. And the addition of more complicated size distribution assumptions isn't easily implemented in either of these techniques in a practical sense. However, for the droplet size distribution shape assumption, we can perhaps further highlight the Rainbow Fourier Transform method of Alexandrov et al. (2012). for evaluating the droplet size distribution shape and it's impact on retrieval techniques that assume gamma distributions.

3. *In figure 5, the effect of the fixed variance of 0.1 on the bispectral retrievals are evaluated. This is done by coupling the bispectral retrievals to the variance retrieved with the polarimetric approach. However, you showed that this retrieved variance often significantly differs from the true value. It performs best in the case around a value of 0.1, which is the value assumed for the bispectral method. So, in this test, often still a 'wrong' variance is used. It is difficult, if not impossible, to deduce solid conclusions from the present results. The authors should couple the bispectral retrievals to the true (2WT) value of variance for each gridbox for this evaluation.*

    a. While this could be changed, the results will be quite similar because the polarimetric retrieval of $v_e$ behaves quite well below $v_e$=0.15, which is shown.

    b. One of the reasons why we coupled to the $v_e$(pol) retrieval was to highlight how the bispectral and polarimetric retrievals could compliment one another. We have added a brief statement of this purpose to the paper. It is also essentially a more formalized comparison of the argument first presented in figure 9 and 10 of Miller et al. (2016). Also as you already noted, this change has very little impact on the quality of the bispectral 2.13 µm retrieval, presumably because of the disconnection between the vertical weighting of the polarimetric retrieval and the 2.13 µm bispectral retrieval.

4. *Figure 8 convolves a lot of information and is therefore hard to interpret. The aim is to show the effect of pixel size and inhomogeneity on the bi-spectral and*

*polarimetric technique. Here the two methods are compared, making it impossible to tell which technique is biased where and how. The authors concluded that "it is evident that as the spatial retrieval footprint reaches 800 m the sub-pixel inhomogeneity tends to increase and the re(2.13 μm) retrieval suffers from an increasingly high bias relative to the polarimetric retrieval." I do not get this from looking at this figure. Using the colors and open circles of different sizes produces a pretty colorful blob, but individual circles are impossible to spot mostly. I see some large circles with high biases, but have the impression that most are hiding in the colorful blob. The conclusions and abstract state that the methods compare well for high resolutions, but biases appear at coarser resolution. This conclusion appears to be based on this figure. Looking at figure 5, I see some very large differences between the methods for some cases, so the bispectral method also has issues at these fine resolutions, if not more. Also, the abstract and conclusion states "This bias largely stems from differences related to sensitivity of the two retrievals to unresolved inhomogeneities in effective variance and optical thickness." This suggests that the polarimetric retrievals also have sensitivity to the spatial resolution. They probably have somewhat (as concluded by Shang et al.), but this is not at all evident from this figure and analysis. Please produce figures that more clearly and systematically support these conclusions (or other conclusions). I suggest producing separate plots showing biases from true effective radius values in bi-spectral and polarimetric techniques as a function of H. Possibly the resolution can be on the y-axis and biases can be color coded?*

    a.  One of the biggest problems with the original figure, as you point out, is that the spatial resolution information is hard to display simultaneously with the retrieval comparison. Also, because $H_\sigma$ is the only variable that has a physical dependence on spatial resolution, showing their correlations separately actually makes more sense. We have attempted to deconvolve this plot further by breaking it into multiple parts.  First of all, the following figures focus on the ATEX cases because there are more broken and inhomogeneous clouds in these two scenes than in DYCOMS-II.

    b.  The first part (in Figure 5) demonstrates the increasing population of inhomogeneous observations at coarser spatial resolutions by displaying the change in the histogram of $H_\sigma$ as a function of resolution.  This drives home the point that at coarse resolutions there is more inhomogeneity.

    c.  The second part (in Figure 6) displays joint histograms of retrieval bias (relative vertically weighted LES properties) and $H_\sigma$ for the ATEX cloud cases. Note that these figures amass all of the data from the coarse resolution (100 to 800 m) ATEX cases because they have the most diversity in terms of $H_\sigma$.

    d.  I think that these figures provide a more detailed perspective than the previous versions so we will be modifying the manuscript to include them. Mixing all of the coarse spatial resolutions together for the joint

histograms is a valid exercise because the primary difference between a joint histogram of a single coarse resolution and all of them is largely just the sampling of the $H_\sigma$ distribution.

[Figure]

Figure 5: Distributions of $H_\sigma$ for the ATEX polluted and clean cases at all coarsened spatial resolution (100, 200, 300, 400, 800 m).

[Figure]

**Figure 6: Joint histograms of retrieval biases (relative to VW) with respect to $H_\sigma$ for the ATEX clean and polluted cases under all observation geometries at coarsened spatial resolution (100, 200, 300, 400, 800 m). The color bar indicates percent occurrence.**

**Minor comments:**

1. *In figure 3, the cases with tau<3 are removed, revealing a better result. However, I am wondering how the 2D histogram for tau<3 looks like. Do the bulk of these retrievals still perform well, or are they all biased? That is not clear from these plots.*

2. *Figure 4 shows that the uncertainty in the retrievals of effective variance is rather high. Firstly, please relate your findings with those found by Alexandrov et al. (2012). Secondly, please discuss the appropriateness of the assumption of a gamma distribution for these LES fields. Is the model producing size distributions that can be well described by a gamma distribution? If not, this could explain part of the spread found in your results. Also, please discuss the possibility of non-parametric size distributions from polarimetry, as presented by Alexandrov et al. (2012; J. Quant. Spectrosc. Radiat. Transfer, 113, 2521-2535, doi:10.1016/j.jqsrt.2012.03.025.)*
   a. The paper now highlights the possibility of deviations from the gamma-distribution assumption. I also cited the rainbow fourier transform paper again during that discussion.

3. *Figure 9 shows the variation in polarization measurements for sub-pixels in a 800m pixel. For the case including the thin cloud parts, there appears to be a substantial spread, but this is mostly in absolute magnitude. Please note in the text that the polarimetry technique is not sensitive to the absolute magnitude of the measurements, and these variations are therefore not an issue for this technique.*
   a. This has been noted in the text.

Minor text edits:
1. *Page 2, line 9: add "which" before "simultaneously"*
2. *Page 2, line 13: Add "Suomi" in front of "National" (and NPP)*

| Page 9: [1] Deleted | Daniel Miller | 2/8/18 2:20 PM |
|---|---|---|

corresponds

| Page 9: [2] Deleted | Daniel Miller | 2/13/18 10:50 AM |
|---|---|---|

2WT

| Page 9: [3] Deleted | Daniel Miller | 3/4/18 12:49 PM |
|---|---|---|

directly

| Page 9: [4] Moved to page 9 (Move #2) | Daniel Miller | 2/13/18 10:53 AM |
|---|---|---|

section 2 of (Miller et al., 2016).

**Page 9: [5] Deleted**  **Daniel Miller**  **2/13/18 1:11 PM**

[revised manuscript text omitted]

1) Cloud vertical profile: In the operational retrievals, both bispectral and polarimetric techniques assume vertically homogenous clouds. However, clouds in reality often have significant vertical variability resulting from various processes (e.g., condensational growth, coalescence, sedimentation, entrainment). Deviations from the assumed profile gives rise to many questions. For example, how do we interpret the $r_e$ and $v_e$ retrievals based on the homogenous cloud assumption? To
* * *
1 Note that throughout this paper, we will refer to "linearly polarized reflectances" simply as "polarized reflectances" in recognition of the negligible contribution of circularly polarized light in the atmosphere (Hansen, 2010).

Daniel Miller 2/14/2018 10:21 AM

Daniel Miller 2/14/2018 10:21 AM

Daniel Miller 3/4/2018 4:53 PM

Daniel Miller 3/4/2018 4:56 PM

Daniel Miller 3/4/2018 4:57 PM

Daniel Miller 3/4/2018 4:53 PM

Daniel Miller 2/26/2018 1:23 PM

Daniel Miller 2/26/2018 1:23 PM

Daniel Miller 3/5/2018 10:18 AM

what extent does cloud vertical profile influence the bispectral and polarimetric techniques? Note that Platnick (2000) developed a method utilizing the so-called "vertical weighting function" to interpret the $r_e$ retrieval from the bispectral method for clouds with vertically varying $r_e$ profile. Recently, Alexandrov et al., (2012b) took an approach that focused on the vertical weighting of the droplet size distribution to interpret the $r_e$ and $v_e$ retrievals from the polarimetric technique.

5    Miller et al. (2016) demonstrated the usefulness of this vertical weighting approach for understanding both bispectral and polarimetric $r_e$ retrievals. In Section 4.1, we will apply the vertical weighting function method to both techniques on the basis of the LES cloud fields, to help understand if cloud vertical structure could lead to significant differences between the two techniques.

2) Sensitivity to observational uncertainty: The uncertainties associated with observations of total and polarized

10    reflectances can differ, indicating that uncertainty may also impact bispectral and polarimetric retrievals differently. Additionally, the two retrievals rely on different number of uncertain observations; a pair of uncertain total reflectances (bispectral) as compared to numerous uncertain polarized reflectances (polarimetric). Furthermore the different algorithmic approaches, two-dimensional interpolation vs. nonlinear optimal curve fitting introduce additional layers of complexity in terms of the impact of uncertainty. The impact of uncertainty on retrieval results for each method are highlighted and

15    explored in section 4.2.

3) Reduced sensitivity: It can be clearly seen from Figure 1 (a) that when clouds are optically thin ($\tau$<3), the LUT for the bispectral retrieval becomes less orthogonal and the isolines of $r_e$ become more densely packed. This reduction in sensitivity can lead to significant retrieval uncertainties in bispectral techniques for optically thin clouds ($\tau$<3). Similarly, the sensitivity of the polarimetric technique to $r_e$ and $v_e$ is reduced when DSD becomes very broad (i.e., $v_e$>0.15), in which case

20    the supernumerary bow features are barely distinguishable (Figure 1 (c)). In Section 4.3 we will investigate the impacts of the reduction of sensitivity on retrieval consistency between the two techniques.

4) Sub-pixel inhomogeneity: The impact of spatial resolution and unresolved sub-pixel cloud inhomogeneity on bispectral retrievals has been well studied (Zhang and Platnick, 2011; Zhang et al., 2012; 2016). An important conclusion from these studies is that the so-called plane-parallel homogenous bias (PPHB) can cause the bispectral technique to

25    significantly overestimate $r_e$. In contrast, the sensitivity of the polarimetric retrieval to unresolved sub-pixel inhomogeneity and resolution has not been thoroughly studied. In Section 4.4, we will compare the impacts of sub-pixel inhomogeneity on bispectral and polarimetric techniques, and investigate whether it can cause deviation between the two techniques.

5) Angular resolution and sampling for polarimetric technique: In addition to spatial resolution, angular resolution and sampling is also important for the polarimetric technique. A coarse angular resolution may not be able to resolve the

30    feature of the supernumerary bows. Similarly, if the scattering angles corresponding to the supernumerary bows are not or only partly sampled, then the polarimetric technique may not have enough information content for retrieval. This issue will be discussed in Section 4.5.

Daniel Miller 2/26/2018 1:23 PM

Daniel Miller 3/4/2018 4:58 PM

Daniel Miller 2/13/2018 10:40 AM

Daniel Miller 2/13/2018 10:17 AM

Daniel Miller 2/13/2018 10:40 AM

Daniel Miller 2/13/2018 10:17 AM

Daniel Miller 2/13/2018 10:18 AM

Daniel Miller 2/13/2018 10:31 AM

Daniel Miller 2/13/2018 10:17 AM

[revised manuscript text omitted]

Daniel Miller 2/12/2018 11:29 AM

**Comment [1]:** Figure updated to correct error in original axes label of panel (a).

**Figure 1: Demonstrations of the microphysical sensitivity of the bispectral and the polarimetric techniques. Panel (a) features the bispectral LUT exhibiting sensitivity to $r_e$ (colored iso-lines), due to absorption in the SWIR/ reflectance. The VNIR reflectances provide sensitivity to optical thickness (gray iso-lines). Panels (b) and (c) demonstrate the sensitivity of polarimetric technique to $r_e$**
5 **and $v_e$ respectively. The supernumerary bow peaks of the polarized phase function ($-P_{12}$) shift and become narrower with increasing droplet size ($r_e$), whereas the peaks erode in magnitude for broadened droplet size distributions ($v_e$).**

[Figure]

**Figure 2:** The optical and microphysical properties ($\tau$, $r_e$, and $v_e$) of the LES cases examined in this study. The panels are arranged such that each LES case appears row-wise and the different properties are appear column-wise. Note that the vertically weighting functions used for the displayed re(VW) and ve(VW) correspond to single-scattering assumptions. Cloud-free masking in each of the images appears in gray. Refer to sections 2 and 3 for discussion and definition of each of these properties. Axes labels have been removed to enlarge each map, but the spatial dimensions of each scene are roughly 7 x 7 km (refer to section 3 for the specific resolutions of each LES case.)

[Figure]

Daniel Miller 2/25/2018 4:25 PM
**Comment [2]:** Figure updated to include data from the new vertical weighting defintions.

**Figure 3: Joint histogram regressions of $r_e$ and $\tau$ in all LES cases comparing the bispectral retrievals to the LES cloud microphysical properties. Panels (a) and (b) are regressions of the bispectral $r_e$(2.13 $\mu$m) and $r_e$(3.75 $\mu$m) retrievals against the physical analogue $r_e$(VW). Panels (c) and (d) are regressions of the bispectral $\tau$(2.13 $\mu$m) and $\tau$(3.75 $\mu$m) retrievals against the physical $\tau$(LES). Panels (e) and (f) display the regression of the bispectral $r_e$(2.13 $\mu$m) and $r_e$(3.75 $\mu$m) retrievals for only optically thick pixels ($\tau$>3). Note that in each panel the correlation is quantified with a linear correlation coefficient (R) and the black and white contours encompass 66% and 95% of the population, respectively.**

Daniel Miller 2/13/2018 2:35 PM

Daniel Miller 2/13/2018 1:38 PM

[Figure]

Daniel Miller 3/2/2018 11:31 AM
**Comment [3]:** Figure updated to include data from the new vertical weighting defintions.

[revised manuscript text omitted]

Daniel Miller 2/14/2018 11:45 AM

---

## Author Response (AR2)

**Author Response 2**

**Anonymous Referee #1**
Submitted on 15 March, 2018

**General remarks:**
*The authors did a good job in answering the reviewers questions and addressing necessary changes. There is only one issue I'm still struggling with. The new vertical weighting functions considering multiple scattering are a good improvement for the study. However, I still have doubts that this approach does provide the correct solution which is needed to quantify the retrieval biases, which is the main purpose of the manuscript.*

**Major comments:**
**1. Vertical Weighting**

*To find the suitable factors $b$, the authors did compare the biases using different values of $b$ and finally chose the value which provides the best agreement between retrieval results and the vertical weighted values. In my understanding this is aiming for "moving target". You can not adjust the weighting functions in a way that the correct results are obtained. Doing so, you already rule out any source of uncertainty related to the vertical structure of the cloud.*

*Additionally, the authors conclude that the new 2WT weighting function using $b=9$ is similar to the old 2WT weighting function ($b=0$). Can this be explain to be the general case? I don't think this is a general rule. It might only be coincidental due to the profile of $R_{\rm e}$ applied here.*
*(By the way, the link to the histograms did not in the PDF.)*

    **a.** The issue of comparing to a "moving target" is of great concern to us, and it was one of the reasons why we focused on a fixed vertical weighting function initially. Our approach to address this issue in this stage of the revision will be to validate our parametric vertical weighting functions more rigorously against the numerically calculated vertical weighting functions from Platnick et al. (2000).

    **b.** Apologies for the broken links in the previous response. Here are the links to the gifs, despite the fact that they are now out of date compared to the current modeling (see response to #3):

$r_e$(2.13)-$r_e$(VW,2.13) Histogram varying $b$
    *https://drive.google.com/open?id=1hKfcrhkYj3fiII3RcR-YHmvkPOo6gRXK*
$r_e$(2.13)-$r_e$(VW,2.13) Histogram varying $b$ (FOR $\tau$>5 ONLY!)
    https://drive.google.com/open?id=18Pnc_AgPK3Y_prf7eYzKnk4b15jL-bAI
$r_e$(3.75)-$r_e$(VW,3.75) histogram varying $b$
    *https://drive.google.com/open?id=1HHQhDvyVww2iqlS7iAnhsiZqhM5DMb7K*

$r_e$(3.75,coupled to $v_e$(pol))-$r_e$(VW,3.75) Histogram varying $b$
https://drive.google.com/open?id=1Sj68TIpGEnziOGS8MWU_FJP4HdAndKnT

2. **Vertical weighting functions**

   *The treatment of clouds with $\tau<5$ is arbitrary in my point of view. The weighting function can not simply be changed just because the bias becomes to large. You need to give a reason why the vertical weighting function of clouds smaller than $\tau<5$ is not correct. An I do not see a reason why the concept of the multiple scattering weighting function does not hold for lower tau. Did you normalize the weighting function appropriately?*

   a. This restriction is only for determining the optimal value of b in the "tuned/moving target approach" mentioned in the previous author response. Now that the approach has changed this is not so relevant, but the initial reasoning was that the peak of the b=9 vertical weighting function defined previously was located at a depth of tau~5, leading to results that weighted the bottom more than the top of the cloud in instances when the cloud wasn't thicker than that. This is obviously incorrect and something that we no longer deal with for the new parametric vertical weighting defined below.

3. **Vertical weighting functions**

   *The authors argue, that their new weighting function considers the inhomogeneous cloud vertical structure compared to the approach by Platnick (2000). Here I disagree. Of course the simulations in the Platnick (2000) approach to calculate the weighting function assumes homogeneous clouds. But the application of the weighting functions can be done for real cloud profiles. The 2WT weighting used by the authors does the same. The 2WT weighting function is also calculated only as a function of $\tau$. Therefore, also here no profile of $R_{\rm e}$ is considered in the calculation. Only when the 2WT weighting is applied to the cloud profile the vertical structure is considered. This is similar to Platnick (2000) and in my point of view no reason to prefer the 2WT weighting functions.*

   *I see, that it would be a big effort to completely change the weighting functions to Platnick (2000). In order not to need to do this, the authors should convincingly show, that their weighting functions are similar to the Platnick (2000) approach and do not cause any bias, that may affect the study. Such a comparison for only one or some representative cloud profiles could also be used to justify the values of $b$.*

**Inhomogeneity Discussion:**

Perhaps we can clarify this in the text, but the assertion that we're making is not that we consider inhomogeneous vertical structure in a way that the Platnick approach doesn't. It is instead that our vertical weighting function

doesn't make any assumptions about the form of the microphysical distribution of cloud droplets. The vertical weighting function defined in eq. 4 of Platnick (2000) explicitly weights the vertical profile of $r_e(z)$ (calculated layer by layer). This approach results in a decoupling in the relationship between $r_e(z)$ and $v_e(z)$. Whereas, if one vertically weights the droplet size distribution, $N(r,z)$, they preserve the relationship between size distribution parameters in each layer by calculating $r_e$(VW) & $v_e$(VW) for the same effective distribution.

**Summary of Differences:**

To summarize points made here and previously more succinctly, the main differences between these approaches are:

1. Mathematical formulation:  Vertically weighting $r_e$(z) or $N$(r,z).
    a. To this end, we will examine in this response how the Platnick vertical weighting function behaves when it is used to vertically weight $N(r,z)$ in the same way as our approach.
    b. This is getting deep into the weeds, but the mathematical difference between these approaches is similar to the spatial averaging discussion in the appendix of the manuscript – with the caveat that vertical inhomogeneity is greater than horizontal inhomogeneity.
2. Derivation:  Fixed Parametric vs. Direct Radiative Modeling
    a. The fixed parametric approach does not consider the radiative impact of the $r_e$(z) profile itself on the definition of the vertical weighting function. The validity of the parametric approach would indeed depend on how important this assumption is.
    b. Direct modeling takes time, and a fixed parametric model might get reasonable results with limited effort.

You do raise some very important points regarding accounting for the impact of dependency on $r_e$ profile, and I think this is one of the core shortcomings of using a fixed empirical function for the entire LES domain. To make sure to emphasize these shortcomings of the fixed parametric vertical weighting function **I've added some additional discussion of this on page 9 of the text.** But it must also be emphasized that the fixed empirical functions are easier to derive, easier to use, and applicable in situations with limited cloud profile information. There are many practical reasons for developing a single parametric function that usefully replicates the behavior of the maximum penetration depth vertical weighting defined in eq. 4 of Platnick et al. (2000). In our case, while it is computationally expensive, it is possible to calculate this weighting function for the entire LES domain, but in other applications a fixed parametric weighting function is the only satisfactory way to describe the impact of vertical penetration on retrieved cloud properties (for example, Zhang et al. (2017)).

**Radiatively Derived Vertical Weighting**

The radiatively derived vertical weighting functions are calculated following the maximum penetration depth weighting function (eq. 4 of Platnick (2000)). Focusing on a single horizontal transect through the DYCOMS-II LES case, we analyzed the behavior of 128 vertical profiles. The figure below depicts vertical weighting functions for a single column within the transect for different solar geometries. These radiatively derived weighting functions will be used to calculate $r_e$(VW) from both the $r_e(z)$ and $N(r,z)$ profile weighting approaches.

[Figure]

Figure 1: Radiatively derived vertical weighting functions for a nadir view of a single column of the DYCOMS-II LES case. The top left, right, and bottom right panels show total reflectance vertical weighting functions for the 0.865, 2.13, and 3.75 μm bands respectively. The bottom left panel shows a polarized reflectance vertical weighting function (derived using Rp=sqrt(Q^2+U^2)) for the 0.865 μm band. Note that proper accounting for polarimetric vertical weighting should also account for varying viewing geometry. Each figure also includes three curves for different solar zenith angle (in degrees) geometries. Note that for nadir viewing the SZA=40 and RAA=30 case corresponds to a scattering angle of 140, which is close to the cloudbow peak.

**A New Parametric Model:**

We have again modified the parametric vertical weighting method used in this study. This change was made in part because we found that the single parameter weighting wasn't adequately representing the radiatively derived vertical weighting functions for the 2.13 μm band. The new two-parameter vertical weighting function is,

$$W(\tau) = c\tau^b \exp\left[-a\tau\left(\frac{1}{\mu} + \frac{1}{\mu_0}\right)\right],$$

"where the parameters $a$ and $b$ are introduced to account for the influence of multiple scattering effects to the vertical weighting function discussed previously in Miller et al. (2016). The parameter $a$ scales the optical depth, modelling the enhancement of transmission caused by multiple scattering whereas the parameter $b$ produces a peaked vertical weighting function associated with the expected penetration depth of the reflected light and $c$ is the normalization factor. Each of these parameters is strictly positive, and for $a$=1 and $b$=0 we obtain the original single-scattering vertical weighting used in Miller et al. (2016). For smaller values of $a$ and larger values of $b$, the vertical weighting function extends deeper into the cloud, leading to droplet size distribution properties deeper in the cloud contributing more information to the vertically weighted value. A single value of $a$ and $b$ are is selected for each retrieval approach based on coefficients that gave best fits to a small number of numerically derived vertical weighting functions based on eq. 4 of Platnick (2000)." - (taken directly from the updated manuscript)

As a brief aside, it should be noted that this new parametric function has a form similar to a gamma distribution with respect to tau.

The radiatively modeled and the parametric weighting functions can now be used to search for the "best fit" parameters, $a$ and $b$, suitable for each LES column and in different spectral bands. The results for of these fits are shown in Figure 2. Below are links to animated gifs of the best-fit solutions found throughout the transect for SZA=20 and nadir viewing:

2.13 μm Vertical Weighting Function
https://drive.google.com/open?id=1wZxnmI2xdS3X4JTFxv-oKFFHteyn9hQT
3.75 μm Vertical Weighting Function
https://drive.google.com/open?id=14j2gyv0zZVC1OtBCkPQzktREcnD9Bfyp

The predominance of high quality fits ($R^2$>0.99) indicates that this functional form is quite reliable. The best-fit $a$ and $b$ for each of the bands are also well clustered, indicating that a selection of a single value of $a$ and $b$ for each wavelength (and observation geometry) should sufficiently represent the radiatively modeled vertical weighting functions. Throughout the rest of this work, we make use of the scene aggregated median values of $a$ and $b$ from

these plots (limited to $R^2>0.99$). The fitting approach also allows us to verify whether the parametric function has any correlation to cloud property variability like the $r_e$ profile or the total cloud optical thickness. To that end, we observed that there was no clear correlation between the best-fit values of $a$ and $b$ and the retrieval bias, $r_e$(VW)-$r_e$(bispectral), or the cloud optical thickness.

[Figure]

**Figure 2: Best-fit parameters from fitting radiatively derived vertical weighting functions. The top panel depicts $R^2$, the middle $a$, and the bottom panel $b$. Parameters are only plotted for higher quality fits with $R^2>0.99$. The case examined here corresponds to VZA=0, SZA=20, and RAA=30.**

With both the parametric and radiatively derived weighting functions we can now derive $r_e$(VW) and ve using both and compare them. and bispectral (NJK) retrievals. It should also be noted that there is a persistent high bias in the $r_e$(3.7μm) retrieval that longer exists if the retrieval is coupled to $v_e$(pol) as previously discussed in the manuscript.

[Figure]

**Figure 3: Comparisons of different approaches for obtaining vertically weighted cloud properties using a single horizontal transect of the DYCOMS-II LES case. The top and bottom rows compare vertical weighting results for the 2.13 and 3.75 µm bands respectively. The RT-model-derived Platnick vertical weighting $r_e(z)$ (blue) is compared to the Parametric vertical weighting of $N(r,z)$ (red) and corresponding bispectral (NJK) retrieval is also shown(purple). The case examined here corresponds to VZA=0, SZA=20, and RAA=30.**

Additionally we can test the difference between vertical weighting of $r_e(z)$ and $N(r,z)$. A quick comparison of Figure 3 and Figure 4 reveals that the most significant source of difference between the two vertically weighted results is the choice of which vertical profile is being weighted. This choice has a weak impact on $r_e$(VW), but a more significant impact on the overall shape of the inferred distribution because $v_e$(VW) changes (for 2.13 µm in particular).

[Figure]

**Figure 4: Similar to Figure 4, except the differences between Platnick vertical weighting (red) and the parametric vertical weighting (blue) are shown for both weighting of the profile of $r_e(z)$ (solid) and $N(z)$ (dashed). The left panels are for $r_e$ in the 2.13 and 3.75 μm bands, while the right panels show $v_e$ in the 2.13 and 3.75 μm bands. The case examined here corresponds to VZA=0, SZA=20, and RAA=30.**

All results shown here have focused on a single solar and viewing geometry but, as indicated in Figure 1, the vertical weighting function is clearly dependent on observation geometry. The parameters of the parametric function should depend on observation geometry, and indeed they do, as shown in the figure 5.

The scattering angle dependence of these results is prominent. For the purposes of the vertical weighting functions used in our study, we have a table of values for $a$ and $b$ that accounts for spectral band, viewing zenith angle, solar zenith angle, and relative azimuth angle.

[Figure]

**Figure 5: The median (scene aggregated) best-fit solutions for *a* and *b* plotted against scattering angle. The left and right columns examine parameters of the 2.13 and 3.75 μm weighting functions respectively. The top row corresponds to parameter *a*, while the bottom row corresponds to parameter *b*. Additionally, results for three solar zenith angles are shown: SZA=20º (blue), SZA=40º (red), and SZA=60º (yellow).**

**Updated Comparison of Retrievals and VW:**

After all of this has been addressed, we can now again return to the comparison of the full LES vertically weighted dataset to the bispectral retrievals. The results of this comparison are shown in Figure 6. Broadly speaking these results are similar to those discussed in previous iterations of the vertical weighting. This is an indication that the particular vertical weighting definition is not the primary source of differences in these comparisons.

There are two modes in the joint histogram of the $r_e$(3.75 μm) histogram. The population with high biased $r_e$(3.75 μm) retrievals relative to VW belongs to the DYCOMS-II LES case and is associated with the impact of $v_e$ on the $r_e$(3.75 μm) retrieval. The When $v_e$(pol) information is provided for the 3.75 μm bispectral retrieval – the vertically weighted result compares very well (as we showed previously for the transect in Figure 3). It should be noted that this feature was present, to a lesser extent, in previous vertical weighting iterations.

[Figure]

Figure 6: Joint histogram comparisons of bispectral retrievals of $r_e$ (2.13 μm left column and 3.75 μm right column) against new vertically weighted $r_e$(VW). The top row is the full population, while the bottom row is restricted to the optically thick population and retrievals coupled to the polarimetric $v_e$ retrieval.

**Anonymous Referee #2**
Submitted on 12 March, 2018

**General remarks:**
*The authors responded sufficiently to most of my comments. However, the first of my minor comments is not addressed. For completeness, I repeat it here, with some updates and some elaboration:*

**Minor comments:**
1. *In figure 4 (and 6), the cases with tau<3 are removed, revealing a better result. However, I am wondering how the 2D histogram for tau<3 looks like. Do the bulk of these retrievals still perform well, or are they all biased? That is not clear from these plots. It is obvious that the outliers are from the cases with tau<3, but the bulk of the results with tau<3 could still lay on the 1:1 line. Also, are the results getting increasingly 'bad' for decreasing tau? I am not asking for an extra figure in the paper, but some brief discussion about this would be good.*
   a. Apologies for missing this during the first response. Each histogram below is normalized with respect to the full population to demonstrate the reduced population of data. The results are shown in Figures 7-10 for different combinations of variables that were sub-selected throughout the study. **The short answer to your question is, no not all of the retrievals are biased, but nearly all of the extreme outliers belong to populations with very small optical thicknesses.** Additionally, less and less data lies near the 1:1 line with decreasing optical thickness, while the population of extreme outliers typically stays similar. Note that the figures here only show results for the 2.13 μm bispectral retrieval, but the results are similar for the 3.75 μm retrieval as well. We will add some discussion of how not all thin cloud retrievals belong to these extreme outlier populations. (Added on line 32 of page 11).

[Figure]

**Figure 7: Joint histogram comparison of $r_e$(2.13 μm) against $r_e$(VW) for subpopulations of the full dataset. The left panel subselects tau<=3, the middle panel tau <=1, and the right panel tau<=0.5. Each histogram is normalized to the overall population.**

[Figure]

**Figure 8: Joint histogram comparison of $r_e$(pol) against $r_e$(VW) for subpopulations of the full dataset. The left panel subselects tau<=3, the middle panel tau <=1, and the right panel tau<=0.5. Each histogram is normalized to the overall population.**

[Figure]

**Figure 9: Joint histogram comparison of $v_e$(pol) against $v_e$(VW) for subpopulations of the full dataset. The left panel subselects tau<=3, the middle panel tau <=1, and the right panel tau<=0.5. Each histogram is normalized to the overall population.**

[Figure]

**Figure 10:: Joint histogram comparison of $r_e$(2.13 μm) against $r_e$(pol) for subpopulations of the full dataset. The left panel subselects tau<=3, the middle panel tau <=1, and the right panel tau<=0.5. Each histogram is normalized to the overall population.**

2. *One minor correction: On Page 6, line 27 it says*
   *"In this study, we focus on three major sources of retrieval uncertainty for both techniques"*
   *The number is now five instead of three.*
   a. Fixed it. Thanks for catching that.

**Main document changes and comments**

| Page 1: Inserted | Daniel Miller | 1/29/18 10:43 AM |
|---|---|---|

retrievals of marine boundary layer

| Page 1: Inserted | Daniel Miller | 1/29/18 10:43 AM |
|---|---|---|

s

| Page 1: Deleted | Daniel Miller | 1/29/18 10:43 AM |
|---|---|---|

al

| Page 1: Inserted | Daniel Miller | 1/29/18 10:43 AM |
|---|---|---|

: Case studies

| Page 1: Deleted | Daniel Miller | 1/29/18 10:43 AM |
|---|---|---|

retrievals

| Page 1: Inserted | Daniel Miller | 1/29/18 10:43 AM |
|---|---|---|

a

| Page 1: Inserted | Daniel Miller | 1/29/18 10:44 AM |
|---|---|---|

s

| Page 1: Deleted | Daniel Miller | 1/29/18 10:44 AM |
|---|---|---|

S

| Page 1: Inserted | Daniel Miller | 3/4/18 5:03 PM |
|---|---|---|

,3

| Page 1: Deleted | Daniel Miller | 3/4/18 5:04 PM |
|---|---|---|

DJ-Miller@umbc.edu

| Page 1: Inserted | Daniel Miller | 3/4/18 5:04 PM |
|---|---|---|

Daniel.J.Miller@nasa.gov

| Page 1: Inserted | Daniel Miller | 3/4/18 5:05 PM |
|---|---|---|

a 1-D

| Page 1: Deleted | Daniel Miller | 3/4/18 5:05 PM |
|---|---|---|

s

| Page 1: Inserted | Daniel Miller | 3/4/18 4:34 PM |
|---|---|---|

h

| Page 1: Deleted | Daniel Miller | 3/4/18 4:34 PM |
|---|---|---|

indeed h

| Page 1: Inserted | Daniel Miller | 3/4/18 9:19 PM |
|---|---|---|

within expected observational uncertainties

| Page 1: Inserted | Daniel Miller | 3/4/18 9:20 PM |
|---|---|---|

The relatively small systematic biases

| Page 1: Deleted | Daniel Miller | 3/4/18 9:20 PM |
|---|---|---|

differences

| Page 2: Inserted | Daniel Miller | 2/13/18 2:53 PM |
|---|---|---|

which

| Page 2: Inserted | Steven Platnick | 2/28/18 4:27 PM |
|---|---|---|

typically

| Page 2: Inserted | Steven Platnick | 2/28/18 10:19 AM |
|---|---|---|

or midwave infared (MWIR)

| Page 2: Deleted | Daniel Miller | 3/2/18 3:04 PM |
|---|---|---|

2016

| Page 2: Inserted | Daniel Miller | 3/2/18 3:04 PM |
|---|---|---|

2017

| Page 2: Inserted | Daniel Miller | 1/23/18 4:16 PM |
|---|---|---|

Suomi

| Page 2: Inserted | Daniel Miller | 1/23/18 4:16 PM |
|---|---|---|

Suomi

| Page 2: Deleted | Steven Platnick | 2/28/18 4:26 PM |
|---|---|---|

vastly

| Page 2: Inserted | Steven Platnick | 2/28/18 4:26 PM |
|---|---|---|

fundamentally

| Page 3: Deleted | Daniel Miller | 1/23/18 4:17 PM |
|---|---|---|

Because

| Page 3: Inserted | Daniel Miller | 1/23/18 4:17 PM |
|---|---|---|

The

| Page 3: Deleted | Daniel Miller | 3/4/18 11:41 AM |
|---|---|---|

,

| Page 3: Inserted | Daniel Miller | 3/4/18 11:41 AM |
|---|---|---|

. It is therefore essential

| Page 3: Deleted | Daniel Miller | 1/23/18 4:18 PM |
|---|---|---|

it is important

| Page 3: Deleted | Daniel Miller | 1/23/18 4:18 PM |
|---|---|---|

the

| Page 3: Inserted | Daniel Miller | 1/23/18 4:21 PM |

between the two techniques

| Page 3: Deleted | Daniel Miller | 1/23/18 4:18 PM |

between them

| Page 3: Deleted | Daniel Miller | 1/23/18 4:21 PM |

the

| Page 3: Inserted | Daniel Miller | 1/23/18 4:21 PM |

their respective

| Page 3: Deleted | Daniel Miller | 1/23/18 4:21 PM |

 of each technique

| Page 3: Inserted | Daniel Miller | 2/26/18 1:23 PM |

Bréon and Doutriaux-Boucher (2005)

| Page 3: Deleted | Daniel Miller | 2/26/18 1:23 PM |

Bréon and Doutriaux-Boucher (2005)

| Page 3: Inserted | Steven Platnick | 2/28/18 4:36 PM |

2 μm

| Page 3: Inserted | Steven Platnick | 2/28/18 4:34 PM |

150 km scale

| Page 3: Deleted | Steven Platnick | 2/28/18 4:36 PM |

by about 2 μm

| Page 3: Inserted | Daniel Miller | 2/26/18 1:23 PM |

Bréon and Doutriaux-Boucher (2005)

| Page 3: Deleted | Daniel Miller | 2/26/18 1:23 PM |

Bréon and Doutriaux-Boucher (2005)

| Page 3: Inserted | Daniel Miller | 2/25/18 10:10 PM |

how well

| Page 3: Deleted | Daniel Miller | 2/25/18 10:10 PM |

whether

| Page 3: Deleted | Daniel Miller | 2/25/18 10:10 PM |

well

| Page 3: Inserted | Daniel Miller | 2/25/18 10:10 PM |

and what situations might cause them to differ

| Page 3: Deleted | Daniel Miller | 2/25/18 10:10 PM |

or not

| Page 3: Inserted | Daniel Miller | 2/25/18 10:11 PM |
| --- | --- | --- |

and

| Page 3: Deleted | Daniel Miller | 2/25/18 10:11 PM |
| --- | --- | --- |

that

| Page 3: Deleted | Daniel Miller | 2/25/18 10:11 PM |
| --- | --- | --- |

e

| Page 3: Inserted | Daniel Miller | 2/25/18 10:11 PM |
| --- | --- | --- |

ing

| Page 3: Inserted | Daniel Miller | 3/2/18 3:54 PM |
| --- | --- | --- |

Zinner et al., 2010

| Page 3: Deleted | Daniel Miller | 3/2/18 3:55 PM |
| --- | --- | --- |

Miller et al., 2016

| Page 3: Inserted | Daniel Miller | 3/2/18 3:55 PM |
| --- | --- | --- |

; Miller et al., 2016

| Page 3: Deleted | Daniel Miller | 3/2/18 3:55 PM |
| --- | --- | --- |

; Zinner et al., 2010

[revised manuscript text omitted]

,

| Page 7: Deleted | Daniel Miller | 2/13/18 10:18 AM |

| Page 7: Inserted | Daniel Miller | 2/13/18 10:18 AM |

| Page 7: Deleted | Daniel Miller | 2/13/18 10:31 AM |

| Page 7: Deleted | Daniel Miller | 2/13/18 10:17 AM |

| Page 7: Inserted | Daniel Miller | 2/13/18 10:17 AM |

| Page 8: Deleted | Daniel Miller | 3/4/18 4:26 PM |

; Miller et al., 2016

| Page 8: Inserted | Daniel Miller | 3/4/18 4:26 PM |

; Miller et al., 2016

| Page 8: Inserted | Daniel Miller | 2/8/18 2:19 PM |
|---|---|---|

a

| Page 8: Deleted | Daniel Miller | 2/8/18 2:19 PM |
|---|---|---|

the

| Page 8: Inserted | Daniel Miller | 2/8/18 2:20 PM |
|---|---|---|

| Page 8: Deleted | Daniel Miller | 2/8/18 2:20 PM |
|---|---|---|

ly fitting

| Page 8: Inserted | Daniel Miller | 2/8/18 2:20 PM |
|---|---|---|

is then used to identify the

| Page 9: Deleted | Daniel Miller | 2/8/18 2:20 PM |
|---|---|---|

corresponds

| Page 9: Inserted | Daniel Miller | 2/8/18 2:20 PM |
|---|---|---|

corresponding

| Page 9: Deleted | Daniel Miller | 2/8/18 2:20 PM |
|---|---|---|

to the resulting

| Page 9: Deleted | Daniel Miller | 2/13/18 10:50 AM |
|---|---|---|

2WT

| Page 9: Inserted | Daniel Miller | 2/13/18 10:50 AM |
|---|---|---|

VW

| Page 9: Deleted | Daniel Miller | 2/13/18 10:50 AM |
|---|---|---|

2WT

| Page 9: Inserted | Daniel Miller | 2/13/18 10:50 AM |
|---|---|---|

VW

| Page 9: Inserted | Daniel Miller | 3/4/18 12:30 PM |
|---|---|---|

weighted (VW)

| Page 9: Inserted | Daniel Miller | 2/13/18 10:50 AM |
|---|---|---|

vertical

| Page 9: Deleted | Daniel Miller | 3/4/18 12:30 PM |
|---|---|---|

integration is weighte

| Page 9: Inserted | Daniel Miller | 3/4/18 12:30 PM |
|---|---|---|

integration is weighted

| Page 9: Deleted | Daniel Miller | 3/4/18 12:30 PM |
|---|---|---|

d

| Page 9: Deleted | Daniel Miller | 2/13/18 10:49 AM |

two-way transmittance (2WT)

| Page 9: Inserted | Daniel Miller | 2/13/18 10:51 AM |

and multiple scattering

| Page 9: Deleted | Daniel Miller | 2/13/18 10:49 AM |

the

| Page 9: Deleted | Daniel Miller | 2/13/18 10:49 AM |

single-scattered

| Page 9: Inserted | Daniel Miller | 2/13/18 10:50 AM |

in the corresponding wavelength associated with each particular retrieval.

| Page 9: Moved from page 9 (Move #1) | Daniel Miller | 2/13/18 10:52 AM |

Thus, $r_e$(VW) and $v_e$(VW) should be directly comparable to the numerically retrieved $r_e$ and $v_e$ from the simulated reflectance (Alexandrov et al., 20152b; Miller et al., 2016; Platnick, 2000; Zhang et al., 2017).

| Page 9: Inserted | Daniel Miller | 2/13/18 10:53 AM |

$r_e$(VW) and $v_e$(VW) should be

| Page 9: Deleted | Daniel Miller | 3/4/18 12:49 PM |

directly

| Page 9: Deleted | Daniel Miller | 2/26/18 6:37 PM |

numerically

| Page 9: Deleted | Daniel Miller | 2/27/18 10:35 AM |

| Page 9: Inserted | Daniel Miller | 2/27/18 10:35 AM |

2b

| Page 9: Deleted | Daniel Miller | 3/4/18 12:51 PM |

Platnick, 2000;

| Page 9: Inserted | Daniel Miller | 3/4/18 12:50 PM |

| Page 9: Inserted | Daniel Miller | 2/13/18 10:51 AM |

The method of vertical weighting in this study is described in detail in

| Page 9: Moved from page 9 (Move #2) | Daniel Miller | 2/13/18 10:53 AM |

section 2 of Miller et al. (2016), however in this study we have modified the vertical weighting function to account for multiple scattering. Motivated by the convenience and flexibility of the parametric approach proposed in eq. 4 of Zhang et al. (2017), we implement a two-variable parametric vertical weighting function:

$$W(\tau) = c\tau^b \exp\left[-a\tau\left(\frac{1}{\mu} + \frac{1}{\mu_0}\right)\right],$$ (4)

| Page 9: Inserted | Daniel Miller | 2/13/18 10:54 AM |

Miller et al. (2016), however in this study we have modified the vertical weighting function to account for multiple scattering. Motivated by the convenience and flexibility of the parametric approach proposed in eq. 4 of Zhang et al. (2017), we implement a two-variable parametric vertical weighting function:

| Page 9: Inserted | Daniel Miller | 2/13/18 11:00 AM |

$$W(\tau) = c\tau^b \exp\left[-a\tau\left(\frac{1}{\mu} + \frac{1}{\mu_0}\right)\right]$$

| Page 9: Inserted | Daniel Miller | 2/13/18 11:00 AM |

,

| Page 9: Inserted | Daniel Miller | 2/13/18 10:49 AM |

where the new parameters $a$ and $b$ are introduced to account for the influence of multiple scattering effects not originally considered in the vertical weighting function discussed previously in Miller et al. (2016). The parameter $a$ scales the optical depth, modelling the enhanced transmission caused by multiple scattering whereas the parameter $b$ produces a peaked vertical weighting function associated with the expected penetration depth of the reflected light and $c$ is the normalization factor. Each of these parameters is strictly positive, and for $a=1$ and $b=0$ we obtain the original single scattering vertical weighting used in Miller et al. (2016). For smaller values of $a$ and larger values of $b$, the vertical weighting function extends deeper into the cloud, leading to droplet size distribution properties deeper in the cloud contributing more information to the vertically weighted value. For the polarimetric retrieval, $a=1$ and $b=0$ were selected due to the dominance of single scattering in polarized reflectances. In contrast, multiple scattering can significantly impact total reflectances. For total reflectances a single value of $a$ and $b$ was selected for each spectral band and observation geometry based on coefficients that fit best to numerically calculated vertical weighting functions based on the method presented in eq. 4 of Platnick (2000).[1] Generally, we found that $a(3.75\ \mu m)$ was larger than $a(2.13\ \mu m)$, as would be expected because of stronger absorption in 3.75 μm reducing transmission into the cloud. We also found that $b$ was dependent on observation geometry (scattering angle) and $b(3.75\ \mu m)$ was less than $b(2.13\ \mu m)$ because multiply scattered light in the 2.13 μm band can penetrate deeper into the cloud before scattering back out.

| Page 9: Deleted | Daniel Miller | 2/13/18 11:11 AM |
* * *
[1] The radiatively derived vertical weighting of Platnick (2000) implicitly depends on the $r_e(z)$ profile whereas a fixed parameter vertical weighting described here does not. However, the importance of this difference should be less than the vertical variability of optical depth or extinction cross section.

(at 0.865 µm). For a complete description on how vertical weighting is accounted for in the calculation of $r_e$(2WT) and $v_e$(2WT), see section 2 of (Miller et al., 2016). The $r_e$(2WT) and $v_e$(2WT) take into account the first-order sensitivity of the retrieval techniques to the vertical profile of clouds.

| Page 9: Moved to page 9 (Move #2) | Daniel Miller | 2/13/18 10:53 AM |
|---|---|---|

section 2 of (Miller et al., 2016).

| Page 9: Moved to page 9 (Move #1) | Daniel Miller | 2/13/18 10:52 AM |
|---|---|---|

Thus, they are directly comparable to the numerically retrieved $r_e$ and $v_e$ from the simulated reflectance (Alexandrov et al., 2015; Miller et al., 2016; Platnick, 2000; Zhang et al., 2010).

| Page 9: Deleted | Daniel Miller | 2/13/18 1:11 PM |
|---|---|---|

. We note that the 2WT vertical weighting function provides a reasonable approximation when the signal is contributed mainly by single scattering (i.e., 3.7 µm or polarimetric reflectances) but becomes less accurate for spectral bands with more multiple scattering (Platnick, 2000).

| Page 9: Deleted | Daniel Miller | 2/13/18 1:11 PM |
|---|---|---|

2WT

| Page 9: Inserted | Daniel Miller | 2/13/18 1:11 PM |
|---|---|---|

VW

| Page 9: Deleted | Daniel Miller | 2/13/18 1:11 PM |
|---|---|---|

2WT

| Page 9: Inserted | Daniel Miller | 2/13/18 1:11 PM |
|---|---|---|

VW

| Page 10: Inserted | Daniel Miller | 2/13/18 2:08 PM |
|---|---|---|

(for $\lambda$=0.865 µm)

| Page 10: Deleted | Daniel Miller | 2/13/18 1:11 PM |
|---|---|---|

2WT

| Page 10: Inserted | Daniel Miller | 2/13/18 1:11 PM |
|---|---|---|

VW

| Page 10: Deleted | Daniel Miller | 2/13/18 1:11 PM |
|---|---|---|

2WT

| Page 10: Inserted | Daniel Miller | 2/13/18 1:11 PM |
|---|---|---|

VW

| Page 10: Inserted | Daniel Miller | 2/13/18 2:06 PM |
|---|---|---|

section 4.1

| Page 10: Deleted | Daniel Miller | 2/13/18 1:11 PM |
|---|---|---|

2WT

| Page 10: Inserted | Daniel Miller | 2/13/18 1:11 PM |
|---|---|---|

VW

| Page 10: Deleted | Daniel Miller | 2/13/18 1:11 PM |
|---|---|---|
2WT

| Page 10: Inserted | Daniel Miller | 2/13/18 1:11 PM |
|---|---|---|
VW

| Page 10: Deleted | Daniel Miller | 2/13/18 1:11 PM |
|---|---|---|
2WT

| Page 10: Inserted | Daniel Miller | 2/13/18 1:11 PM |
|---|---|---|
VW

| Page 10: Deleted | Daniel Miller | 2/13/18 1:11 PM |
|---|---|---|
2WT

| Page 10: Inserted | Daniel Miller | 2/13/18 1:11 PM |
|---|---|---|
VW

| Page 10: Deleted | Daniel Miller | 2/13/18 1:11 PM |
|---|---|---|
2WT

| Page 10: Inserted | Daniel Miller | 2/13/18 1:11 PM |
|---|---|---|
VW

| Page 10: Deleted | Daniel Miller | 2/13/18 1:12 PM |
|---|---|---|
2WT

| Page 10: Inserted | Daniel Miller | 2/13/18 1:12 PM |
|---|---|---|
VW

| Page 10: Deleted | Daniel Miller | 3/4/18 12:58 PM |
|---|---|---|
that have

| Page 10: Inserted | Daniel Miller | 3/4/18 12:58 PM |
|---|---|---|
with

| Page 10: Deleted | Daniel Miller | 2/13/18 1:12 PM |
|---|---|---|
small-scale

| Page 10: Deleted | Daniel Miller | 2/13/18 1:12 PM |
|---|---|---|
tion

| Page 10: Inserted | Daniel Miller | 2/13/18 1:12 PM |
|---|---|---|
bility in the unresolved

| Page 10: Deleted | Daniel Miller | 2/27/18 9:44 AM |
|---|---|---|
 of DSD

| Page 10: Inserted | Daniel Miller | 2/27/18 9:44 AM |
|---|---|---|
 microphysics,

| Page 10: Deleted | Daniel Miller | 2/27/18 9:45 AM |

(

| Page 10: Inserted | Daniel Miller | 2/27/18 9:45 AM |

| Page 10: Deleted | Daniel Miller | 2/27/18 9:45 AM |

Shang  et al. 2015),

| Page 10: Deleted | Daniel Miller | 2/13/18 1:13 PM |

2WT

| Page 10: Inserted | Daniel Miller | 2/13/18 1:13 PM |

VW

| Page 10: Deleted | Daniel Miller | 2/13/18 1:13 PM |

2WT

| Page 10: Inserted | Daniel Miller | 2/13/18 1:13 PM |

VW

| Page 10: Inserted | Daniel Miller | 3/4/18 12:59 PM |

average

| Page 10: Deleted | Daniel Miller | 3/4/18 12:59 PM |

the

| Page 10: Inserted | Daniel Miller | 2/26/18 1:23 PM |

Stevens et al. (2005)

| Page 10: Deleted | Daniel Miller | 2/26/18 1:23 PM |

Stevens et al. (2005)

| Page 10: Inserted | Daniel Miller | 2/26/18 1:23 PM |

Ackerman et al. (2009)

| Page 10: Deleted | Daniel Miller | 2/26/18 1:23 PM |

Ackerman et al. (2009)

| Page 10: Inserted | Daniel Miller | 2/26/18 1:23 PM |

Fridlind and Ackerman (2011)

| Page 10: Deleted | Daniel Miller | 2/26/18 1:23 PM |

Fridlind and Ackerman (2011)

| Page 10: Inserted | Daniel Miller | 4/27/18 12:01 AM |

Figure 2

| Page 10: Inserted | Daniel Miller | 4/27/18 12:01 AM |

Table 1

| Page 11: Deleted | Daniel Miller | 2/13/18 9:35 AM |
|---|---|---|

at

| Page 11: Inserted | Daniel Miller | 2/13/18 9:35 AM |
|---|---|---|

for

| Page 11: Inserted | Daniel Miller | 2/13/18 2:23 PM |
|---|---|---|

pixel

| Page 11: Inserted | Daniel Miller | 4/27/18 12:01 AM |
|---|---|---|

Table 1

[revised manuscript text omitted]

2WT

| Page 17: Inserted | Daniel Miller | 2/13/18 1:38 PM |

VW

| Page 17: Inserted | Daniel Miller | 2/27/18 10:23 AM |

(Alexandrov et al., 2012b)

| Page 18: Inserted | Daniel Miller | 2/26/18 7:22 PM |

$v_e$

| Page 18: Deleted | Daniel Miller | 2/26/18 7:22 PM |

a

| Page 18: Deleted | Daniel Miller | 2/26/18 7:22 PM |

$v_e$ and varying $\tau$

| Page 18: Deleted | Daniel Miller | 2/13/18 1:38 PM |

2WT

| Page 18: Inserted | Daniel Miller | 2/13/18 1:38 PM |

VW

| Page 18: Inserted | Daniel Miller | 2/26/18 1:56 PM |

Each of these effects reduces sensitivity to the cloudbow features; and thus unresolved variability in $\tau$ and $v_e$ could influence coarse resolution retrievals. For example, Shang et al. (2015) found that unresolved spatial inhomogeneity of $\tau$ and $v_e$ increased retrieval biases in $v_e$(pol), while they were not able to discern a trend in retrieval biases in their study. However, in our case study featured in Figure 11Figure (b) we do not see a significant difference between coarse ($v_e$(pol)$_{800\ m}$) and fine scale ($<v_e$(pol)$_{50\ m}>$) retrievals, but both

retrievals are low-biased relative to the mean LES property ($<v_e(\mathrm{VW})_{50\,m}>$). This result was surprising, because both fine and coarse resolution retrievals were biased similarly. It appears as though coarse resolution retrievals arrive at the same answer as the fine scale retrievals through different processes. The average of fine scale retrievals (that are systematically biased low) and the retrieval based on the average of fine scale reflectances (which are reduced for reasons discussed above) results in a similar retrieval outcome. Unlike the bispectral retrieval, where retrievals differ from one another at different resolutions, the polarimetric retrieval seems to compare well to itself at both resolutions – even when it might be biased relative to the underlying microphysics of the physical scene. To examine this this further we

| Page 18: Deleted | Daniel Miller | 3/4/18 9:50 PM |

Figure

| Page 18: Deleted | Daniel Miller | 2/26/18 2:00 PM |

we find that these features do not systematically bias the $v_e(\mathrm{pol})$ retrieval in this case. In fact, rather surprisingly, we

| Page 18: Inserted | Daniel Miller | 3/4/18 2:44 PM |

performed

| Page 18: Deleted | Daniel Miller | 3/4/18 2:33 PM |

find that the most important bias for the coarse spatial resolution $v_e(\mathrm{pol})$ retrieval is the lack of sensitivity on $v_e(2\mathrm{WT})>0.15$, a feature that was also present for the high spatial resolution retrievals. This finding is also supported for

| Page 18: Inserted | Daniel Miller | 3/4/18 2:44 PM |

| Page 18: Deleted | Daniel Miller | 3/4/18 2:44 PM |

| Page 18: Inserted | Daniel Miller | 3/4/18 2:44 PM |

on

| Page 18: Deleted | Daniel Miller | 3/4/18 2:33 PM |

performed on

| Page 18: Inserted | Daniel Miller | 2/26/18 8:54 PM |

populations

| Page 18: Deleted | Daniel Miller | 2/26/18 8:54 PM |

samples

| Page 18: Inserted | Daniel Miller | 3/4/18 2:36 PM |

50 m

| Page 18: Inserted | Daniel Miller | 2/26/18 8:54 PM |

s within

| Page 18: Deleted | Daniel Miller | 2/26/18 8:54 PM |

of

| Page 18: Deleted | Daniel Miller | 2/13/18 1:38 PM |

2WT

| Page 18: Inserted | Daniel Miller | 2/13/18 1:38 PM |

VW

[revised manuscript text omitted]

Figure changed and updated.

| Page 42: Comment [11] | Daniel Miller | 4/26/18 6:53 PM |
|---|---|---|

Figure updated with new vertically weighted retrieval information, font sizes increased, and background adjusted.

| Page 42: Inserted | Daniel Miller | 4/27/18 12:01 AM |
|---|---|---|

| Page 42: Deleted | Daniel Miller | 2/14/18 11:45 AM |
|---|---|---|

| Page 42: Inserted | Daniel Miller | 3/4/18 4:47 PM |
|---|---|---|

bispectral

| Page 42: Deleted | Daniel Miller | 3/4/18 4:47 PM |
|---|---|---|

SWIR

| Page 42: Deleted | Daniel Miller | 2/13/18 1:38 PM |
|---|---|---|

2WT

| Page 42: Inserted | Daniel Miller | 2/13/18 1:38 PM |
|---|---|---|

VW

| Page 43: Inserted | Daniel Miller | 4/27/18 12:01 AM |
|---|---|---|

| Page 43: Deleted | Daniel Miller | 2/14/18 11:45 AM |
|---|---|---|

| Page 44: Inserted | Daniel Miller | 4/27/18 12:01 AM |
|---|---|---|

| Page 44: Deleted | Daniel Miller | 2/14/18 11:45 AM |
|---|---|---|

**Header and footer changes**

**Text box changes**

**Header and footer text box changes**

**Footnote changes**

[revised manuscript text omitted]

Daniel Miller 2/13/2018 1:11 PM

Daniel Miller 2/13/2018 1:11 PM

Daniel Miller 2/13/2018 1:11 PM

Daniel Miller 2/13/2018 1:11 PM

Daniel Miller 2/13/2018 1:11 PM

Daniel Miller 2/13/2018 1:11 PM

Daniel Miller 2/13/2018 1:11 PM

Daniel Miller 2/13/2018 1:12 PM

Daniel Miller 3/4/2018 12:58 PM

Daniel Miller 2/13/2018 1:12 PM

Daniel Miller 2/13/2018 1:12 PM

Daniel Miller 2/27/2018 9:44 AM

Daniel Miller 2/27/2018 9:45 AM

Daniel Miller 2/27/2018 9:45 AM

Daniel Miller 2/13/2018 1:13 PM

Daniel Miller 2/13/2018 1:13 PM

Daniel Miller 3/4/2018 12:59 PM

Daniel Miller 2/26/2018 1:23 PM

Daniel Miller 2/26/2018 1:23 PM

Daniel Miller 2/26/2018 1:23 PM

[revised manuscript text omitted]

Daniel Miller 4/26/2018 10:52 PM

Daniel Miller 4/26/2018 10:52 PM

Daniel Miller 2/24/2018 12:28 PM

Daniel Miller 3/4/2018 4:45 PM

Daniel Miller 2/24/2018 12:28 PM

Daniel Miller 4/26/2018 10:55 PM

Daniel Miller 2/24/2018 12:28 PM

Daniel Miller 2/13/2018 10:16 AM

Daniel Miller 2/14/2018 3:30 PM

Daniel Miller 2/14/2018 3:31 PM

Daniel Miller 2/14/2018 3:31 PM

Daniel Miller 2/14/2018 3:32 PM

Daniel Miller 2/14/2018 3:31 PM

Daniel Miller 2/14/2018 12:54 PM

Daniel Miller 2/14/2018 12:54 PM

Daniel Miller 2/24/2018 8:25 PM

Daniel Miller 2/14/2018 12:55 PM

polarimetric retrieval, resulting in histograms that clearly show the two retrievals diverging from one another with increasing sub-pixel inhomogeneity, tends to result in larger biases. In contrast, the polarimetric $r_e(\text{pol})$ retrieval in Figure 10 (c) does not appear to have a clear systematic bias. The $v_e(\text{pol})$ retrieval in Figure 10 (d) tells a more complicated story, the median value of the bias is clearly close to zero, but there is a tendency toward low biased retrievals with increasing

5  inhomogeneity. It should be noted that the $v_e(\text{VW})$ itself increases with increasing $H_\sigma$, which is presumably a consequence of the anticorrelation between $\tau$ and $v_e(\text{VW})$. This might explain why for large values of $H_\sigma$, where the $v_e(\text{VW})>0.15$ population is more common, there are more negative biases.

To further emphasize how unresolved inhomogeneity can influence these two retrieval techniques, we will highlight a particularly inhomogeneous pixel from the ATEX clean case at the coarsest resolution (800m). Focusing first on the

10  bispectral retrieval using the 2.13 μm SWIR band, the LUT scatterplot in Figure 11 (a) reveals that there is significant variability in the sub-pixel (i.e., 50 m) VNIR reflectances, indicated by a large value of the sub-pixel inhomogeneity index ($H_\sigma$=0.5637). In contrast to the variability of VNIR reflectances, the microphysical properties are largely homogeneous in this 800 m pixel, indicated by the narrow distribution of sub-pixel $r_e(\text{VW})_{50\,m}$ (color of the points). The sub-pixel mean of $<r_e(\text{VW})>_{50\,m}$ =19.71 μm agrees well with the mean of both sub-pixel retrievals, $<r_e(2.13\ \mu m)>_{50\,m}$=18.73 μm and

15  $<r_e(\text{pol})>_{50\,m}$=18.92 μm. This combination of optical inhomogeneity and microphysical homogeneity leads to an average reflectance (indicated by the black star) for the 800 m pixel that falls significantly below the $r_e$=20 μm isoline (i.e., the closest isoline to the mean sub-pixel retrievals). Thus, the coarse resolution 800 m reflectance results in an 800 m bispectral retrieval with $r_e(2.13\ \mu m)_{800\,m}$=23.62 μm, which is biased high by ~4 μm. This effect is attributable to the well-documented PPH bias induced by the curvature of the bispectral LUT with respect to the optical thickness (Zhang and Platnick, 2011;

20  Zhang et al., 2012; 2016). The PPH bias has a stronger influence on the 2.13 μm retrieval compared to the 3.75 μm retrieval (shown in Figure 11 (b)) because the curvature of the LUT space is more pronounced.

The polarimetric retrieval has a fundamentally different relationship to the unresolved sub-pixel inhomogeneity. This can be demonstrated with the sub-pixel polarized reflectance histogram in Figure 11 (b). The reflectances in this figure have been binned by scattering angle to create a distribution of polarized reflectances for the 50 m sub-pixels within the

25  selected 800 m pixel footprint. Within the plot there are also two curves, shifted in amplitude away from the histogram for clarity, that display the mean 800 m multi-angular polarized reflectance and corresponding 800 m retrieved polarized phase function (with appropriate fitting coefficients). Note that, while this histogram gives a sense of the variability of the magnitude and scale of the polarized reflectances, what ultimately matters for the coarse resolution polarimetric retrieval is the relative shape of the 800 m averaged polarized reflectance curve. It is evident from this histogram and these curves that

30  the mean angular position of the supernumerary bow does not shift, indicating that there is no significant difference between $r_e(\text{pol})_{800\,m}$, $<r_e(\text{pol})_{50\,m}>$, and $<r_e(\text{VW})_{50\,m}>$. This agrees with previous studies on the impact of unresolved inhomogeneity on polarimetric $r_e$ retrievals (Shang et al., 2015). In contrast, there is clear variability in the amplitude of sub-pixel polarized reflectances. This variability owes itself to both optical ($\tau$), and microphysical inhomogeneity (i.e., $v_e(\text{VW})>0.15$) within the coarse resolution pixel. For thin clouds ($\tau<3$) the supernumerary bow amplitude is dependent on both $\tau$ and $v_e$ (Alexandrov et

Daniel Miller 2/14/2018 3:39 PM
Deleted: panels (a) and (b) of Figure 8 compare $r_e(\text{pol})$ to the $r_e(2.13\ \mu m)$ and $r_e(3.75\ \mu m)$ respectively at increasingly coarsened spatial resolutions (as indicated by the size of the circles). In addition to the spatial resolution, these plots also indicate the magnitude the of sub-pixel ... [11]

Daniel Miller 2/24/2018 8:34 PM

Daniel Miller 2/26/2018 12:38 PM

Daniel Miller 2/24/2018 8:34 PM

Daniel Miller 2/14/2018 3:40 PM

Daniel Miller 2/24/2018 12:28 PM

Daniel Miller 2/24/2018 12:10 PM

Daniel Miller 2/24/2018 12:10 PM

Daniel Miller 2/13/2018 1:38 PM

Daniel Miller 2/13/2018 1:38 PM

Daniel Miller 4/26/2018 10:56 PM

Daniel Miller 2/25/2018 8:47 PM

Daniel Miller 4/26/2018 10:57 PM

Daniel Miller 2/16/2018 1:04 PM

Daniel Miller 2/24/2018 12:28 PM

Daniel Miller 2/26/2018 12:58 PM

Daniel Miller 2/26/2018 12:58 PM

Daniel Miller 2/26/2018 12:58 PM

Daniel Miller 2/16/2018 1:07 PM

Daniel Miller 2/13/2018 1:38 PM

Daniel Miller 2/13/2018 1:38 PM

[revised manuscript text omitted]

Daniel Miller 2/12/2018 11:29 AM

**Comment [1]:** Figure updated to correct error in original axes label of panel (a).

**Figure 1: Demonstrations of the microphysical sensitivity of the bispectral and the polarimetric techniques. Panel (a) features the bispectral LUT exhibiting sensitivity to $r_e$ (colored iso-lines), due to absorption in the SWIR/ reflectance. The VNIR reflectances provide sensitivity to optical thickness (gray iso-lines). Panels (b) and (c) demonstrate the sensitivity of polarimetric technique to $r_e$ and $v_e$ respectively. The supernumerary bow peaks of the polarized phase function ($-P_{12}$) shift and become narrower with increasing droplet size ($r_e$), whereas the peaks erode in magnitude for broadened droplet size distributions ($v_e$).**

[Figure]

**Figure 2:** The optical and microphysical properties ($\tau$, $r_e$, and $v_e$) of the LES cases examined in this study. The panels are arranged such that each LES case appears row-wise and the different properties are appear column-wise. Note that the vertically weighting functions used for the displayed re(VW) and ve(VW) correspond to single-scattering assumptions. Cloud-free masking in each of the images appears in gray. Refer to sections 2 and 3 for discussion and definition of each of these properties. Axes labels have been removed to enlarge each map, but the spatial dimensions of each scene are roughly 7 x 7 km (refer to section 3 for the specific resolutions of each LES case.)

[Figure]

Figure 3: Joint histogram regressions of $r_e$ and $\tau$ in all LES cases comparing the bispectral retrievals to the LES cloud microphysical properties. Panels (a) and (b) are regressions of the bispectral $r_e(2.13\ \mu m)$ and $r_e(3.75\ \mu m)$ retrievals against the physical analogue $r_e(VW)$. Panels (c) and (d) are regressions of the bispectral $\tau(2.13\ \mu m)$ and $\tau(3.75\ \mu m)$ retrievals against the physical $\tau(LES)$. Panels (e) and (f) display the regression of the bispectral $r_e(2.13\ \mu m)$ and $r_e(3.75\ \mu m)$ retrievals for only optically thick pixels ($\tau > 3$). Note that in each panel the correlation is quantified with a linear correlation coefficient (R) and the black and white contours encompass 66% and 95% of the population, respectively.

Daniel Miller 4/26/2018 3:00 PM
**Comment [2]:** Figure updated again to newest vertical weighted properties

Daniel Miller 2/25/2018 4:25 PM
**Comment [3]:** Figure updated to include data from the new vertical weighting defintions.

Daniel Miller 2/13/2018 2:35 PM

Daniel Miller 2/13/2018 1:38 PM

[Figure]

**Figure 4: Joint histogram regressions of $r_e$, $v_e$, and $\tau$ in all LES cases comparing the polarimetric retrievals to the LES cloud microphysical properties. Panel (a) depicts the regression of the polarimetric $r_e$(pol) retrieval against the physical analogue $r_e$(VW), while panel (b) is sub-selection of the same regression for low $v_e$. Panel (c) depicts the regression of the polarimetric $v_e$(pol) retrieval against the physical analogue $v_e$(VW), while panel (d) is a sub-selection of the same regression for thick clouds ($\tau$>3). Panel (e) depicts the regression of the polarimetric $\tau$(pol) retrieval against the physical analogue $\tau$(LES), while panel (f) is sub-selection of the same regression for low $v_e$. Note that in each panel the correlation is quantified with a linear correlation coefficient (R) and the black and white contours encompass 66% and 95% of the population, respectively.**

Daniel Miller 4/26/2018 9:45 PM
**Comment [4]:** Updated for new VW properties…

Daniel Miller 3/2/2018 11:31 AM
**Comment [5]:** Figure updated to include data from the new vertical weighting defintions.

Daniel Miller 2/13/2018 1:38 PM

Daniel Miller 2/13/2018 1:38 PM

[Figure]

**Figure 5:** Histograms of the percent retrieval bias of retrievals based on perturbed reflectances stated relative to unperturbed retrievals. Panel (a) displays retrieval biases for the bispectral $r_e$ retrieval. Panel (b) displays retrieval biases for the bispectral and polarimetric $\tau$ retrievals. Refer to the text for more information about the polarimetric $r_e$ and $v_e$ retrieval biases.

[Figure]

Daniel Miller 4/26/2018 3:02 PM
**Comment [6]:** Figure updated for newest vertical weighting definition.

Daniel Miller 3/2/2018 12:13 PM
**Comment [7]:** Figure updated to include data from the new vertical weighting defintions.

Daniel Miller 2/23/2018 4:08 PM

[revised manuscript text omitted]

Daniel Miller 2/14/2018 11:45 AM